# Noncoding variants alter GATA2 expression in rhombomere 4 motor neurons and cause dominant hereditary congenital facial paresis

Hereditary congenital facial paresis type 1 (HCFP1) is an autosomal dominant disorder of absent or limited facial movement that maps to chromosome 3q21-q22 and is hypothesized to result from facial branchial motor neuron (FBMN) maldevelopment. In the present study, we report that HCFP1 results from heterozygous duplications within a neuron-specific *GATA2* regulatory region that includes two enhancers and one silencer, and from noncoding single-nucleotide variants (SNVs) within the silencer. Some SNVs impair binding of NR2F1 to the silencer in vitro and in vivo and attenuate in vivo enhancer reporter expression in FBMNs. *Gata2* and its effector *Gata3* are essential for inner-ear efferent neuron (IEE) but not FBMN development. A humanized HCFP1 mouse model extends *Gata2* expression, favors the formation of IEEs over FBMNs and is rescued by conditional loss of *Gata3*. These findings highlight the importance of temporal gene regulation in development and of noncoding variation in rare mendelian disease.

The noncoding human genome contains *cis*-regulatory elements (cREs) that can be bound by transcription factors (TFs) and act as cell-type-specific enhancers or silencers to define complex gene regulatory programs[1–3]. Recent advances have revealed that cRE variants may cause rare disease[4–6]; however, determination of the precise mechanism is difficult due to the need to study cREs in their relevant cellular and temporal context. Such studies are particularly challenging for developmental disorders where the fate of a small number of progenitors is defined by dynamic transcriptional states[7–13].

HCFP1 is a rare autosomal dominant disorder of absent or limited facial movement that was mapped to a 3-cM region of chromosome 3q21.2–22 (refs. 14,15). Neuropathology revealed a decreased number of FBMNs and facial nerve hypoplasia[16]. Sequencing of genes in the critical region, including *GATA2*, did not identify pathogenic coding variants[17].

In the present study, we report that HCFP1 results from noncoding variants within a cell-type-specific *GATA2* regulatory region. We identified two adjacent clusters of noncoding SNVs that alter a conserved cRE (cRE2) and overlapping tandem duplications of cRE2 and the adjacent *GATA2* enhancers, cRE1 and cRE3. We demonstrate that one cRE2 SNV cluster impairs binding of nuclear receptor subfamily 2 group F member 1 (NR2F1; COUP-TF1) and attenuates its repressive activity in a cell-specific manner. We show that GATA2, and its downstream effector GATA3 (refs. 18,19), are necessary to differentiate rhombomere 4 motor neurons (r4MNs) to IEEs but are dispensable for FBMN development. By contrast, a humanized cRE1 duplication mouse has ectopic expression of *Gata2* in developing FBMNs and this phenotype is rescued by genetically ablating *Gata3*. This mechanism highlights the importance of tight temporal control of TF expression in a cell-type-specific manner during development and supports whole-genome sequencing (WGS) to identify noncoding variation underlying rare Mendelian disorders.

✉e-mail: elizabeth.engle@childrens.harvard.edu

## Results

### Tandem duplications and noncoding SNVs at the HCFP1 locus

We enrolled families and simplex cases with nonsyndromic congenital facial paresis (CFP, cohort 1 US-based study) and performed genome-wide single-nucleotide polymorphism (SNP) analysis and whole-exome sequencing (WES) in two large dominant pedigrees, family 1 (Fam1) and family 9 (Fam9; Fig. 1a). SNP-based multipoint parametric linkage analysis assumed autosomal dominant inheritance and full penetrance yielded maximum lod (logarithm of odds) scores suggestive of linkage at an overlapping 63-Mb chr3 region encompassing the previously reported HCFP1 locus[14,15] (Fig. 1b and Extended Data Fig. 1a). WES analysis did not identify pathogenic coding variants within the suggestive regions of linkage in either family. To identify HCPF1 variants, we performed WGS from members of Fam1, Fam9 and seven additional HCFP pedigrees in cohort 1 (two vertical, one horizontal transmission and four simplex cases). Structural variation analysis[20] revealed 31-kb and 20-kb overlapping tandem duplications within the HCPF1 locus in Fam1 and Fam2 (de novo), respectively (Fig. 1a,b and Extended Data Fig. 1b,c). We next analyzed WGS for SNVs or indels (insertions and deletions) within the Fam1/Fam2 -18-kb minimum duplication region. Fam3, Fam7 and Fam9 each harbored a unique SNV within an ~270-bp, noncoding, conserved element (chr3:128,178,158–128,178,397; GRCh37/hg19). We resequenced and conducted double droplet PCR (ddPCR) of this element in the remaining cohort 1 probands: 2 pedigrees with vertical transmission, 4 sibling pairs and 31 simplex cases. SNVs were identified in dominant Fam4 and Fam8 and simplex Fam5 (de novo) and Fam6 (Fig. 1a,c and Extended Data Fig. 1d–f).

Cohort 2 (Europe-based study) included the two pedigrees that originally defined the HCFP1 locus[14,15], in whom we identified a 23-kb tandem duplication in Fam10 and an SNV in Fam14. WGS analysis of 14 additional probands in cohort 2 (4 vertical, 2 horizontal, 2 unknown transmission and 6 simplex cases) identified variations that segregated with affected individuals in three dominant pedigrees: duplications were detected in Fam11 and Fam12 and an SNV was detected in Fam13 (Fig. 1a–c and Extended Data Fig. 1b,e,f).

Each copy number variant (CNV) was fully penetrant and breakpoints were confirmed (Fig. 1a,b and Extended Data Fig. 1e). The five overlapping duplications defined a 12.7-kb minimum region (chr3:128,174,929–128,187,620) absent from the Database of Genomic Variants (DGV)[21] and gnomAD (v.2.1.1) (ref. 22). Fam2 and Fam10–12 had breakpoint microhomology, suggesting that they originated by replication-based, microhomology-mediated repair[23,24], whereas Fam1 had a three-nucleotide base-pair insertion (GAA) at the breakpoint (Extended Data Fig. 1e).

All seven SNVs fall within a conserved noncoding region and alter six highly conserved nucleotides located in two clusters (Fig. 1a and Extended Data Fig. 1f). Cluster A variants alter three adjacent nucleotides whereas Cluster B variants alter three of four adjacent nucleotides (Fig. 1c). Six SNVs are absent from gnomAD and other public databases, including chr3:128,178,298G>A, which appears to have risen independently in Fam9 and Fam14 (Extended Data Fig. 1b). By contrast, Fam7 and Fam8 share a rare ancestral haplotype flanking chr3:128,178,297A>G (Extended Data Fig. 1b), a variant present in six gnomAD v.3.1.2 individuals (rs987263273, minor allele frequency = $4 \times 10^{-5}$). Although Cluster A variants were fully penetrant, Cluster B variants in Fam7, Fam14 and possibly Fam6 had reduced penetrance.

### HCFP1 facial weakness is a neurogenic disorder

We examined a subset of participants to determine whether SNVs and duplications resulted in similar phenotypes. Among the 37 variant-positive participants with detailed phenotypic documentation, 2 were clinically unaffected and 4 had mild weakness but considered themselves unaffected (Fig. 1a and Supplementary Table 1). These six individuals all harbor SNVs, suggesting that SNVs can cause a milder phenotype. Among the 35 participants with visible facial weakness, 83% (29 of 35) had bilateral weakness, which was typically asymmetrical with regard to both sidedness and upper versus lower face, and facial nerves (cranial nerve VII) were hypoplastic on magnetic resonance imaging (MRI; Fig. 2a–q). Electromyography, nerve conduction studies, blink studies, acoustic stapedial reflex testing and auditory brainstem response studies were consistent with facial nerve neuropathy in the seven participants tested (Supplementary Clinical Note and Supplementary Tables 1 and 2). Thus, HCFP1 is neurogenic[16] and both SNVs and duplications cause nonsyndromic, mild-to-moderate severity CFP, supporting a shared neurodevelopmental mechanism.

### Variants alter cREs within a *GATA2* regulatory region

All five CNVs duplicate highly conserved noncoding regions that we refer to as cRE1, cRE2 and cRE3 located 3′ of *GATA2* and flanking *DNAJB8*. All seven SNVs are located within cRE2 (Fig. 1b,c and Extended Data Fig. 2a). *GATA2* encodes a pleiotropic TF that regulates numerous genes critical for embryonic development and neuronal cell fate[25,26] and haploinsufficiency results in blood and immune disorders. Multiple cREs contribute to regulation of *GATA2* expression in the blood, kidney and brain[27,28]. Among these, cRE1 and cRE3 function as enhancers and drive β-galactosidase expression in mice in a pattern recapitulating native *Gata2* expression, including in r4 of the developing hindbrain[29]. Examination of published data[1,30,31] (Extended Data Fig. 2b) reveals that *GATA2*, but not *DNAJB8*, is transcribed in many cell types. The cRE1–3 overlaps with regions of chromatin open only in neuroblastoma cell lines, where *GATA2* is also transcribed. Published chromatin immunoprecipitation sequencing (ChIP-seq) experiments in neuroblastoma lines show binding of GATA2 and GATA3 to cRE1 and cRE3, but not cRE2 (Extended Data Fig. 2c)[1,32]. These data highlight co-regulation and cell-type specificity of cRE1–3 and support them as part of a *GATA2* regulatory region in human neuroblastoma cell lines and in mice[29,33].

**Fig. 1 | Tandem duplications and noncoding heterozygous variants segregate with HCFP1. a,** Pedigrees of families 1–14. Above each pedigree is the chromosomal location of its CFP-causing variant. Below each individual is the pedigree position and, for participating individuals, the genotype for the variant allele (abbreviated pedigree is shown for Fam10, see ref. 15, and for Fam14, see ref. 14). For Fam1, -2 and -10 to -12, the WT allele is denoted by a black '+' and the duplication allele by a red 'dup'. For Fam3–9, -13 and -14, the WT and variant nucleotides are denoted by black and red letters, respectively. Squares show males, circles females; black fill shows affected and gray fill shows self-reported, unaffected but mild facial weakness on examination; and dotted square or circle shows nonpenetrant phenotype. **b,** Schematic genomic representation based on UCSC (University of California, Santa Cruz) Genome Browser output. Gray horizontal bars above chr3 ideogram denote previously reported HCFP1 linkage regions (chr3:127,454,048–130,530,963, all human coordinates are from GRCh37/hg19) (refs. 14,15) for Fam10 and Fam14, and regions consistent with linkage for Fam1 and Fam9 (63 Mb minimum overlap chr3:76,924,329–140,632,237). Under the ideogram are: GRCh37/hg19 nucleotide positions; thick blue horizontal bars denoting Fam1, -2 and -10 to -12 overlapping duplications; genes in the region; structural variants in the DGV (blue duplications, red deletions); and conservation based on the PhyloP score. Hg19 genomic coordinates are: *GATA2* (chr3:128,198,270–128,212,044), cRE1 (chr3:128,176,017–128,176,396), cRE2 (chr3:128,178,158–128,178,397) and cRE3 (chr3:128,187,090–128,187,620). **c,** Magnification of the sequence and multispecies alignment of the cRE2-conserved region harboring all seven SNVs. The WT nucleotide of each SNV is boxed with the family ID harboring an SNV indicated above the box. The two clusters of variants lie 32 bp apart and are labeled 'Cluster A' and 'Cluster B'. Multispecies alignment reveals, in mice, a 4-bp deletion between Cluster A and Cluster B, and lack of conservation of the Fam6 variant. See also Extended Data Figs. 1 and 2.

### *Gata2* and *Gata3* are regulators of IEE but not FBMN fate

The overall organization of the developing and mature facial nucleus is conserved between mice and humans[34] (Fig. 3a). In mice, *Hoxb1*

expression begins at approximately embryonic day 8.5 (~E8.5) and determines the identity of hindbrain r4 (refs. 19,35). FBMNs are born in the r4 ventricular zone between ~E9 and E12 and migrate caudally

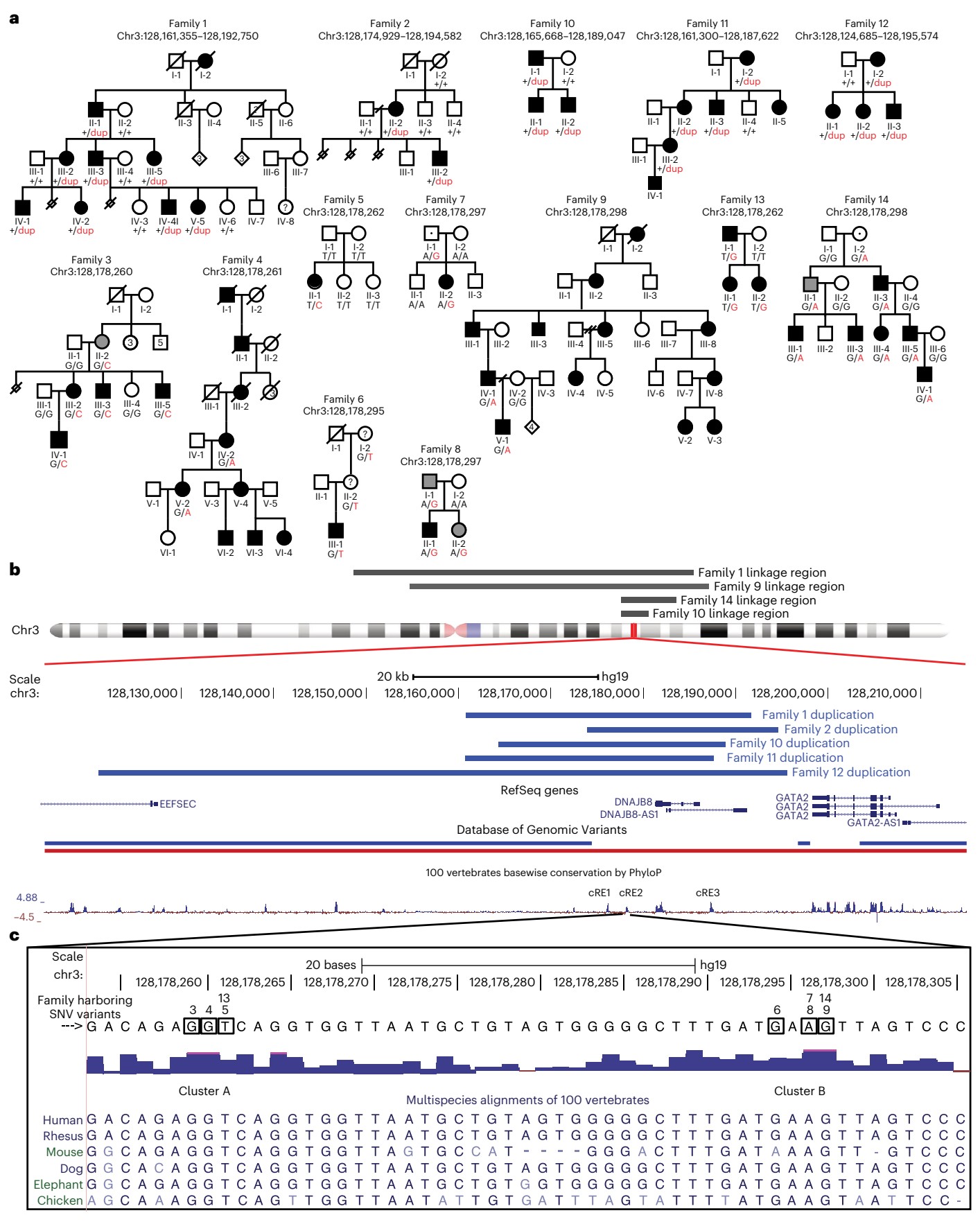

to r6 (refs. 36,37), while simultaneously extending axons into the periphery to form the facial nerve that innervates facial muscles[36–39]. FBMNs share their ventricular zone origin with a second population of r4 cholinergic 'motor neurons' (r4MNs), the IEEs. IEEs migrate laterally or contralaterally within r4, dividing into ventral olivocochlear neurons (OCNs) that modulate auditory gain and focus and dorsal vestibular efferent neurons (VENs) that may reduce sensitivity to self-induced head movements[40,41].

*Gata2* is expressed in r4 as early as E8.5 (ref. 19) and has been proposed to work through *Gata3* to regulate IEE and FBMN development under the control of HOXB1 (refs. 19,42–45). We found that expression of *Isl1*, a crucial determinant of motor neuron identity[46], marked both developing r4MNs and the stream of caudally migrating FBMNs (Fig. 3a,b). *Gata2* expression overlapped with *Isl1* in r4 and was prominent in parasagittal stripes of interneurons[19] but absent from migrating FBMNs (Fig. 3b).

The precise role of *Gata2* and *Gata3* in FBMN development has not been delineated due to early embryonic lethality of constitutive knockout mice[47]. To circumvent this, we crossed *Gata2^{KO/flox}* and *Gata3^{tlz/flox}* mice to Phox2b-Cre+ mice, conditionally deleting *Gata2* or *Gata3* from developing r4MNs[47–50]. IEEs were not visualized in either conditional knockout (cKO) mice at E14.5, based on the absence of ISL1 protein in r4MNs in appropriate anatomical positions compared with wild-type (WT) littermates (Fig. 3c,d,e,i). By contrast, embryonic facial motor nuclei appeared normal (Fig. 3f,g,h,i).

The mouse facial nerve innervates large, extrinsic muscles that displace the whisker pad and small, intrinsic muscles surrounding each vibrissal follicle[51]. To examine facial nerve function, we developed a semiquantitative whisking assay, collecting high-speed video recordings of vibrissal movement as mice ran on a treadmill, and scored left and right whisker movements (Fig. 3j). *Gata2^{KO/flox}*;Phox2b-Cre+ and *Gata3^{tlz/flox}*;Phoxb2-Cre+ mice showed full and indistinguishable whisking from WT (Fig. 3k and Supplementary Videos 1a–c). Thus, *Gata2* and *Gata3* are master regulators of IEE but not FBMN development.

## WT but not mutant cRE2 silences cRE1 and cRE3 in FBMNs

As HCFP1 duplications and SNVs cause the same phenotype in humans and cRE1 and cRE3 are *Gata2* enhancers in mice[29], we hypothesized that cRE2 was a cell-type-specific *Gata2* silencer[13,52]. If so, SNVs could weaken the silencing by attenuating TF binding and duplications could disrupt regulatory balance. Either would cause abnormal *Gata2* expression. To test this hypothesis in vivo, we evaluated whether different cRE combinations drove β-galactosidase expression when coupled to a *lacZ* reporter targeting a specific locus in the mouse genome[53]. We designed

donor DNA constructs containing different cRE combinations (Fig. 4a). The cRE1 alone drove β-galactosidase expression in the region of r4MN precursors and migrating FBMNs, as well as in midbrain and spinal cord (Fig. 4b,c and Extended Data Fig. 3a), similar to published data[29]. The cRE3 alone drove expression restricted to r4MNs, lateral r4 where migrating IEEs and nascent FBMN/IEE axons overlap, and migrating FBMNs (Fig. 4d and Extended Data Fig. 3b). Thus, although cRE1 and cRE3 enhance β-galactosidase expression in a *Gata2* pattern, they also mark *Gata2*-negative migrating FBMNs. By contrast, cRE2 alone did not

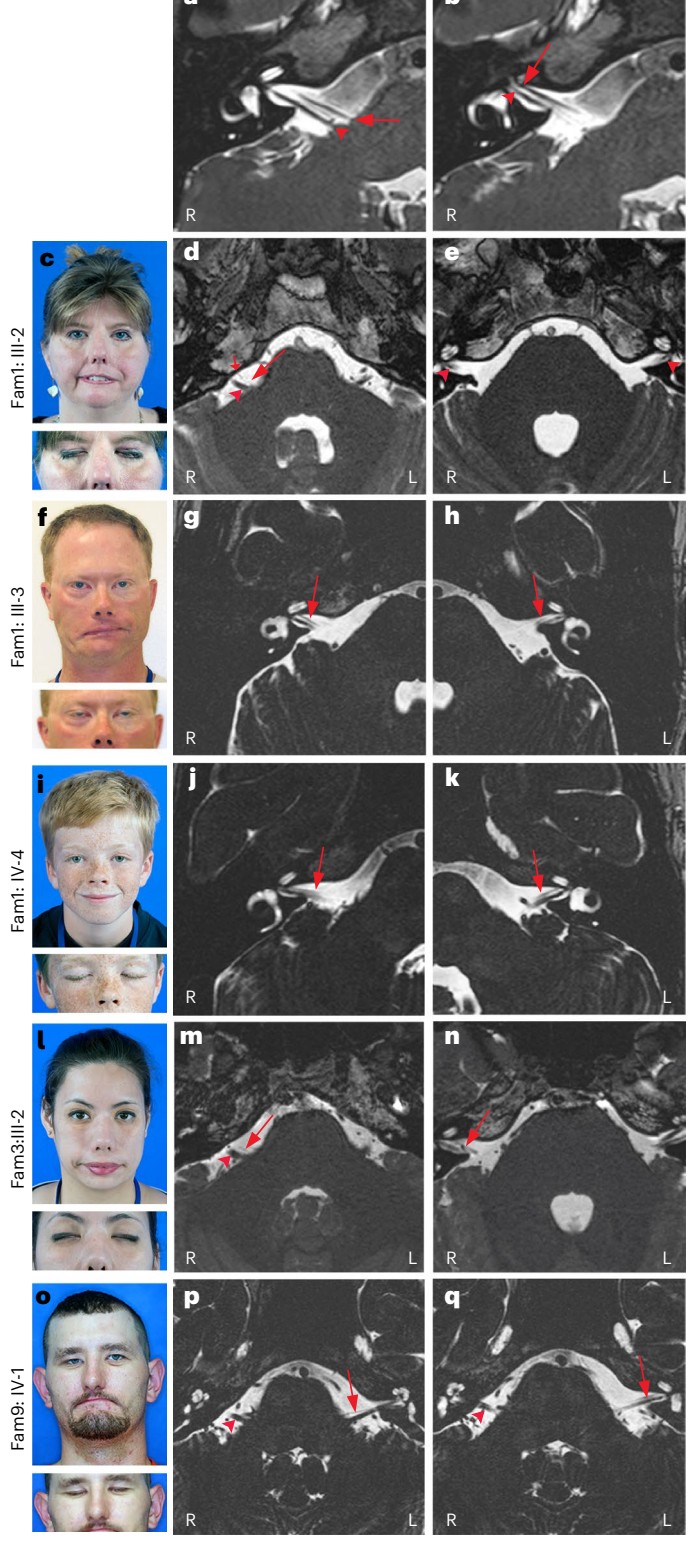

**Fig. 2 | HCFP1 phenotype and facial nerve MRI. c,f,i,l,o,** Photos of affected individuals attempting to smile (top) and close eyes (bottom) highlighting facial weakness (FW), lagophthalmos, absent forehead wrinkles and nasolabial folds, asymmetrical smile, upturned nasal tip and slit-like nares. **a,b,d,e,g,h,j,k,m,n,p,q,** MR images of facial nerve (VII, arrows) and vestibulocochlear nerve (VIII, arrowheads) in normal and HCFP1 individuals. R, right side; L, left side. **a,b,** Normal VII anatomy at the level of the right internal auditory canal (IAC) demonstrates origin and cisternal segments of right VII coursing parallel and ventral to VIII (**a**) and, more laterally, the right VII coursing through the IAC ventral to the superior vestibular branch of VIII (**b**). **c–e,** Fam1: III-2, L > R: FW, mild left lagophthalmos (**c**); markedly hypoplastic right and absent left VII (short arrow: anterior inferior cerebellar artery) (**d**); and VII not visualized within the IACs (**e**). **f–h,** Fam1: III-3, R > L: FW, bilateral lagophthalmos despite gold weight insertions (**f**); mild right VII hypoplasia (**g**); and left IAC narrowed, left VII markedly hypoplastic (**h**). **i–k,** Fam1: IV-4: asymmetrical R > L FW with good eyelid closure (**i**); and bilateral R > L VII hypoplasia (**j,k**). **l–n,** Fam3: III-2: bilateral L > R FW, R > L lagophthalmos (**l**); markedly hypoplastic right VII cisternal segment (**m**); mildly hypoplastic right VII IAC segment (**n**); and absent left VII cisternal segment (**m,n**). **o–q,** Fam9: IV-1: L > R FW, minimal lagophthalmos (**o**); right VII cisternal segment not visible; hypoplastic left VII cisternal and IAC segments (**p,q**).

drive β-galactosidase expression, consistent with silencing activity (Fig. 4e and Extended Data Fig. 3c). Combining cRE2 with cRE1 or cRE3, we detected β-galactosidase expression in r4MNs and migrating IEEs but no longer in migrating FBMNs, consistent with absence of *Gata2* expression in these cells (Figs. 3b and 4f,g and Extended Data Fig. 3d,e).

The cRE2 with Fam3-5 Cluster A SNVs, when combined with cRE1 (cRE1 + cRE2*A), no longer attenuated cRE1-driven *lacZ* signal in migrating FBMNs, indicating that these SNVs prevented cRE2-mediated silencing (Fig. 4h and Extended Data Fig. 3f). The effect of cRE1 with the three Cluster B SNVs (CRE1 + CRE2*B) was less clear, because the signal was attenuated in only one of eight embryos tested (Fig. 4i and Extended Data Fig. 3g). It is interesting that expression of cRE2-mutant clusters alone (cRE2*A or cRE2*B) showed some neuronal signal only in tandem, not single, transgenic embryos (Extended Data Fig. 3h,i). Similarly, cRE1 + cRE2*A showed an overall stronger and more intricate *lacZ* pattern compared with cRE1 + cRE2 (Extended Data Fig. 3d,f). Overall, these in vivo data support our hypothesis that HCFP1 SNVs disrupt a cell-specific regulatory element (cRE2) that normally downregulates *Gata2* expression in developing FBMNs.

## Cluster A SNVs attenuate binding of NR2F1 to cRE2

We performed in silico prediction of TF-binding sites conserved between the cRE2 of humans and that of mice[54]. Cluster B SNVs were not predicted to alter conserved TF-binding sites. By contrast, Cluster A SNVs alter three nucleotides (5′-A*GGT*CA-3′) of a consensus sequence of the COUP-TF family, NR2F1 and NR2F2 (Fig. 4j)[55]. *Nr2f1* is a determinant of cell-type specification and temporal fate of the developing cortical neurons and glia[55]. It is expressed throughout the hindbrain by E8.5 and enriched in facial and other cranial motor nuclei by E9.5 (refs. 56,57). Re-analysis of published ChIP-seq data from human induced pluripotent stem cell-derived cranial neural crest cells[58], which share a similar origin with neuroblastoma cells, revealed NR2F1 binding to cRE2 but not cRE1 or cRE3 (Extended Data Fig. 2c). NR2F2 did not bind cRE2 in human cranial neural crest cells[59]. Notably the mouse, but not the human, cRE1 sequence contains a COUP-TF-binding site (mm10 chr6:88,226,527–88,226,549). This, together with a murine-specific 4-bp deletion between cRE2 Clusters A and B (Fig. 1c), suggests differential cRE1–cRE3 binding and function of COUP-TF in the two species.

We performed an electrophoretic mobility shift assay (EMSA) that both confirmed interaction of NR2F1 with cRE2 sequence and demonstrated attenuated interaction with HCFP1 Cluster A variants in vitro (Fig. 4k and Extended Data Fig. 4a–f). To evaluate the effect of cRE2 Cluster A SNVs in vivo, we generated a knockin mouse carrying the Fam5 SNV (Extended Data Fig. 5a). *Fam5^snu/snu^* mice (chr6:88,224,892A>G) were viable and fertile and had normally developed facial motor nuclei and whisking (Fig. 3k, Supplementary Video 1d and Extended Data Fig. 5b–e). Despite the absent phenotype, conservation between mouse and human Cluster A sequences led us to test whether NR2F1 bound to WT Cluster A in r4MNs in vivo and whether the Fam5 SNV disrupted this interaction.

We dissected and FAC-sorted green fluorescent protein-positive (GFP⁺) cells from the r4 hindbrain of E10.5 WT;Isl1^MN^-GFP and *Fam5^snu/snu^*; Isl1^MN^-GFP embryos, in which GFP specifically labels motor neurons[60], and performed single-cell CUT&Tag[61] using an anti-NR2F1 antibody (Fig. 5a,b). We detected specific binding of NR2F1 to WT cRE1, cRE2 and, to a lesser extent, cRE3. By contrast, *Fam5^snu/snu^* r4MNs showed reduced cRE2 peak height compared with WT, without change in cRE1 and cRE3 peaks (Fig. 5c). Together, this shows that NR2F1 binds cRE2 in vitro and in r4MNs, and Cluster A SNVs attenuate this binding.

## Mice heterozygous for a humanized cRE1 duplication have HCFP

We generated a human cRE1 duplication mouse by inserting tandem copies of the human cRE1 sequence between mouse cRE1 and cRE2 (Extended Data Fig. 5f). We chose this approach because the cRE1 NR2F1-binding site in mice but not humans could alter the mouse phenotype. Mice heterozygous for the human cRE1 duplication (*cRE1^dup/+^*) were viable and fertile, and had absent whisker movement consistent with HCFP1 (Fig. 3k and Supplementary Video 1e).

## *Gata2* expression is altered in developing *cRE1^dup/+^* r4MNs

To identify transcriptomic changes in nascent and migrating FBMNs and IEEs caused by duplication of cRE1, we performed single-cell RNA-sequencing (scRNA-seq) on dissociated, FAC-sorted, GFP⁺ and the surrounding negative cells from hindbrain axial levels r3–r7 of E9.5–E12.5 *cRE1^dup/+^*;Isl^MN^-GFP and WT;Isl^MN^-GFP littermates (Extended Data Figs. 6a and 7). We limited bioinformatic analysis to *Isl1⁺* and/or *Hoxb1⁺* cells, thus focusing on developmental trajectories of r4 and neighboring Isl^MN^-GFP-expressing motor neurons[19,46].

Informed by known cell identity markers and those identified in the present study, we merged data from both genotypes, classified 16 clusters on the Unifold Manifold Approximation and Projection (UMAP) plot and found that clustering and cell-cycle phase were similar between the two genotypes (Fig. 6a–c, Extended Data Fig. 6a,b and Supplementary Table 3). Clusters 1–6 defined a developmental trajectory of r4MNs comprising mitotic progenitors of r3–r7 neurons (Cluster 1) through to bipotent r4MNs (Cluster 4) that gave rise to IEEs (Cluster 5) and FBMNs (Cluster 6) (Fig. 6a–c and Extended Data Fig. 6c,d). Cluster 5 IEE cellular density was increased whereas Cluster 6 FBMN cellular density was decreased in *cRE1^dup/+^* embryos compared with WT (Fig. 6a–c). *Dnajb8* was not expressed in any clusters of either genotype (Extended Data Fig. 6c,d).

Differential expression analysis revealed *Gata2* and *Gata3* as the transcripts most enriched in *cRE1^dup/+^* Clusters 1–6 compared with WT (Fig. 6d). The downregulation of *Gata2* expression in WT Cluster 4 between E9.5 and E10.5 was not observed in *cRE1^dup/+^* embryos (Fig. 6e,f). In both genotypes, *Nr2f1* expression marked r4 progenitors and was maintained across the trajectory, declining only in maturing IEEs (Fig. 6e,f), whereas *Nr2f2* was initially expressed in r4 progenitors

---

**Fig. 3 | Conditional loss of *Gata2* or *Gata3* prevents IEE development but does not impede FBMN development. a**, Migration schema of OCN (orange) and VEN (pink) IEEs and FBMNs (blue). **b**, E11.5 whole-mount *Isl1* and *Gata2* in situ hybridization: r4MN progenitor zone (black arrowheads), caudally migrating FBMNs (black arrows), parasagittal interneuron column (yellow arrowheads), developing inner ear (yellow arrows) (*n* = 3 WT, 10 *cRE1^dup/+^* embryos). Scale bar, 200 μm. **c–h**, ISL1 (blue), GATA2 (red) and GATA3 (green) immunofluorescence on E14.5 WT (**c**,**f**), conditional *Gata2^KO/flox^*;Phox2b-Cre⁺ (**d**,**g**) and *Gata3^tlz/flox^*; Phox2b-Cre⁺ KO hindbrains at r4 (**c**–**e**) and r6 (**f**–**h**). White arrows show OCN IEEs, yellow arrowheads show interneurons and the white arrowhead shows the trigeminal motor nucleus. Blue (r4) and white (r6) boxed regions are magnified below with a dotted oval denoting OCN IEE location (*n* = 3 (**c**,**f**), 6 (**d**,**g**) and 3 (**e**,**h**)). The borders of the hindbrain are outlined in gray. Scale bar, 200 μm (**c**) and applies to **c**–**h**. **i**, Schematics of E14.5 hindbrain cytoarchitecture based

on **c**–**h** as viewed ventrally (left) and in cross-section at the level of r4 (middle) and r6 (right) in WT (left side of each schema) and *Gata2* or *Gata3* cKOs (right side of each schema). ISL1^ON^;GATA2^ON^ IEEs (orange neurons) were absent from cKOs whereas ISL1^ON^;GATA2^OFF^ FBMNs (gray) appeared normal. **j**, Whisking assay schematic. **k**, Whisker movement assessment. Both left and right whiskers scored 3 for all WT (*n* = 5 male (M), 4 female (F), *Gata2^KO/flox^*;Phox2b-Cre⁺ (*n* = 2 M, 3 F), *Gata3^tlz/flox^*;Phoxb2-Cre⁺ (*n* = 2 M, 4 F) and *cRE2 Fam5^snu/snu^* (*n* = 2 M, 4 F) mice. Both left and right whiskers scored 0 for all *cRE1^dup/+^* mice (*n* = 8 M, 10 F). Of the *cRE1^dup/+^*;*Gata3^tlz/flox^*;Phox2b-Cre⁺ rescue mice (*n* = 1 M, 6 F), 2 had full (3) and 1 had no (0) whisker movement bilaterally, whereas the remaining 4 had intermediate movement (0 < *x* < 3). Pairwise, two-sided Bonferroni's corrected Wilcoxon's test (*P* values as shown). The filled circle shows mean and the error bar the s.e.m. Schemas in **j** were created with BioRender.com.

and subsequently upregulated in bipotent r4MNs, FBMNs and, to a lesser degree, IEEs (Fig. 6f). Last, although only a small number of *Nr2f1*-expressing neurons from E9.5 and E10.5 Cluster 4 bipotent r4MNs

coexpressed *Gata2*, the majority from *cRE1*[dup/+] embryos did (Fig. 6g). Thus, the *cRE1*[dup/+] scRNA-seq data revealed sustained *Gata2* expression in r4MNs normally destined to become FBMNs.

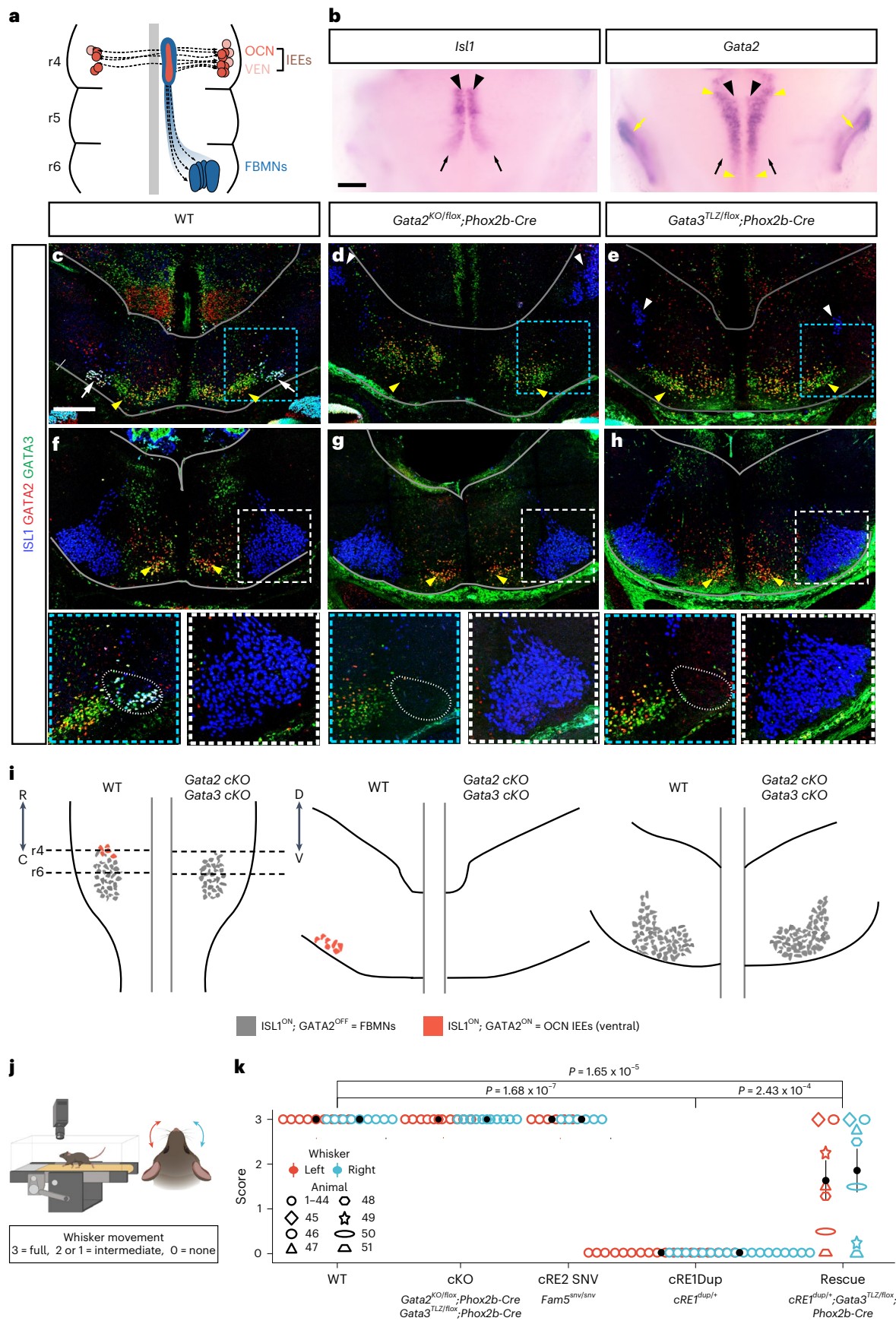

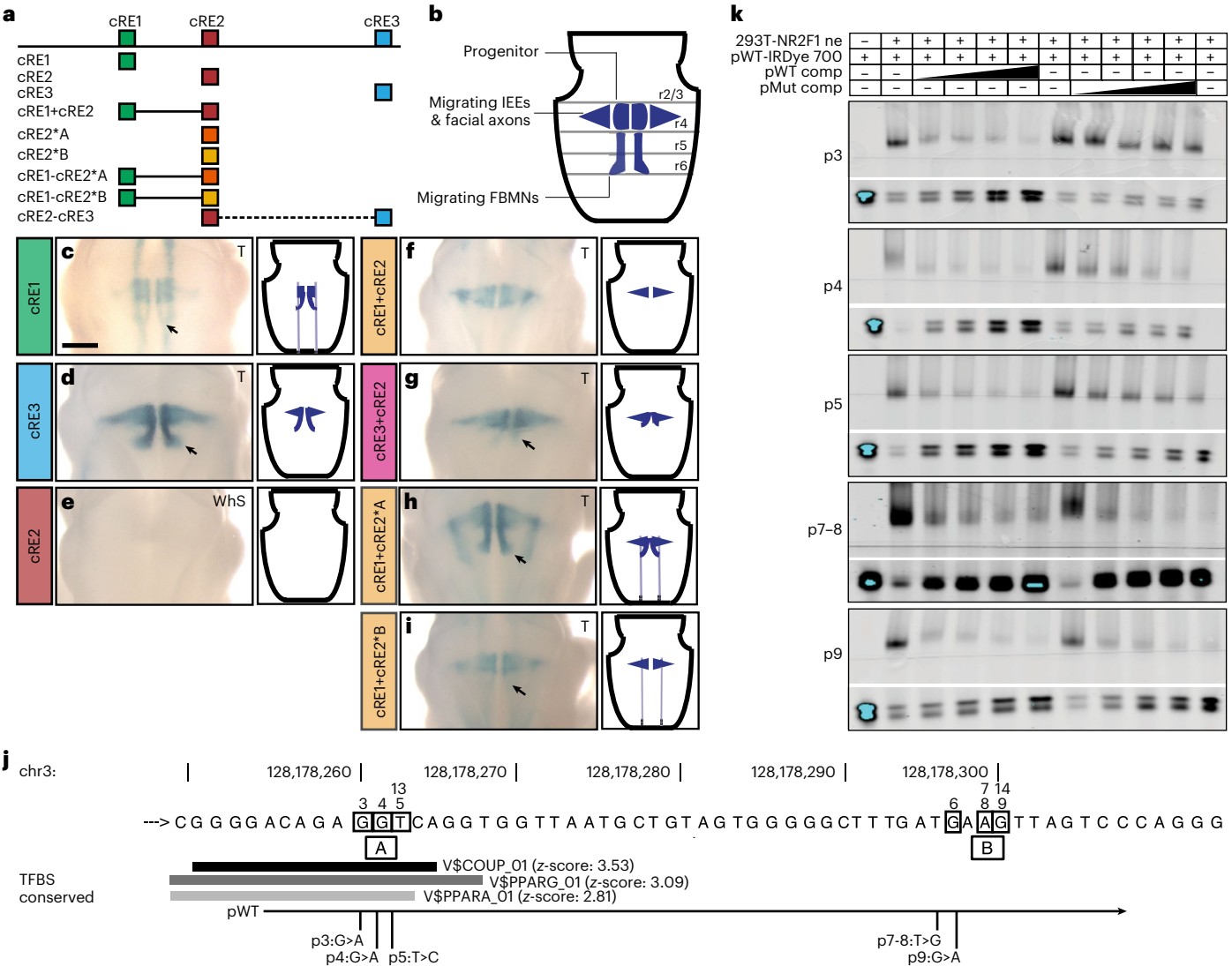

**Fig. 4 | Cluster A SNVs impair cRE2-mediated silencing in a reporter expression assay in vivo and reduce NR2F1 binding in vitro. a,b**, Schematics for in vivo *lacZ* reporter assay constructs (**a**) and hindbrain β-galactosidase expression viewed dorsally through the fourth ventricle (**b**). In **b**, midline ovals denote IEE/FBMN progenitors, triangles denote migrating IEEs and leg-like columns denote migrating FBMNs that are highlighted by black arrows in **c**, **d** and **g–i**. **c–i**, Selected images of ectopic β-galactosidase in transfected embryos (left) and schema (right): cRE1 alone (**c**, n = 13), cRE3 alone (**d**, n = 6) cRE2 alone (**e**, n = 8), cRE1 with cRE2 (**f**, n = 10), cRE3 with cRE2 (**g**, n = 7), cRE1 with cRE2 carrying Cluster A variants (**h**, n = 13) and cRE1 with cRE2 carrying Cluster B variants (**i**, n = 8). The asterisk denotes a mutant cluster. Scale bar (**c**), 500 µm and applies to **c–i**. Additional images are shown in Extended Data Fig. 3. **j**, Partial cRE2 sequence, as per Fig. 1. Gray horizontal bars denote overlap with in silico, conserved, transcription-binding consensus sequences from TRANSFAC (indicated by $). The shade of gray

correlates with a prediction *z*-score. WT (pWT) and mutant (pMut) EMSA probes are aligned below. TFBS, TF-binding sites. **k**, EMSA results showing the effect of SNVs on NR2F1-binding activity from transfected nuclear extract (293T-NR2F1 ne) in the presence of increasing molar excess (25× to 50× to 100× to 200× as denoted by black slope) of pWT or pMut competitor 'cold' probes compared with conjugated 'hot' probe (pWT-IRDye 700). For each SNV: NR2F1 binding (upper gel); free probe (bottom gel; lower and upper bands reflect unannealed and annealed probe, respectively). In all five experiments, pWT shows decreasing NR2F1 binding and increasing free probes. Cluster A variant competitor probes (p3, p4 and p5) compete less well than pWT for NR2F1 binding (more NR2F1 shifted and less free probe available). Cluster B variants (p7–8 and p9), where no NR2F1 binding is expected, show no substantial effect. The same trend was observed in replicate experiments: WT = 11; p3 = 5; p4 = 8; p5 = 4; p7-8 = 3; and p9 = 7. Full gels are given in Source data.

## GATA2 localization is expanded in developing *cRE1^dup/+* r4MNs

We used multichannel immunofluorescent staining of IEEs and FBMNs in E10.5–E16.5 r4–r6 hindbrain sections to determine whether changes in r4MN organization supported a WT IEE-to-FBMN developmental switch that was altered in *cRE1^dup/+* embryos. We focused on E14.5, when the broad contours of IEE and FBMN organization are first apparent and *Gata2* is not yet downregulated (Fig. 7, single channels in Extended Data Fig. 8).

In WT embryos at E10.5, FBMNs (defined as ISL1^ON;GATA2^OFF; GATA3^OFF) were distinguishable from IEEs (defined at this age as

ISL1^ON;GATA2^ON with variable GATA3 expression and at later ages as ISL1^ON;GATA2^ON;GATA3^ON) (Extended Data Fig. 9a,b). By E12.5, FBMNs formed dorsal clusters flanking the r4 midline, whereas GATA2 and GATA3 delineated smaller ventral populations of IEEs that were migrating laterally and ventrally to form the OCN nucleus. Bilateral columns of ISL1^OFF;GATA2^ON;GATA3^ON interneurons were detected between the midline r4MN clusters and developing IEEs[43] and NR2F1 expression was elevated in FBMNs and reduced in IEEs (Extended Data Fig. 9c–n). At E14.5, IEEs formed variably detected dorsal VEN clusters and more prominent ventral OCN clusters (Fig. 7a,b). FBMNs

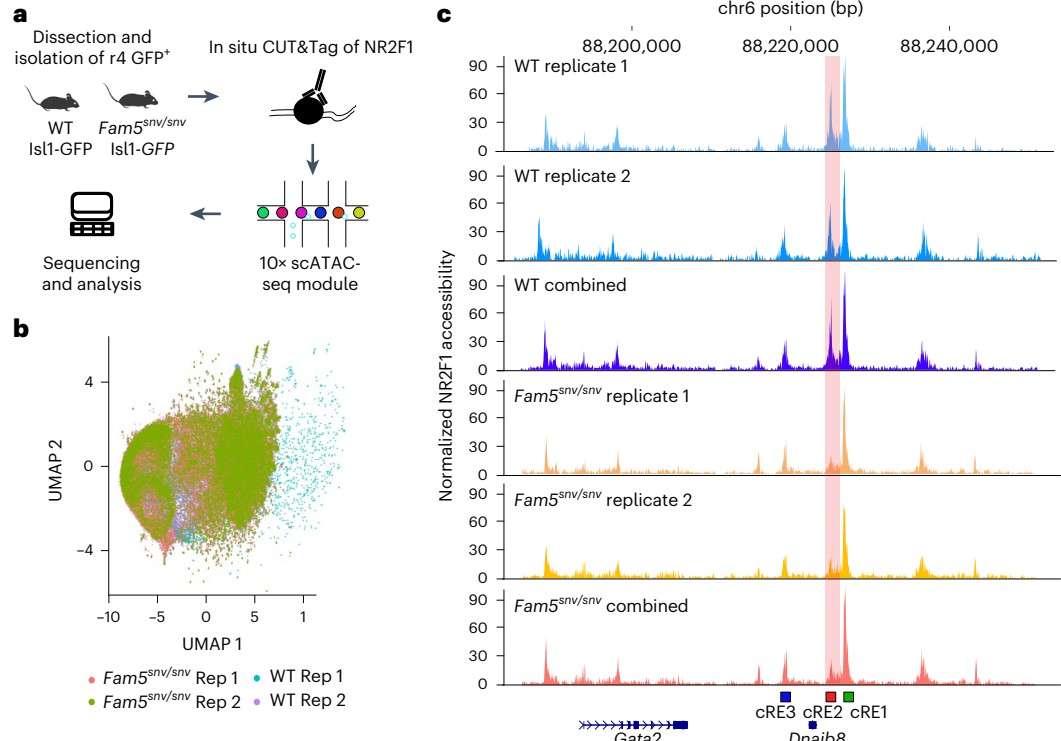

**Fig. 5 | NR2F1 binds cRE2 in E10.5 r4MNs and binding is reduced by Fam5 SNV.**
**a**, Schematic representation of single-cell CUT&Tag of E10.5 WT and *Fam5^snv* r4
*Isl1^+* neurons targeting NR2F1. scATAC-seq, single-cell assay for transposase-
accessible chromatin with high-throughput sequencing. **b**, UMAP embedding
of NR2F1 single-cell CUT&Tag experiment for two *Fam5^SNV/SNV* (replicate (Rep)
1 = 2,274 cells, Rep 2 = 2,740 cells) and two WT (Rep 1 = 2,572 cells, Rep 2 = 1,377)
age-matched biological replicates. **c**, Pseudobulk single-cell CUT&Tag profile of
NR2F1 around the *Gata2* regulatory region shown as individual and combined
replicates (WT in blue and *Fam5^SNV/SNV* in red–yellow). Location of the cREs
relative to mouse *Gata2* and *Dnajb8* is shown. Note the reduction in the height of
the cRE2 peaks (vertical pink shading) in the *Fam5^SNV/SNV* replicates.

had largely completed migration to form the facial motor nucleus
in the ventral r6 hindbrain and expressed NR2F1 and not GATA2
or GATA3, consistent with NR2F1 blocking IEE fate (Figs. 3a and 7c,d).
By E16.5, IEEs formed compact dorsal VEN and ventral OCN clusters
in which GATA2 was downregulated and FBMNs aggregated into
facial motor nuclei (Extended Data Fig. 9o–r).

In *cRE1^dup/+* embryos at E10.5, GATA2 and GATA3 expression
extended ectopically throughout r4MNs (Extended Data Fig. 9a,b).
By E12.5, most r4MNs had adopted an 'IEE' molecular identity with many
ectopically occupying the dorsal region of r4, and FBMNs expressed
NR2F1 but were reduced at the r4 midline compared with WT (Extended
Data Fig. 9c–n). At E14.5, OCNs occupied normal positions in the ventral
hindbrain but also extended caudally into r6 and a larger population of
ectopic 'IEEs' occupied positions in the dorsal hindbrain in the region
of WT VENs (Fig. 7e,f). Ectopic 'FBMNs' were scattered throughout r4
and also formed a hypotrophic facial nucleus that extended from r4 to
r6 (Fig. 7e–h; schema in Fig. 7i–k). At E16.5, the *cRE1^dup/+* ventral OCN
cluster extended ectopically into r6, the dorsal ectopic IEEs formed
an expanded VEN cluster and the facial nucleus appeared small to
absent (Extended Data Fig. 9o–r).

We quantified ectopic cell positions and changes in r4MN gene
expression caused by cRE1 duplication by determining the size and
position of ISL1^ON;GATA2^ON IEE and ISL1^ON;GATA2^OFF FBMN subpopu-
lations in E14.5 WT and *cRE1^dup/+* hindbrains. The average number of
r4-born motor neurons did not differ between genotypes (Fig. 7l).
However, although WT embryos generated a 1:9.3 ratio of IEE:FBMN
cells, the *cRE1^dup/+* embryo ratio was 1:1.3, with the number of IEEs adopt-
ing an OCN and VEN identity increasing over threefold and tenfold,
respectively (Fig. 7m,n). Last, *cRE1^dup/+* embryos had a 32% decrease

in FBMNs (Fig. 7m) and, although 92% of E14.5 WT FBMNs completed
migration into ventral r6, only 37% of *cRE1^dup/+* FBMNs had, with the
balance assuming ectopic positions in r4–5 (Fig. 7o).

To determine IEE and FBMN birthdates, we applied 5-ethynyl-
2′-deoxyuridine (EdU) in utero to litters containing WT and *cRE1^dup/+*
embryos across an E9.25–E10.5 time course. High levels of EdU indel-
ibly mark cells undergoing terminal cell division during the EdU pulse,
permitting us to classify and count E14.5 EdU-positive cells as IEEs or
FBMNs, regardless of position (Fig. 7p). After EdU injection at E9.25,
88% of WT and 85% of *cRE1^dup/+* r4-derived motor neurons adopted
the IEE fate. Application of EdU in E10.0 WT embryos marked nearly
equal proportions of IEEs (55%) and FBMNs (45%), but in *cRE1^dup/+*
embryos a greater proportion of labeled cells became IEEs (73%). With
EdU application at E10.5, 2% of WT versus 34% of *cRE1^dup/+*-labeled
r4MNs became IEEs.

As *Dnajb8* lies between cRE1 and *Gata2*, we evaluated it as an HCFP1
target gene. In situ hybridization with *Dnajb8* riboprobe revealed no
expression in developing WT or *Cre1^dup/+* hindbrain, whereas staining
with *Isl1* and *Gata2* probes recapitulated protein antibody staining
(Extended Data Fig. 10a–c). These observations are consistent with
scRNA-seq data and confirm that changes in *Dnajb8* expression are
unlikely to underlie HCFP1.

These data establish that the humanized duplication of cRE1 per-
turbs r4-derived MN expression of *Gata2* but not *Dnajb8*. They provide
evidence of an IEE-to-FBMN birth order, with a developmental switch
active from E9.25 to E10.5 in WT embryos that extends beyond E11.0 in
*cRE1^dup/+* embryos, producing IEEs at the expense of FBMNs. The 73%
reduction in FBMNs correctly positioned in the caudal hindbrain in
E14.5 *cRE1^dup/+* embryos probably underlies their facial paralysis.

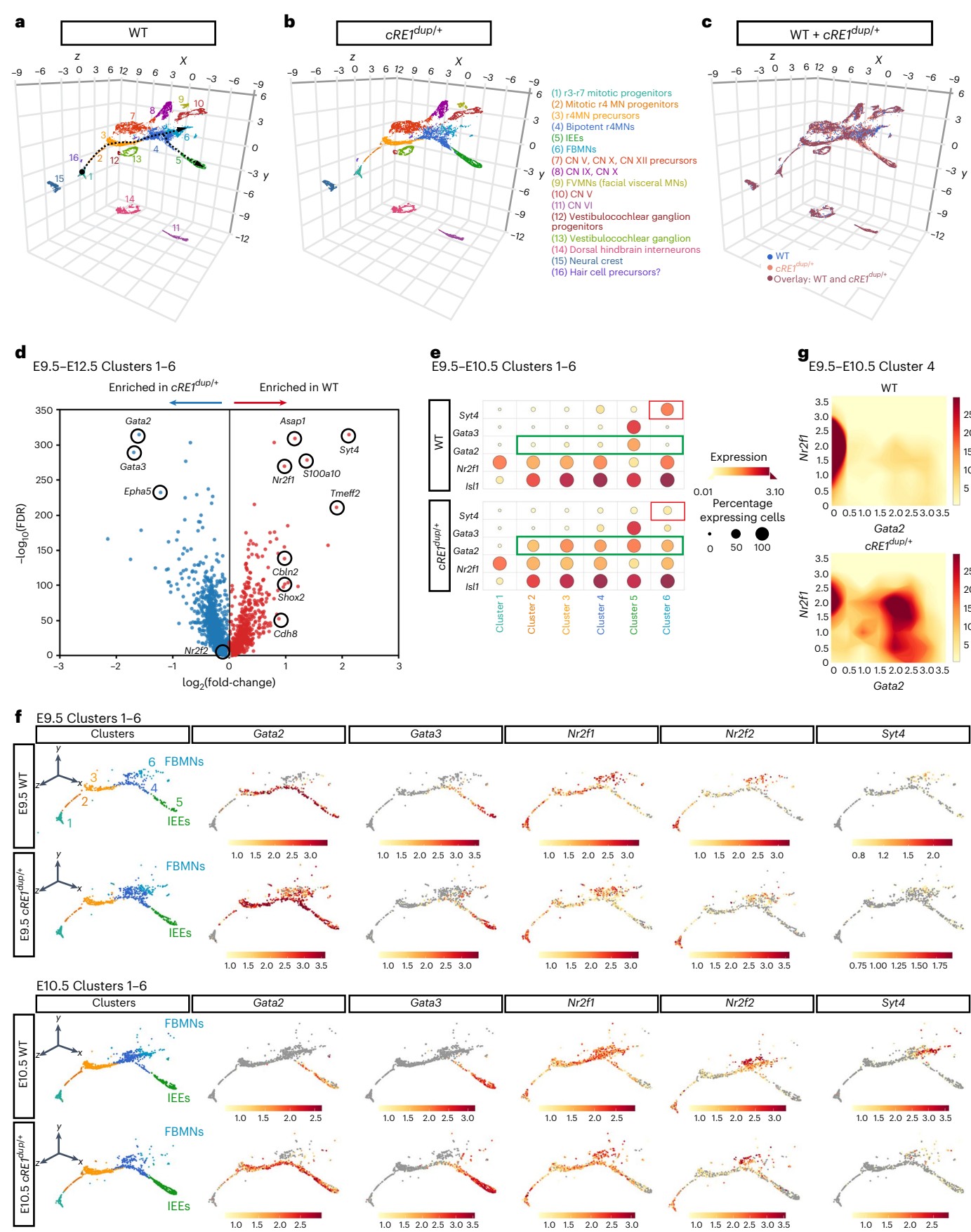

**Fig. 6 | Single-cell transcriptomic analysis of WT and *cRE1^(dup/+)* r4 motor neurons. a,b**, Three-dimensional (3D) UMAP plot of WT (**a**) and *cRE1^(dup/+)* (**b**) components of a E9.5–E12.5 scRNA-seq object comprising *Isl1*⁺ and/or *Hoxb1*⁺ FAC-sorted Isl1^(MN)-GFP cranial motor neurons (MNs) (with GFP⁻ cells spiked in) spanning r3–r7. Seurat clusters are numbered and annotated according to proposed cellular identity at the right. CN, cranial nucleus. The black dotted arrows trace the proposed pseudotime developmental trajectory of r4MNs from mitotic progenitors of r3–r7 neurons (Cluster 1), r4MN mitotic progenitors (Cluster 2) and r4MN precursors (Cluster 3), 'bipotent r4MNs' (Cluster 4), which gave rise to separate populations of IEEs (Cluster 5) defined by *Gata2* and *Gata3* expression[18,19], and FBMNs (Cluster 6) defined by *Syt4*, *Shox2* and *Cdh8* expression and enriched for *Nr2f1* (refs. 18,19,74,75) (Extended Data Fig. 6c,d). **c**, Overlapping feature plots of WT (blue, bottom layer) and *cRE1^(dup/+)* (peach, top layer) 3D UMAPs shown in **a** and **b**. Sixty percent opacity of *cRE1^(dup/+)* data points reveals WT data and highlights overlap of the genotypes (burgundy). **d**, Volcano plot of differential expression analysis between WT and *cRE1^(dup/+)* r4MN

trajectories across the E9.5–E12.5 timepoints. Circled genes display log(fold-change) > 1 and −log₁₀(FDR) > 200 or are additional genes of interest (where FDR is false recovery rate). **e**, Dotplot comparison of FBMN and IEE marker expression in E9.5–E10.5 Cluster 1–6 r4MN developmental trajectories in WT (upper) and *cRE1^(dup/+)* (lower) embryos. Red and green outlines highlight differences in *Syt4* and *Gata2* expression, respectively, between WT and *cRE1^(dup/+)* samples. Scales indicate the mean expression level and percentage expressing cells within each cluster. **f**, Feature plots of WT and *cRE1^(dup/+)* r4MN trajectory determinants and markers at E9.5 (upper two rows) and E10.5 (lower two rows). At E9.5 in both WT and *cRE1^(dup/+)* embryos, r4MN precursors, a subset of IEE-directed bipotent r4MNs and IEEs (Clusters 3–5), expressed *Gata2*, with additional ectopic expression seen in *cRE1^(dup/+)* FBMNs (Cluster 6). By E10.5, WT embryos expressed *Gata2* only in Cluster 5 IEEs, but *cRE1^(dup/+)* embryos maintained *Gata2* expression in Clusters 3–5. **g**, Density plots for *Nr2f1* and *Gata2* expression in E9.5–E10.5 WT and *cRE1^(dup/+)* r4MNs. See also Extended Data Fig. 6.

## Loss of *Gata3* in *cRE1^(dup/+)* mice partially rescues CFP

If *cRE1* duplication results in the HCFP1 phenotype by causing ectopic expansion of *Gata2* in r4MNs, then removal of *Gata2* from *cRE1^(dup/+)* mice should rescue the phenotype. Linkage disequilibrium prevented crossing the *cRE1^(dup)* allele on to the *Gata2^(KO/flox)*;Phox2b-Cre⁺ cKO background. As *Gata3* is a *Gata2* transcriptional target and conditional removal of *Gata2* or *Gata3* eliminates IEE generation but preserves FBMNs (Fig. 3), we tested whether conditional *Gata3* deletion would rescue the *cRE1^(dup)* CFP phenotype.

We evaluated whisking after conditional removal of *Gata3* from *cRE1^(dup/+)* mice. Six of seven *cRE1^(dup/+)*;*Gata3^(tlz/flox)*;Phox2b-Cre⁺ mice had variable and often asymmetrical rescue of whisking, ranging from subtle movement in subsets of whiskers to complete restoration of whisking (Fig. 3k and Supplementary Video 1f–h). Comparison of E14.5 histologies revealed that conditional removal of *Gata3* from *cRE1^(dup/+)* embryos eliminated the large r4 ectopic population of dorsal ISL1^(ON) (and ISL1^(ON);GATA2^(ON)) cells as well as IEEs, and generated an elongated column of FBMNs that extended into ventral r6 to form a structure closer in size and shape to the facial nucleus seen in WT controls (Fig. 8a–j). These data establish that human cRE1, in concert with cRE2 and cRE3, modulates the *Gata2–Gata3* axis that defines the IEE-to-FBMN switch, and human HCFP1 pathogenic variants probably alter this regulatory pathway (Fig. 8k).

## Discussion

We report that heterozygous noncoding SNVs and CNVs at the HCFP1 locus alter regulation of *GATA2* and account for >90% of autosomal dominant, nonsyndromic CFP. Remarkably, the SNVs alter six nucleotides located in two clusters within a conserved noncoding region that we refer to as cRE2, located 3′ of *DNAJB8* and *GATA2*. *DNAJB8* is not a triplosensitive gene (pTriplo score 0.22) (ref. 62) nor is it expressed in r4MNs or surrounding tissue in WT or *cRE1^(dup/+)* mice,

excluding its involvement in HCFP1. Instead, our data support cRE2 as a tissue-specific regulatory element to which NR2F1 binds, restricting r4MN *GATA2* expression to developing IEEs.

The importance of *Gata2* expression in an r4MN IEE-to-FBMN fate transition and the perturbation of its spatial and temporal hindbrain expression as the cause of HCFP1 are supported by our data and those of others. First, we established GATA2 and GATA3 as essential regulators of IEE fate and dispensable for FBMN development and migration. Second, we found that *Gata2* enhancers cRE1 and cRE3 drive reporter expression in migrating FBMNs where *Gata2* is not expressed and cRE2 silenced this expression. Moreover, this silencing is attenuated by HCFP1 SNVs. Although the cRE2 silencing mechanism remains unknown, cRE1–3 and *Gata2* are within the same regulatory region and the cREs might compete for binding to the *Gata2* promoter. Third, our humanized cRE1 duplication mouse model has CFP, and scRNA-seq and histology revealed ectopic *Gata2* expression in later-born *cRE1^(dup/+)* r4MNs that expanded the IEE and depleted the FBMN populations. This phenotype could be partially rescued by removal of *Gata3*. Last, monoallelic loss-of-function variants in *GATA2* and in the +9.5-kb blood *GATA2* enhancer element cause blood and immune dysfunction without facial weakness[63,64], consistent with altered, not reduced, GATA2 expression in HCFP1 and highlighting the importance of tissue-specific regulation.

Several lines of evidence support a cell-type-specific function of NR2F1 in r4MN IEE-to-FBMN fate transition and attenuation of this function in HCFP1. First, we demonstrated that NR2F1 binds to cRE1 and cRE2 in WT r4MNs, and binding to cRE2 is reduced in r4MNs isolated from mice carrying a Cluster A SNV. Second, we found dynamic expression of *Nr2f1* in developing FBMNs, with reduced expression in IEEs. Third, although human haploinsufficiency of *NR2F1* causes a variable phenotype characterized primarily by intellectual disability and optic nerve degeneration[65], several individuals are reported to have a thin facial nerve or mild facial weakness[66,67].

**Fig. 7 | GATA2 expression and IEE birth epoch are expanded in developing *cRE1^(dup/+)* hindbrain. a–h**, E14.5 WT (**a–d**) and *cRE1^(dup/+)* (**e–h**) hindbrain sections at r4 (**a,b,e,f**) and r6 (**c,d,g,h**) axial levels showing immunofluorescence with ISL1 (blue) and GATA3 (green) (**a–h**) together with GATA2 (red: **a,c,e,g**) or NR2F1 (red: **b,d,f,h**). Ectopic dorsal r4MNs are present in **e** and **f** compared with **a** and **b**. Dotted yellow and blue rectangles (**a,b,e,f**) surround IEE VEN and OCN regions, respectively, and are magnified below. Dashed white squares (**c,d,g,h**) surround facial nuclei and are magnified below. White arrows show OCNs and white arrowheads FBMNs. The borders of the hindbrain are outlined in gray. Scale bars, 200 μm (**a,c**) apply to **a**, **b**, **e** and **f**, and **c**, **d**, **g** and **h**, respectively (*n* = 3 (**a,c**), 3 (**b,d**), 8 (**e,g**) and 7 (**f,h**) embryos). **i–k**, Schematics of E14.5 hindbrain cytoarchitecture based on **a–h** as viewed ventrally (**i**) and in cross-section at the level of r4 (**j**) and r6 (**k**) in WT (left side) and *cRE1^(dup/+)* (right side) of hindbrains. **l–o**, Quantification of E14.5 r4MN transcriptional and positional identity in

*cRE1^(dup/+)* and WT littermates detected in confocal z stacks. Unilateral counts are presented; each point represents an individual embryo and each color a litter (color coded A–F) (*n* = 9 *cRE1^(dup/+)* and nine WT littermate pairs from six litters). On average per side, WT versus *cRE1^(dup/+)* embryos had: 9,470 versus 10,422 r4-born MNs (**l**); 903 versus 4,405 IEEs (**m**); 8,408 versus 5,691 FBMNs (**m**); 719 versus 2,478 OCNs (**n**); 184 versus 1,927 VENs (**n**); and 7,721 versus 2,098 FBMNs completing migration into ventral r6 (**o**). In the box plot, the center line is the median, the box limits represent 50% of the values and the whiskers represent 98% of the values. **p**, The r4MN birthdating in the 18-mouse cohort in **l–o** using in utero labeling of mitotic cells with thymidine homolog EdU in IEE (OCN + VEN) and FBMN (FBMN + r4 ectopic); definitions as per **m**. Point is the mean. For **l–p**, all indicated *P* values are calculated using two-sided, pairwise Student's *t*-test without correcting for multiple testing; the error bar = ± s.e.m. See also Extended Data Figs. 8 and 9.

We favor NR2F1 over NR2F2 as key to the IEE-to-FBMN switch. We found no evidence that NR2F2 binds to cRE2 in public databases[59], and it shows low expression in developing r4MN, despite being upregulated in lateral FBMNs at late embryonic stages[38]. NR2F2 appears important for metabolic and cardiac processes[68] rather than neuronal development[69] and NR2F2 haploinsufficiency in

humans is associated with congenital heart defects without reports of facial weakness[70].

It is of interest that we did not observe a CFP phenotype in the Cluster A *Fam5*[SNV/SNV] mice, despite alterations in NR2F1 binding. HCFP1 SNV variants are less penetrant than CNVs, and the *Fam5*[SNV] mouse may cause a perturbation too mild to cause CFP. It is also

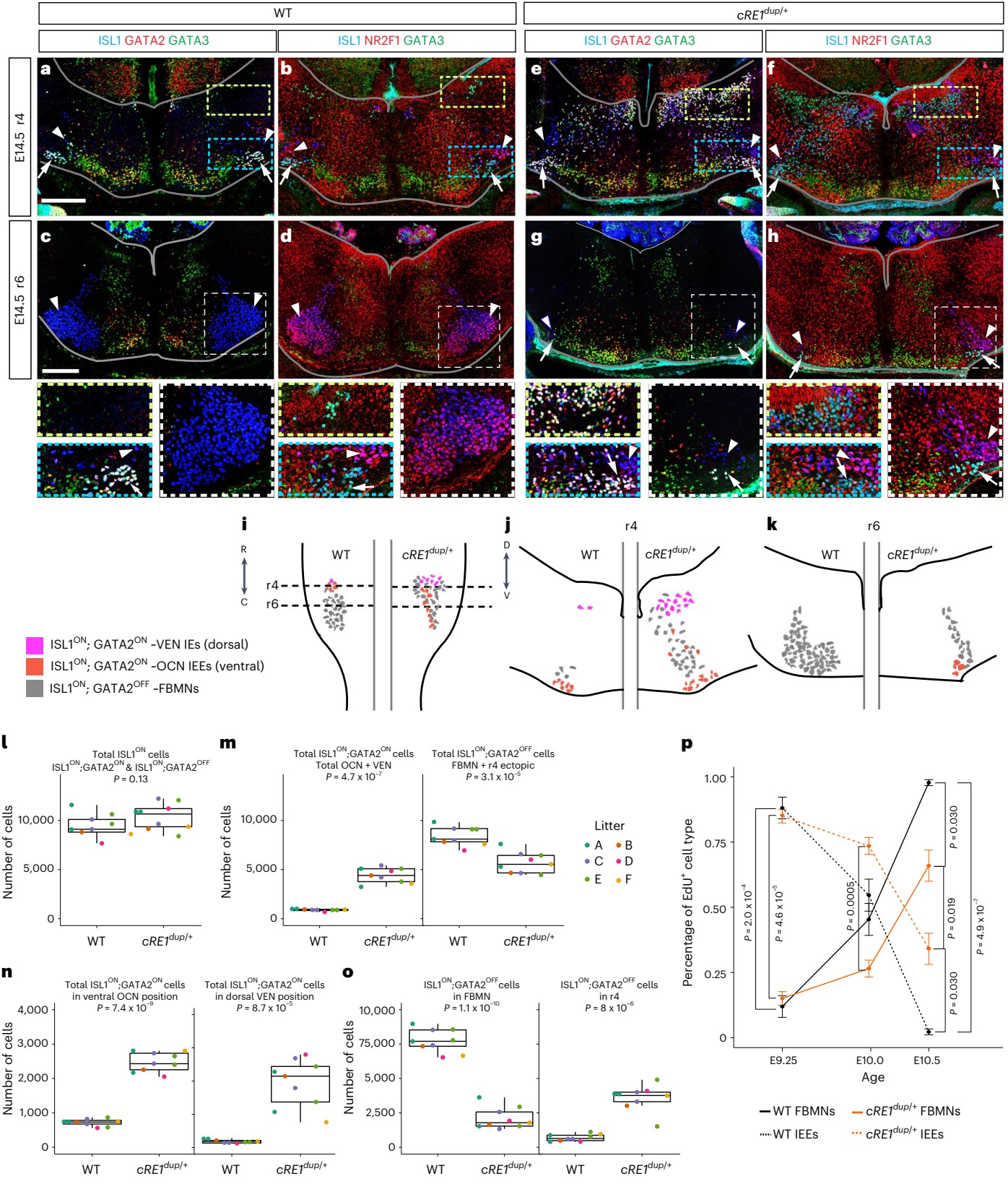

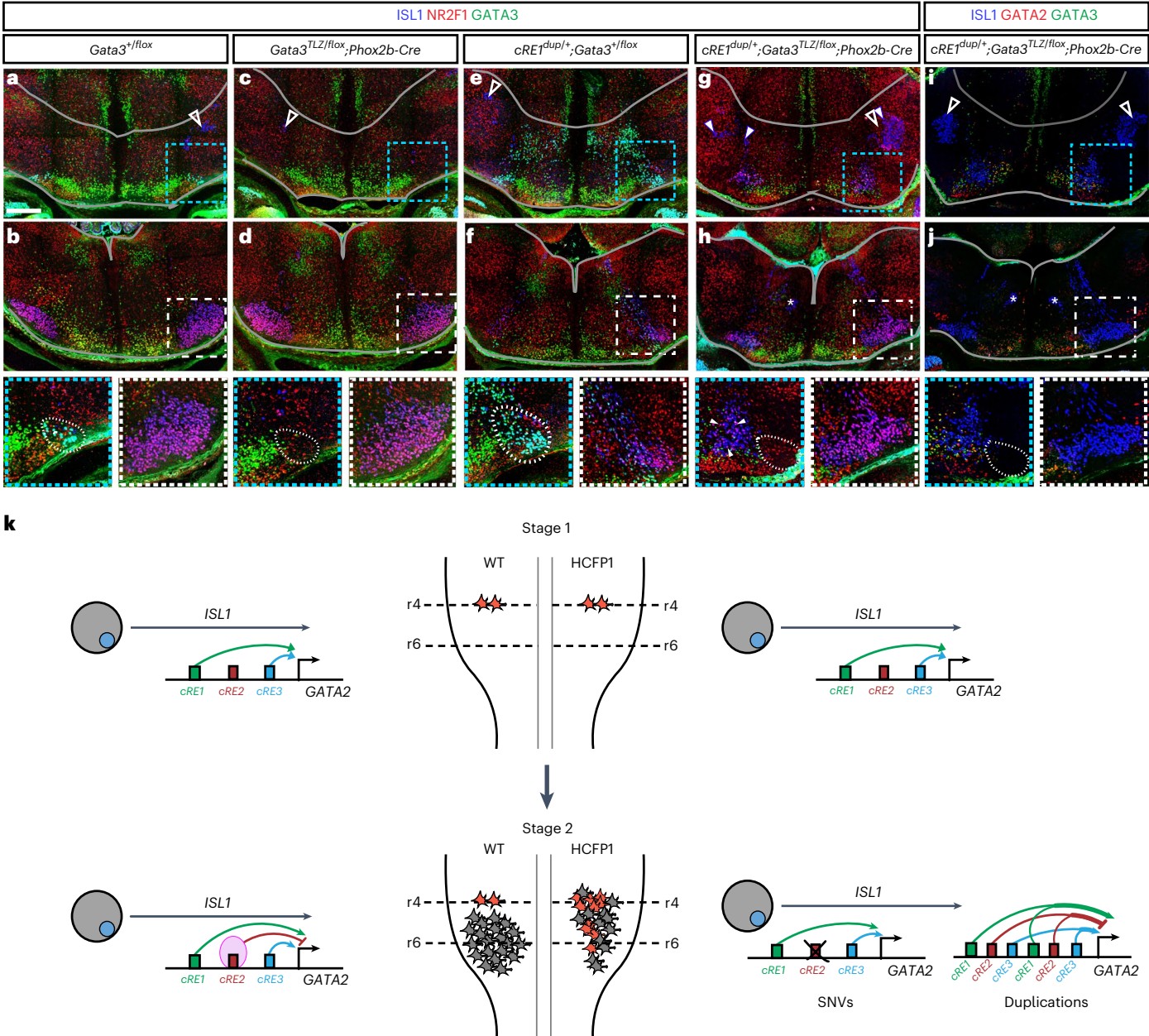

**Fig. 8 | Combining *cRE1^dup* with *Gata3* conditional inactivation partially rescues the HCFP1 phenotypes. a–h,** ISL1 (blue), NR2F1 (red) and GATA3 (green) immunofluorescent staining of hindbrain cross-sections at r4 (top row) and r6 (middle row) axial levels in E14.5 *Gata3^flox/+*;Phox2b-Cre⁻ WT (**a,b**), *Gata3^tlz/flox*; Phox2b-Cre⁺ conditional *Gata3* knockout (**c,d**), *cRE1^dup/+*;*Gata3^flox/+* duplication (**e,f**) and *cRE1^dup/+*;*Gata3^TLZ/flox*;Phox2b-Cre⁺ rescue (**g,h**) embryos. **i,j,** A rescue embryo with ISL1 (blue), GATA2 (red) and GATA3 (green) immunofluorescence (for WT and *cRE1^dup/+* comparators, see Fig. 7a,c,e,g). Dotted blue squares surround IEE OCNs in **a**, **c**, **e**, **g** and **i** and are magnified (bottom row). Dotted white squares marking the right facial nucleus are boxed in **b**, **d**, **f**, **h** and **j** and magnified (bottom row). Rescue embryos lack OCNs (**g,i**) and form an FBMN nucleus (**h,j**) intermediate in cross-sectional area between WT (**b**) and *cRE1^dup/+* (**f**) embryos. White arrows in magnification of **g** highlight r4 ISL1^ON;NR2F1^ON

FBMNs. White open arrowheads show trigeminal motor neurons and asterisks the abducens nucleus. Dorsal and ventral borders of the hindbrain are outlined in gray. Scale bar, 200 μm in **a** applies to **a–j** (*n* = 3 (**a,b**), 3 (**c,d**), 4 (**e,f**), 3 (**g,h**) and 5 (**i,j**) embryos. **k**, Model depicting the effect of HCFP1 variants. Stage 1: in both WT (left side) and HCFP1 (right side) hindbrains, early born r4MN progenitors express *Gata2*, driven in part by cRE1 and cRE3 enhancers, and assume an IEE identity (red cells). Stage 2: in WT, NR2F1 (pink oval) binds to cRE2 in later-born r4MNs, silencing *Gata2* and directing these cells to an FBMN identity (gray cells). In HCFP1, cRE2 SNVs disrupt NR2F1 binding (demarcated with X) and unimpeded cRE1 and cRE3 enhancers drive *GATA2* expression in later-born r4MNs. Duplications of cRE1, cRE2 and cRE3 generate a net increase in *GATA2* enhancer level, similarly expanding GATA2 expression. Either will increase IEEs at the expense of FBMNs, deplete the FBMN progenitor pool and result in CFP.

possible that the nonconserved NR2F1-binding site in mouse cRE1 attenuates the role of cRE2 in mouse r4MNs. Finally, introduction of cRE2 SNVs in our *lacZ* assay unveiled enhancer activity, probably through opportunistic binding of other TFs, which could vary between mice and humans[71].

We do not know the mechanism of Cluster B SNVs. In silico analysis predicted few if any TF consensus sequences in the Cluster B WT sequence. By EMSA, Cluster B SNVs did not alter NR2F1 binding and had less effect on β-galactosidase reporter expression. Loss of a nonconserved TF-binding site in Cluster B that acts in concert with

NR2F1 could result in the indistinguishable Cluster A and Cluster B SNV phenotypes. Alternatively, COUP-TFs recruit co-factors to leverage their inhibitory activity[55,72] and aberrant binding of TFs to mutant Cluster B could attenuate NR2F1 function through steric hindrance or impaired cooperative binding[73].

In summary, our results show that cell-type-specific *Gata2* expression is critical for development of r4 IEEs and its subsequent down-regulation drives a fate switch to FBMNs. This transition is tightly regulated by binding of TFs, including NR2F1, to the FBMN–IEE-specific regulatory elements cRE1, cRE2 and cRE3. HCFP1 noncoding variants alter this regulatory framework by pathologically prolonging *Gata2* expression, favoring the formation of IEEs at the expense of FBMNs.

## Online content

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

Alan P. Tenney[1,2,41], Silvio Alessandro Di Gioia [1,2,39,41], Bryn D. Webb[3,4,41], Wai-Man Chan [1,2,5], Elke de Boer [6,7], Sarah J. Garnai [1,8], Brenda J. Barry[1,2,5], Tammy Ray[1], Michael Kosicki[9], Caroline D. Robson [10], Zhongyang Zhang [4], Thomas E. Collins[1], Alon Gelber[1], Brandon M. Pratt [1], Yuko Fujiwara[11], Arushi Varshney [12], Monkol Lek [13], Peter E. Warburton[4,14], Carol Van Ryzin[15], Tanya J. Lehky[16], Christopher Zalewski[17], Kelly A. King[17], Carmen C. Brewer[17], Audrey Thurm[18], Joseph Snow[19], Flavia M. Facio[20,40], Narisu Narisu [20], Lori L. Bonnycastle[20], Amy Swift[20], Peter S. Chines[20,42], Jessica L. Bell [21], Suresh Mohan [22], Mary C. Whitman [2,21], Sandra E. Staffieri [23,24], James E. Elder[24], Joseph L. Demer [25], Alcy Torres[1,26], Elza Rachid [27], Christiane Al-Haddad[27], Rose-Mary Boustany [28], David A. Mackey[29], Angela F. Brady[30], María Fenollar-Cortés[31], Melanie Fradin[32], Tjitske Kleefstra[6,7,33], George W. Padberg[34], Salmo Raskin [35], Mario Teruo Sato[36], Stuart H. Orkin [5,11], Stephen C. J. Parker [12], Tessa A. Hadlock[22], Lisenka E. L. M. Vissers [6,7], Hans van Bokhoven [6,7], Ethylin Wang Jabs [4,37,38], Francis S. Collins[20], Len A. Pennacchio [9], Irini Manoli[15] & Elizabeth C. Engle [1,2,5,21] ✉

[1]Department of Neurology, Boston Children's Hospital, Harvard Medical School, Boston, MA, USA. [2]F.M. Kirby Neurobiology Center, Boston Children's Hospital, Boston, MA, USA. [3]Department of Pediatrics, University of Wisconsin School of Medicine and Public Health, Madison, WI, USA. [4]Department of Genetics and Genomic Sciences, Icahn School of Medicine at Mount Sinai, New York, NY, USA. [5]Howard Hughes Medical Institute, Chevy Chase, MD, USA. [6]Department of Human Genetics, Radboud University Medical Center, Nijmegen, the Netherlands. [7]Donders Institute for Brain, Cognition and Behaviour, Radboud University Medical Center, Nijmegen, the Netherlands. [8]Harvard-MIT Health Sciences and Technology, Harvard Medical School, Boston, MA, USA. [9]Environmental Genomics & System Biology Division, Lawrence Berkeley National Laboratory, Berkeley, CA, USA. [10]Department of Radiology, Boston Children's Hospital, Harvard Medical School, Boston, MA, USA. [11]Dana-Farber/Boston Children's Cancer and Blood Disorders Center, Boston, MA, USA. [12]Department of Computational Medicine and Bioinformatics, University of Michigan, Ann Arbor, MI, USA. [13]Department of Genetics, Yale University School of Medicine, New Haven, CT, USA. [14]Center for Advanced Genomics Technology, Icahn School of Medicine at Mount Sinai, New York, NY, USA. [15]Metabolic Medicine Branch, National Human Genome Research Institute, NIH, Bethesda, MD, USA. [16]EMG Section, National Institute of Neurological Disorders and Stroke, NIH, Bethesda, MD, USA. [17]Audiology Unit, Otolaryngology Branch, National Institute on Deafness and Other Communication Disorders, NIH, Bethesda, MD, USA. [18]Neurodevelopmental and Behavioral Phenotyping Service, National Institute of Mental Health, NIH, Bethesda, MD, USA. [19]Office of the Clinical Director, National Institute of Mental Health, NIH, Bethesda, MD, USA. [20]Center for Precision Health Research, National Human Genome Research Institute, NIH, Bethesda, MD, USA. [21]Department of Ophthalmology, Boston Children's Hospital, Harvard Medical School, Boston, MA, USA. [22]Department of Otolaryngology, Massachusetts Eye and Ear Infirmary, Harvard Medical School, Boston, MA, USA. [23]Centre for Eye Research Australia, Royal Victorian Eye and Ear Hospital, and University of Melbourne, Melbourne, Victoria, Australia. [24]Department of Ophthalmology, Royal Children's Hospital, Parkville, Victoria, Australia. [25]Stein Eye Institute and Departments of Ophthalmology, Neurology, and Bioengineering, University of California, Los Angeles, Los Angeles, CA, USA. [26]Department of Pediatrics, Boston Medical Center, Boston University Aram V. Chobanian & Edward Avedisian School of Medicine, Boston, MA, USA. [27]Department of Ophthalmology, American University of Beirut Medical Center, Beirut, Lebanon. [28]Pediatrics & Adolescent Medicine/Biochemistry & Molecular Genetics, American University of Beirut Medical Center, Beirut, Lebanon. [29]Lions Eye Institute, University of Western Australia, Perth, Australia. [30]North West Thames Regional Genetics Service, Northwick Park Hospital, Harrow, UK. [31]Unidad de Genética Clínica, Instituto de Medicina del Laboratorio. IdISSC, Hospital Clínico San Carlos, Madrid, Spain. [32]Service de Génétique Clinique, CHU Rennes, Centre Labellisé Anomalies du Développement, Rennes, France. [33]Center of Excellence for Neuropsychiatry, Vincent van Gogh Institute for Psychiatry, Venray, the Netherlands. [34]Department of Neurology, Radboud University Medical Center, Nijmegen, the Netherlands. [35]Centro de Aconselhamento e Laboratório Genetika, Curitiba, Paraná, Brazil. [36]Department of Ophthalmology & Otorhinolaryngology, Federal University of Paraná, Curitiba, Paraná, Brazil. [37]Department of Cell, Developmental, and Regenerative Biology, Icahn School of Medicine at Mount Sinai, New York, NY, USA. [38]Department of Pediatrics, Icahn School of Medicine at Mount Sinai, New York, NY, USA. [39]Present address: Regeneron Pharmaceuticals, Tarrytown, NY, USA. [40]Present address: Invitae Corporation, San Francisco, CA, USA. [41]These authors contributed equally: Alan P. Tenney, Silvio Alessandro Di Gioia, Bryn D. Webb [42]Deceased: Peter S. Chines ✉e-mail: elizabeth.engle@childrens.harvard.edu

## Methods

Additional methods information can be found in Supplementary Information. Data were excluded from the study only if rendered uninterpretable for technical reasons, including damage to cryosections that precluded quantification. In these instances, a replicate sample was processed and included in the study. For scRNA-seq, one E9.5 dataset was excluded from the study due to high free RNA content and the experiment was repeated to generate a usable dataset.

### Research participants

For the US-based cohort 1, research participants were enrolled under protocols approved by the Institutional Review Boards of Boston Children's Hospital, Boston (ClinicalTrials.gov identifier: NCT03059420); Icahn School of Medicine at Mount Sinai, New York; National Human Genome Research Institute, National Institutes of Health (NIH), Bethesda (ClinicalTrials.gov identifier: NCT02055248); American University of Beirut Medical Center, Beirut, Lebanon; and Royal Victorian Eye and Ear Hospital, Victoria, Australia. For the European-based cohort 2, research participants were enrolled under a protocol approved by the Institutional Review Board of CMO Radboudumc and METC East Nijmegen, the Netherlands.

Adult participants and guardians of children provided written informed consent for participation. No participant compensation was provided. The NIH paid travel and visit expenses for participation in the NIH Clinical Center evaluation. Photographs were selected from participants who consented to publication of identifying two-dimensional face photographs. Sex, number and age of participants are provided in Supplementary Table 1. Phenotypes of the affected members were obtained through a visit to the NIH Clinical Center or through examinations conducted by co-authors. A blood and/or saliva sample was collected from each participant for extraction of genomic DNA.

### Clinical evaluation

Multidisciplinary phenotyping studies were performed prospectively during a 1-week visit to the NIH Clinical Center for the 12 participants indicated in Supplementary Table 1. Studies included standardized examinations by clinical genetics, ophthalmology, audiology, dental/craniofacial, rehabilitation medicine, speech therapy, neurology, cardiology, neurocognitive and behavioral testing, as well as brain imaging, neurophysiology and laboratory studies, per protocol NCT02055248. Additional details are provided in Supplementary Methods.

### Genome build

Human genomic coordinates are GRCh37/hg19 and mouse genomic coordinates GRCm38/mm10.

### SNP generation, linkage and CNV analysis

Infinium Omni2.5Exome-8 arrays (Illumina) were used to generate whole-genome SNP data from participating members of Fam9 and a subset of the participating members of Fam1 (II-1, II-2, III-1, III-3, III-5, IV-1, IV-2, IV-5 and IV-6). Infinium Omin2.5-8 arrays were used for the remaining individuals from Fam1 (III-2, III-4, IV-3 and IV-4). Omni2.5Exome-8 SNP data were generated at the National Human Genome Research Institute Genomics Core (NHGRI/DIR), and Omni2.5 data were generated at the HMS Ocular Genomics Institute (OGI, Massachusetts Eye and Ear Institute). SNPs were processed for linkage analysis using LINKDATAGEN[76] (2016 release) and multipoint genome-wide parametric linkage analysis was performed using MERLIN v.0.5.4 (ref. 77) assuming an autosomal dominant model with full penetrance. For CNV analysis, informative SNPs were passed through PennCNV v.1.05 (ref. 78) and QuantiSNP v.2.3 (ref. 79) was used to generate CNV calls, and the resulting CNVs filtered based on specific criteria. Additional details are provided in Supplementary Methods.

### Exome sequencing

DNA libraries were prepared using Nimblegen SeqCap EZ Exome v.2 (Roche) or SureSelect Human All Exon v.4 kit (Agilent) and sequenced on either Illumina Hiseq 2000 or Illumina Hiseq 2500. All samples had at least 98% of exonic regions with at least 10× coverage. Additional details are provided in Supplementary Methods.

### Whole-genome sequencing

WGS was performed and interpreted independently for the two cohorts. Additional details are provided in Supplementary Methods.

### Targeted sequencing and variant validation and haplotypes

The cRE2-conserved noncoding region on chromosome 3 was amplified with KAPA2G Fast ReadyMix (KAPA Biosystems) and Sanger sequenced bidirectionally (Genewiz). SNV confirmation and segregation were evaluated in all available family members by Sanger sequencing. Alignment of the electropherograms was performed using Geneious Prime v.2021.1.1 (Dotmatics). Screening by ddPCR was performed for CNV screening in the conserved chromosome 3 region and *DNAJB8*. The *hTERT* (catalog no. 4403316) or *RNaseP* probes (Thermo Fisher Scientific, catalog no. 4403326) served as an internal copy number control. CNVs were confirmed using breakpoint spanning PCR when possible. All primers and probes are listed in Supplementary Table 4. Additional details are provided in Supplementary Methods.

### Electrophoretic mobility shift assays

For the EMSA experiment, 5'-IRDye 700-labeled high-performance liquid chromatography-purified probes from IDT were incubated with HeLaScribe nuclear extract, Gel shift assay grade (Promega, catalog no. E352A) or HEK293T cell nuclear extract (American Type Culture Collection, catalog no. CRL-3216). For the supershift assay, 1 µg of anti-NR2F1 antibody (D4H2 rabbit monoclonal antibody, Cell Signaling, catalog no. 6364; mouse monoclonal antibody, Perseus Proteomics, catalog no. PP-H8124-00) and respective isotype controls (WNT3A rabbit monoclonal antibody, Cell Signaling, catalog no. 2721; anti-hemagglutinin, immunoglobulin G2a mouse monoclonal antibody, Thermo Fisher Scientific, catalog no. 5B1D10) were added during the preincubation step. Gels were visualized using an Odyssey imaging system (LI-COR Biosciences). Additional details are provided in Supplementary Methods.

### Mouse husbandry

Animal husbandry was according to NIH guidelines and approved by the Institutional Animal Care and Use Committees of Boston Children's Hospital (protocol no. 00001852), the Icahn School of Medicine at Mount Sinai (protocol no. 2015-0052) and the Lawrence Berkeley National Laboratory (protocol nos. 290003 and 290008). Breeding pairs were separated after the detection of a vaginal plug at 9am, which was considered to be E0.5. The sex of the experimental embryos was not determined.

### Experimental mouse lines

Generation and acquisition of transgenic mouse lines, breeding strategies for experimental crosses and species, strain, sex, number and age of experimental animals are described in Supplementary Methods.

### *LacZ* assay

Transgenic E11.5 mouse embryos were generated and analyzed as described previously[80]. Additional details are provided in Supplementary Methods.

### Whisker movement assay

Mice aged 4 weeks to 5 months (20 males, 31 females) of the indicated genotypes were recorded in the .MOV format with the 'Slo-Mo' function on an iPhone v.6 (which records at ~120 frames per s) while walking on a treadmill. Each video recorded the superior view of the mouse's face

and body and was at least 2 min in length at the decreased frame rate. After a training session to standardize interpretation, four independent reviewers blinded to mouse genotype reviewed the unedited videos using Apple QuickTime Player (v.10.5) and scored left-side and right-side whisker movement on a scale of 0–3: '3' indicated the full trajectory of all whiskers as observed in WT mice, '2' indicated a slight reduction in range of motion or in number of whiskers moving, '1' indicated a dramatic reduction in range of motion or in number of whiskers moving and '0' indicated no detected whisker movement. Statistical analysis was performed using unpaired, two-sided Wilcoxon's testing. For presentation as a supplementary video, recordings were cropped, enlarged and edited for length in iMovie 10.3.5 (Apple, Inc.) for representative examples of treadmill walking 8–12 s in duration. Videos were 'cropped to fit' in iMovie to enlarge and focus on the head. Video segments were compiled into a single video file, with annotations generated in Microsoft 365 PowerPoint and imported as separate slides with iMovie.

### Dissection and dissociation of embryonic r4 motor neurons
ISL1$^{MN}$-GFP$^+$ and surrounding GFP$^-$ tissues were microdissected from E9.5, E10.5, E11.5 and E12.5 WT, and *cRE1$^{dup/+}$* hindbrains. To capture the anatomical extent of lateral IEE and caudal FBMN migration, the developing hindbrain from the caudal edge, trigeminal motor nucleus through the rostral third of the glossopharyngeal/vagus nuclei was collected. Single-cell suspensions were generated from dissected hindbrain tissue with enzymatic digestion and trituration (Papain Dissociation System, catalog no. LK003150) (ref. 81).

### FACS
GFP$^+$ cranial motor neurons were collected from single-cell suspensions of dissociated embryonic hindbrains using a BD FACSARIA II Cell Sorter equipped with BD FACSDiva 8.0.2 software and a 100-µm nozzle. Isl1$^{MN}$-GFP r4MNs were selected based on GFP reporter expression and found to comprise 2–6% of the total cellular input. Immediately before completion of Isl1$^{MN}$GFP$^+$ cell sorting, GFP gates were lifted to sample a representative spike of GFP$^-$ cells from the surrounding tissues and to reach an optimal number of total cells for the 10× protocol. These cells were collected into a single well of a 96-well plate containing 5 µl of 0.4% bovine serum albumin (BSA) in Hibernate E Low Fluorescence medium (HE-Lf, Brainbits).

### Single-cell CUT&Tag and data analysis
Single-cell CUT&Tag experiments were performed using the protocol single-cell CUT&Tag on 10× Genomics platform from www.protocol.io (https://www.protocols.io/view/single-cell-cut-and-tag-on-10x-genomics-platform-bqbnmsme) with the modification of using the CUTANA pAG-Tn5 enzyme (Epicypher, 15-1117) and all buffers (antibody, digitonin, digitonin-300 and tagmentation) contain 2% of BSA. Raw single-cell CUT&Tag data were processed using Cell Ranger-ATAC 2.0.0 (10× Genomics). Data analysis was performed using Signac v.1.5.0ca (ref. 82) and Seurat v.4.2.0 (ref. 83) packages. Additional details are provided in Supplementary Methods.

### ScRNA-seq
ScRNA-seq was performed using the Single Cell 3′ Reagent kits v.3.1 User Guide (10× Genomics). The resulting libraries were sequenced on a NextSeq500 platform (Illumina). Additional details are provided in Supplementary Methods.

### ScRNA-seq analysis
The raw scRNA-seq data were processed using the Cell Ranger v.7.1 analysis toolkit (10× Genomics). Data analysis was performed using R v.4.2.1, and Seurat v.4.2.0. Differential gene expression analysis was performed with the BBrowser Single Cell Browser v.3.5.26 and the BioVinci data visualization package v.3.0.0 (BioTuring)[84]. Additional details are provided in Supplementary Methods.

### Immunohistochemistry and in situ hybridization
Timed litters from crosses of WT female C57/Bl6 mice to *cRE1$^{dup/+}$* males were collected at E10.5, E12.5, E14.5 and E16.5, cryosectioned and processed for immunofluorescent staining as described previously[38], using combinations of primary antibodies against ISL1, GATA2, GATA3 and ISL1, NR2F1 and GATA3. Similar E10.5, E12.5 and E14.5 litters, as well as testes from WT and *cRE1$^{dup/+}$* adult males, were collected, cryosectioned and processed for in situ hybridization as described previously[85] using riboprobes for *Isl1* and *Gata2*. Whole-mount E11.5 embryos were collected from WT crosses and processed for in situ hybridization as described previously[86] using the *Isl1* and *Gata2* riboprobes. Additional details are provided in Supplementary Methods.

### Histological examination of r4MN identity, migration and birthdate
For examination of r4MN migration, cell identity and birthdate, WT female C57/Bl6 mice were crossed to *cRE1$^{dup/+}$* males and received single 50 mg kg$^{-1}$ of intraperitoneal injections of EdU (Thermo Fisher Scientific, catalog no. A10044) at E9.25, E10 or E10.5 development timepoints. E14.5 embryos were dissected, fixed, cryosectioned, collected on to glass slides, immunostained with guinea-pig anti-ISL1 and rabbit anti-GATA2 primary antibodies, incubated with Alexa Fluor-488 anti-guinea-pig and Alexa Fluor-647 anti-rabbit secondary antibodies, processed for EdU detection using azide-conjugated Alexa Fluor-555 and coverslipped. The methods used are as described previously[38]. Sections were imaged on a Zeiss LSM 980 confocal microscope with a ×20 objective and a 3-µm step size. For each embryo, bilateral ISL1$^{ON}$ r4MNs were analyzed caudally to rostrally, beginning at the first section rostral to the hypoglossal nucleus and ending at the first section in which IEEs were no longer present (at the level of the trigeminal motor nucleus). Cells from every fourth cryosection were counted semi-automatically in three dimensions using arivis Vision4D ×64 analysis operations. Additional details are provided in Supplementary Methods.

### Cell count statistical analysis
Statistical analysis and all plotting were performed using Rstudio build 554 and R v.4.2.1 with tidyverse package v.1.3.1. Statistics was calculated using unpaired, two-sided Student's $t$-test using the function Stat_compare_means from the ggpubr 0.4.0 package.

### Birthdating statistical analysis
The average unilateral number of r4MNs labeled by single EdU injections at E8.5, E9.25, E10.0 and E10.5 was determined as above and in Supplementary Methods. The proportions of EdU-labeled IEEs and FBMNs were calculated by dividing the number of cells labeled from each population by the total number of EdU-labeled r4MNs detected for each embryo and averaging these percentages. Statistical significance was defined by $P < 0.05$ from an unpaired, two-sided Student's $t$-test, calculated and plotted using R v.4.2.1.

### Reporting summary
Further information on research design is available in the Nature Portfolio Reporting Summary linked to this article.

## Data availability
Publicly available ChIP-seq datasets used in the present study: accession nos. GSM1817193 and GSM714811 for NR2F1; GSM714812 for NR2F2; GSM935589 for GATA2; and GSM1010738 and GSM1602667 for GATA3. Conserved TF-binding sites were obtained using rVista 2.0 (https://rvista.dcode.org). Additional epigenetic data were explored using the ENCODE database (https://www.encodeproject.org). GRCh37/hg19 human reference genome under Sequence Read Archive (SRA) accession no. PRJNA31257 and GRCm38/mm10 mouse reference genome under SRA accession no. PRJNA20689 were used for the alignment of human and mouse sequencing data, respectively. GnomAD and 1,000

genome frequencies were extracted from https://gnomad.broadinstitute.org and https://www.internationalgenome.org, respectively. Common structural variant data were obtained from the DGV (http://dgv.tcag.ca/dgv/app/home) and GoNL SV database (https://www.nlgenome.nl/login). Exome sequence and SNP data from a subset of participants are available through dbGaP Phs001383.v1.p1. WGS data from Cohort 1 participants are available through dbGaP Phs001247.v1.p1; Radboudumc consent does not allow for broad sharing via repositories and, thus, Cohort 2 WGS data are available on request and after a positive evaluation by a local data access committee confirming that the proposed re-use is in line with original consent obtained. ScRNA-seq and CUT&Tag sequencing data are available through the National Center for Biotechnology Information Gene Expression Omnibus SuperSeries accession no. GSE223274. LacZ images are uploaded to the Vista enhancer browser (https://enhancer.lbl.gov) and can be retrieved by their human coordinates as follows: hs2664 (cRE1) chr3:128,175,331–128,177,163; hs2665 (cRE2) chr3:128,177,164–128,179,169; hs2666 (cRE3) chr3:128,186,421–128,188,215; hs2667 (cRE1 + cRE2) chr3:128,175,331–128,179,169; and hs2668 (cRE2 + cRE3) chr3:128,177,164–128,188,215. Mice are available on request. Source data are provided with this paper.

## Code availability

The codes used for scRNA-seq and single-cell CUT&Tag data processing and analyses are available at https://zenodo.org/badge/latestdoi/637923997.

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

## Acknowledgements

We thank the following: all members of the U01HD079068 Consortium; V. McCarrell and the Moebius Syndrome Foundation, C. Andrews, K. Hao, K. Ismail, J. Lee, B. B. Biesecker, A. Zingaro, S. Dogar, L. Goodrich, B. Fritzsch and members of the Engle laboratory for their contributions and insightful discussions; the Regev laboratory and Broad Institute Klarman Cell Observatory; and the BioTuring User Support Team: E. van Beusekom, M. Kwint, Ro. van Beek, T. Mantere, K. Neveling, E. van der Looij, M. Schouten, J. van Reeuwijk, A. den Ouden, R. Derks, J. C. Galbany, C. Gilissen, the Radboudumc Technology Center Genomics and the Radboudumc Cell Culture Facility for their clinical, technical and bioinformatic support. The *Gata3^{tlz}* and *Gata3^{flox}* mice were kindly shared by F. Grosveld (Erasmus Medical Center, the Netherlands) and J. Zhu (NIH–National Institute of Allergy and Infectious Diseases), respectively, and provided to us by L. Goodrich (Harvard Medical School). The anti-ISL1 antibody was a generous gift from S. Morton and T. Jessell (Columbia University). We also acknowledge the use of: Boston Children's Hospital F.M. Kirby Neurobiology Center and IDDRC Gene Editing, Neurodevelopmental Behavioral, Molecular Genetics and Administrative Cores; Boston Children's Hospital Hematology/Oncology Flow Cytometry Research Facility; the Broad Institute Genomics Platform; Harvard Medical School OGI Core; the NIH Intramural Sequencing Center; the NHGRI Genomic Core; and the Gene Targeting and Transgenic Facility at Albert Einstein College of Medicine. The work was supported by: an NIH Gabriella Miller Kids First Pediatric Research Program (grant no. X01 HL132377 to E.C.E.); NIH (grant no. U01HD079068 to E.W.J., I.M. and E.C.E.); William Randolph Hearst Fund Grant (to S.A.D.G.); Moebius Syndrome Foundation grants (to A.P.T., B.D.W., Z.Z. and F.M.F.); Boston Children's Hospital—Broad Institute Collaborative Grant (to E.C.E.); NIH (grant no. R01HG003988 to L.A.P.); NIH Intramural Research Programs of the National Institute on Deafness and Other Communication Disorders (to C.Z., K.A.K. and C.C.B.), National Institute of Neurological Disorders and Stroke (T.J.L.), National Institute of Mental Health (J.S.); NIH Intramural projects (no. 1ZICMH002961 to A.T.) and NIH ZIA (grant no. HG200389 to F.S.C., N.N., L.B. and A.S.); Boston Children's Hospital Intellectual and Developmental Disabilities Research Center (grant no. 1U54HD090255); Solve-RD project (to E.d.b., T.K. and L.V.) which received funding from the European Union's Horizon 2020 research and innovation program under grant agreement no. 779257; HMS George Cheyne Shattuck Memorial Fund (to S.J.G.); American University of Beirut OpenMinds Fund (to R.-M.B.); and National Health and Medical Research Council CRE Translation of Genetic Eye research grant (no. GNT1116360 to S.E.S. and A.M.). This work was supported in part through the computational resources and staff expertise provided by Scientific Computing at the Icahn School of Medicine at Mount Sinai and Clinical and Translational Science Awards (grant no. UL1TR004419) from the National Center for Advancing Translational Sciences, NIH. The research of L.A.P. and M.K. was conducted at the E.O. Lawrence Berkeley National Laboratory and performed under the US Department of Energy contract (no. DE-AC02-05CH11231), University of California. The Centre for Eye Research Australia (S.E.S.) receives operational infrastructure support from the government in Victoria. E.C.E. and S.H.O. are Howard Hughes Medical Institute investigators.

## Author contributions

A.P.T., S.A.D.G., B.D.W. and E.C.E. conceptualized the experimental design. S.A.D.G., B.D.W., W.-M.C., E.D.B., S.J.G., L.E.L.M.V. and H.V.B. performed linkage analysis, haplotype analysis, WGS analysis, ddPCR and targeted sequencing. A.S. and P.S.C. generated and Z.Z. analyzed SNP-based CNV analysis. S.A.D.G., B.D.W., W.-M.C., N.N., L.L.B., A.S.

and P.S.C. analyzed exome sequences. E.D.B., Z.Z. and M.L. performed genome-based CNV analysis. A.V., N.N., L.L.B., S.C.J.P. and F.S.C. aligned and analyzed ENCODE data. S.A.D.G. interpreted epigenetic data and conducted the EMSA experiments. A.P.T. and W.-M.C. performed the single-cell CUT&Tag experiments. Y.F. and S.H.O. developed the *Gata2^flox* mouse line. A.P.T. developed the *Fam5^SNV* mouse line. B.D.W. developed the *cRE1^dup* mouse line and, with P.E.W., confirmed the sequence. B.D.W. performed initial gross phenotyping of the *cRE1^dup* mouse line. A.P.T. performed the whisking, birthdating and immunohistochemistry experiments. S.A.D.G., B.D.W., W.-M.C. and E.C.E. blindly scored whisking. A.P.T., T.R. and E.C.E. conducted the neuron counting experiments. A.P.T., W.-M.C., J.L.B., S.M., M.C.W. and E.C.E. performed *Fam5^snv/snv* mouse phenotyping. W.-M.C., A.P.T., B.M.P. and A.G. performed RNA-seq library preparations and sequencing. W.-M.C., A.P.T. and T.E.C. analyzed RNA-seq data. M.K. and L.A.P. conducted the mouse enhancer *lacZ* experiments. I.M. supervised and C.V.R. and F.M.F. coordinated the NIH phenotyping studies. I.M., T.J.L., C.Z., K.A.K., C.C.B., A.T. and J.S. conducted phenotyping studies at the NIH. B.J.B. coordinated non-NIH phenotyping studies. C.D.R. and E.C.E. interpreted MR images. B.D.W., E.D.B., S.E.S., J.E.E., J.L.D., A.T., E.R., C.A.-H., R.-M.B., D.A.M., A.F.B., M.F.-C., M.F., T.K., G.W.P., S.R., M.T.S., T.A.H., E.W.J. and E.C.E. provided biospecimens and clinical data. I.M., F.S.C., E.W.J., H.V.B., L.E.L.M.V. and E.C.E. conceived the study. E.C.E.

supervised the overall study. A.P.T., S.A.D.G. and E.C.E. wrote the initial manuscript. All authors saw, had the opportunity to comment on and approved the final manuscript.

## Competing interests

S.A.D.G. is a Regeneron Pharmaceuticals employee and stockholder.

## Additional information

**Extended data** is available for this paper at https://doi.org/10.1038/s41588-023-01424-9.

**Correspondence and requests for materials** should be addressed to Elizabeth C. Engle.

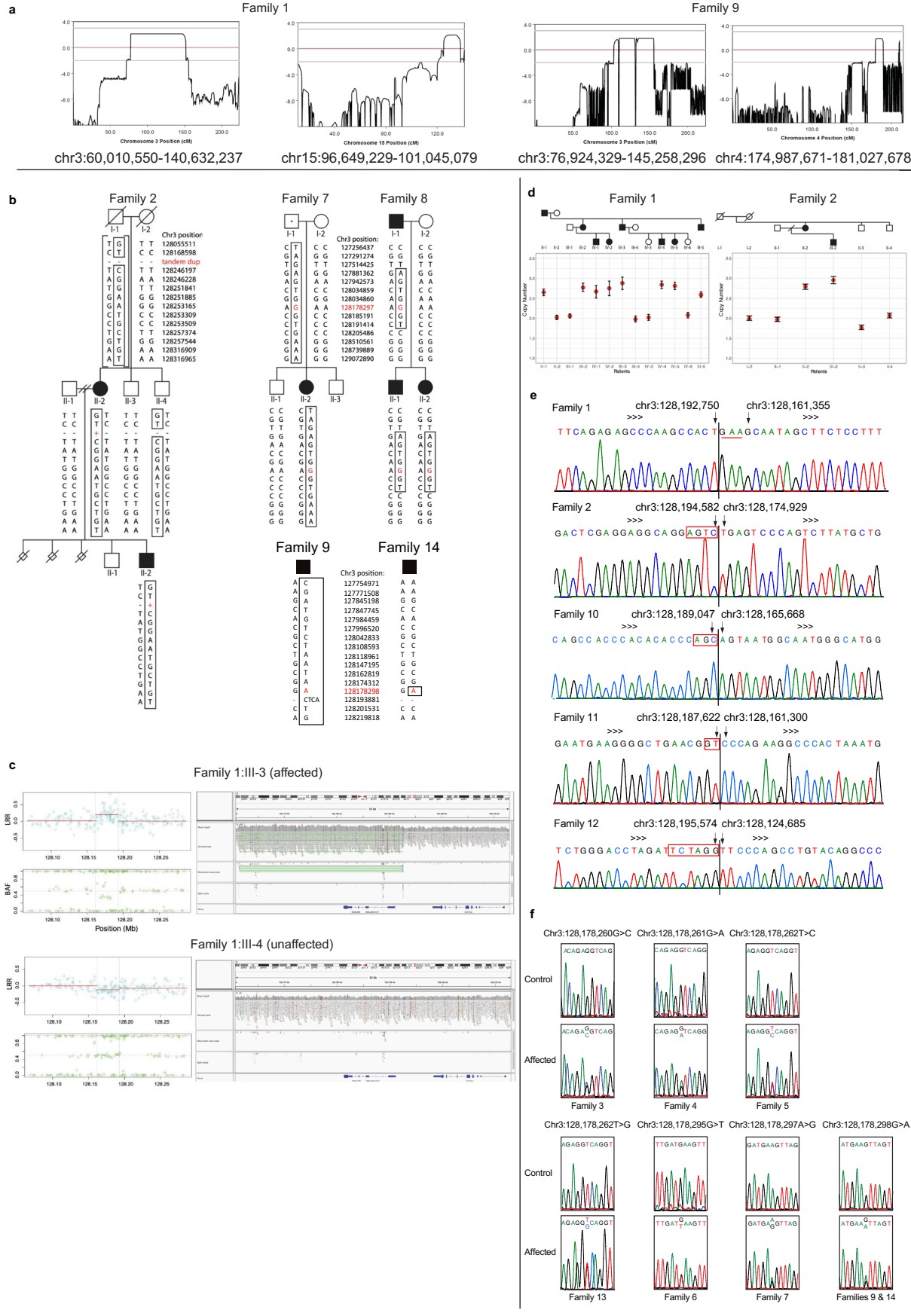

**Extended Data Fig. 1 | See next page for caption.**

**Extended Data Fig. 1 | Genetic analysis of HCFP1 pedigrees. (a)** The HCFP1 locus linkage data in Fam1 (LOD 2.1) and Fam9 (LOD 1.8). **(b)** Fam2, Fam7-9, Fam14 haplotypes. hg19 position of each SNP on chromosome 3 provided; disease-causing variants are indicated in red. Fam2 duplication arose *de novo* in II-2 on an allele inherited from I-1 (Fam2:I-1 haplotypes assumed). Fam7 and Fam8 harbored the same HCFP1 SNV and shared a ≥310 kb haploidentical region (chr3:127,881,362-chr3:128,191,414), suggesting the SNV is derived from a common ancestor. Fam9 and Fam14 harbor the same SNV on different haplotypes, suggesting independent mutational events. **(c)** SNP array data (left) and genome sequence (right) encompassing the Fam1 duplication; affected Fam1:III-3 (top), unaffected Fam1:III-4 (bottom). For SNP arrays, the Log R Ratio (LRR) is displayed in blue (top) and the B Allele Frequency (BAF) in green (bottom). Boundaries of the duplication are indicated by vertical dashed lines. LRR value reflects total copy number with the mean value, indicated by red horizontal lines, higher in duplicated than in flanking regions. BAF value is the proportion of B allele among A and B alleles at each SNP; 0, 0.5, and 1

correspond to AA, AB, and BB genotypes. The deviation from 0.5 to 0.33 or 0.66 within Fam1:III-3 corresponds to unbalanced genotypes AAB or ABB, reflecting a duplication signal. (Right) Aligned reads near the breakpoints of each of the two duplications visualized with Integrative Genomics Viewer. Location of the chromosomal region along with read depth, all read pairs, discordant read pairs, and split reads is shown. Duplicated region is highlighted in green. **(d)** Copy number quantification using digital droplet PCR for Fam1 and Fam2 duplications. Copy number values are the average of three experiments. Error bars indicate standard error. **(e)** Sanger sequencing traces that define duplication breakpoints (vertical black line) for each pedigree. Arrows preceding and following the vertical line indicate the most distal and proximal nucleotide in the duplication, respectively. Fam1 has an insertion of nucleotides GAA at the breakpoint (underlined). Fam2, Fam10-Fam12 have microhomology identified at the breakpoint (red-boxed nucleotides). **(f)** Sanger sequencing traces for each SNV with representative results of an affected and control individual.

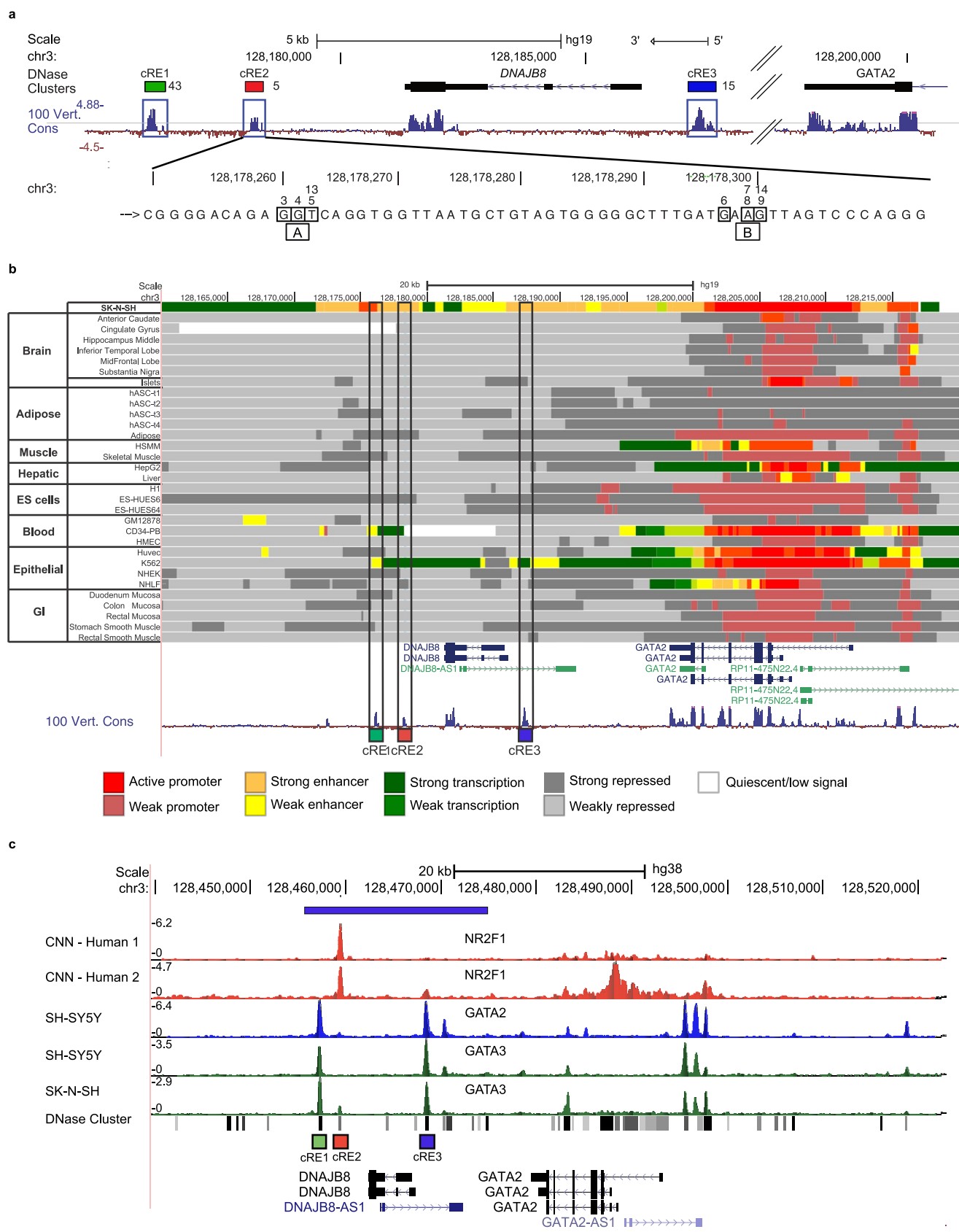

**Extended Data Fig. 2 | See next page for caption.**

**Extended Data Fig. 2 | Detailed analysis of HCFP1 region. (a)** Magnification of the UCSC Genome Browser output from Fig. 1 with multispecies conservation and the three cREs boxed in blue. Green, red, and blue boxes above cREs denote DNAse clusters reported by ENCODE with the number of unique cell lines/tissue in which the cRE has been found open. **(b)** Chromatin state segmentation of the HCFP1 region in different human tissues and cell lines from CistromeDB[30,31] and ENCODE[1] data. In SK-N-SH neuroblastoma cells (top track), there are uninterrupted stretch enhancer regions 14.8kb in length encompassing cRE3 and 3.4kb in length encompassing cRE1 and cRE2[87]. The active promoter and active transcription chromatin states indicate an overall high regulatory activity of this region in SK-N-SH cells. The largely repressive chromatin state of the corresponding region in a wide range of other tissues and cells (remaining tracks) highlights how cell-type-specific epigenomic states could potentially influence cRE activity and *GATA2* expression. *DNAJB8*, a molecular chaperone not known to be associated with human disease, is not widely transcribed. **(c)** ChIP-seq results for NR2F1 and GATA3 from published datasets[58]. Blue horizontal bar above the ChIP-seq results indicates the minimal duplication region, and the green, red, and blue squares under the ChIP-seq results indicate the positions of cRE1, cRE2, and cRE3, respectively. NR2F1 shows specific binding to cRE2 in human iPSC-derived neural crest cells. By contrast, GATA2 and its effector transcription factor (TF) GATA3 bind specifically to cRE1 and cRE3, but not to cRE2 in neuroblastoma SK-N-SH and SH-SY5Y cells.

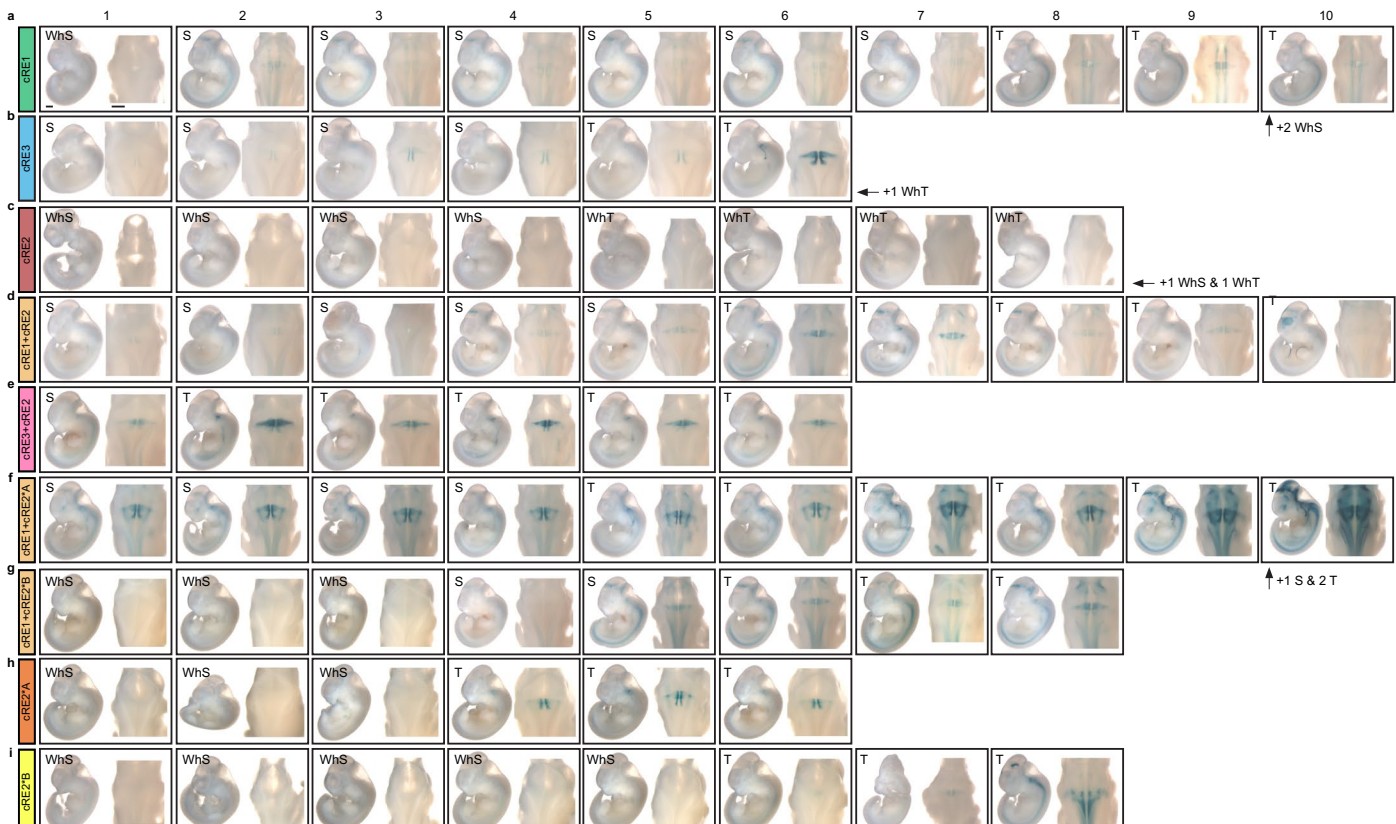

**Extended Data Fig. 3 | Summary of LacZ expression experiments. (a-i)** Replicate embryos from the *lacZ* reporter injections as indicated in schematic in Fig. 4a. (a) cRE1, (b) cRE3, (c) cRE2, (d) cRE1+cRE2, (e) cRE2+cRE3, (f) cRE1+cRE2*A, (g) cRE1+cRE2*B, (h) cRE2*A, (i) cRE2*B. Shown are all embryos up to a maximum of 10 per genotype, with text denoting embryos >10. 'S' and 'T' indicate embryos with single or tandem transgene insertion, respectively. Tandem insertions show stronger signals but less specificity than single insertions. White Single (WhS) and White Tandem (WhT) indicate embryos carrying single or tandem transgene insertion, respectively, that do not show β-galactosidase coloration. Embryos have an average crown-rump length of 6 mm. Scale bars in (a) = 500μm for the whole embryo (left) and dorsal hindbrain view through 4th ventricle (right) images and apply to (a-i) as approximate measurements.

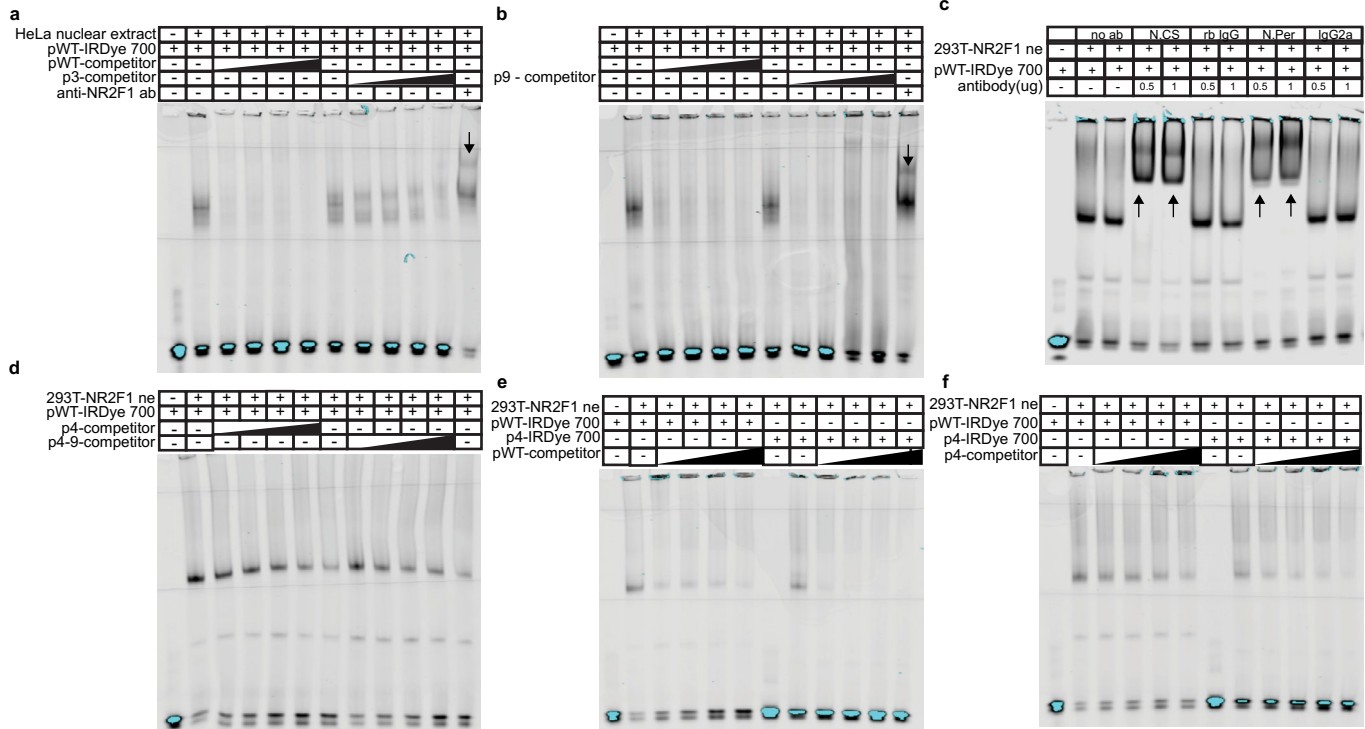

**Extended Data Fig. 4 | Additional EMSA data.** Electrophoretic mobility shift assay (EMSA) to confirm the interaction of NR2F1 with cRE2 sequence and to test whether HCFP1 SNVs attenuated this interaction *in vitro*. Blots are unmodified. Oligonucleotide probes containing the Cluster A and B region conjugated to a IRDye 700 fluorophore and competed with WT or mutant non-conjugated probes were designed. As per Fig. 4, EMSA results showing the effect of SNVs on NR2F1 binding (*293T-NR2F1 ne* denotes nuclear extract from *NR2F1* transfected 293T cells) in the presence of increasing molar excess (25x-50x-100x-200x as denoted by black slope) of WT (pWT) or mutant (pMut) competitor probe compared to hot probe (pWT-IRDye 700). (**a-b**) EMSA for Cluster A p3 (a) and Cluster B p9 (b) using HeLa nuclear extract (refer to Fig. 4j for probe maps) (n = 4). The shifted band in the second lane of each gel is abolished with the addition of small amounts of WT or Cluster B p9 competitor. The Cluster A p3 is a less efficient competitor, suggesting that the variant alters the binding of a TF to the DNA.

The addition of an anti-NR2F1 antibody causes a supershift of the TF-conjugated probe complex, indicating that this interaction is mediated by NR2F1. (**c**) A stronger shift is obtained with nuclear extract from *NR2F1*-transfected 293T cells. Specific supershift is observed using two different commercial NR2F1 antibody preparations (N.CS: Cell Signaling; N.Per: Perseus) each at two concentrations (0.5 ug and 1 ug). No supershift was observed using two isotype-specific controls (rabbit IgG for the N.CS antibody and IgG2a for the N.Per antibody), (n = 4). (**d**) No additional effects on competition are observed combining a variant in Cluster A and a variant in Cluster B on a single competitor probe (p4-p9 probe compared to p4 only (n = 6)). (**e**) Unlabeled pWT competes with labeled p4 more effectively than with labeled pWT (n = 2). (**f**) Unlabeled p4 does not compete well with labeled pWT. Comparing unlabeled pWT and unlabeled p4, the former competes better with labeled p4 (n = 2).

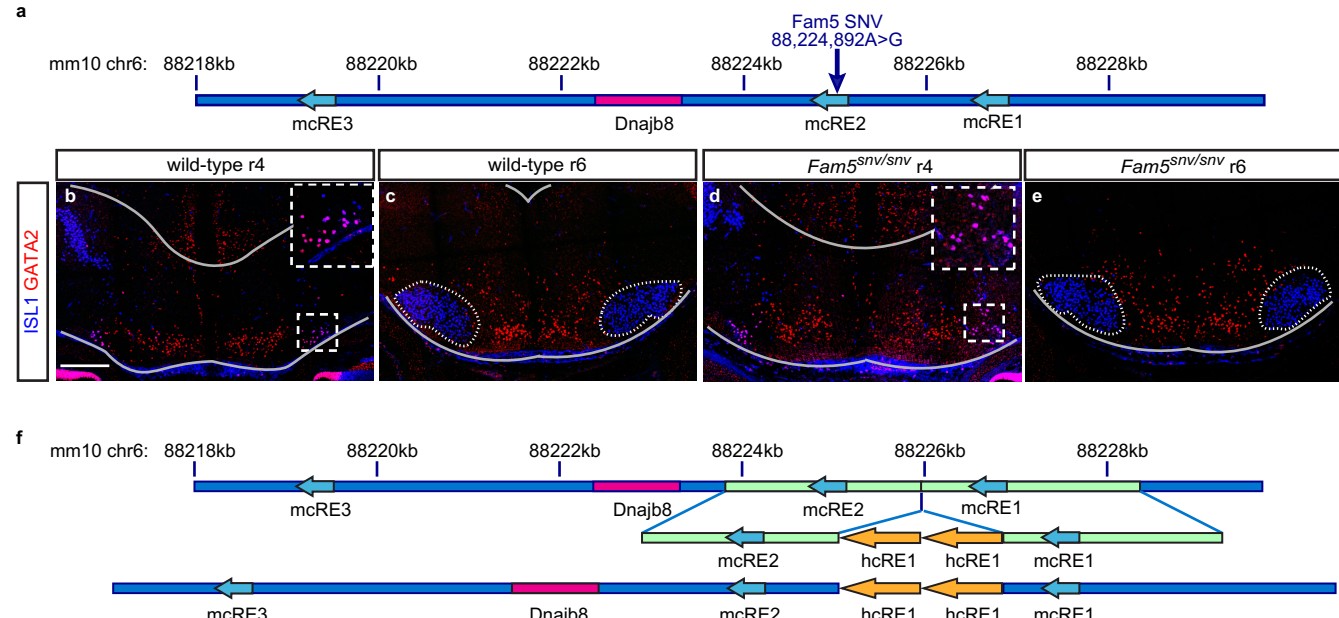

**Extended Data Fig. 5 | Facial motor nucleus formation in SNV HCFP1 mice.**
**(a)** Schematic of the orthologous *Fam5snv* variant introduced into mouse. **(b-e)**
Immunofluorescent staining for ISL1 (blue) and GATA2 (red) on cross sections
from WT (b,c) and *Fam5snv/snv* (d,e) E14.5 hindbrains at r4 (b,d) and r6 (c,e) levels.
Development of ISL1$^{ON}$;GATA2$^{ON}$ IEEs was similar in WT (dashed region in b;
inset) and *Fam5snv/snv* (dashed region in d; inset) embryos, as was the formation
of the facial motor nucleus (dotted regions in c,e). n = 4 (b,c), 4 (d,e) embryos.
Scale bar = 200µm in (b) and applies to (b-e). **(f)** Generation of a humanized
*cRE1* duplication model. Tandem copies of human *cRE1* (yellow arrows, (hg19
chr3:128,175,708-128,176,563) were inserted between the endogenous murine
*cRE1* and *cRE2* loci.

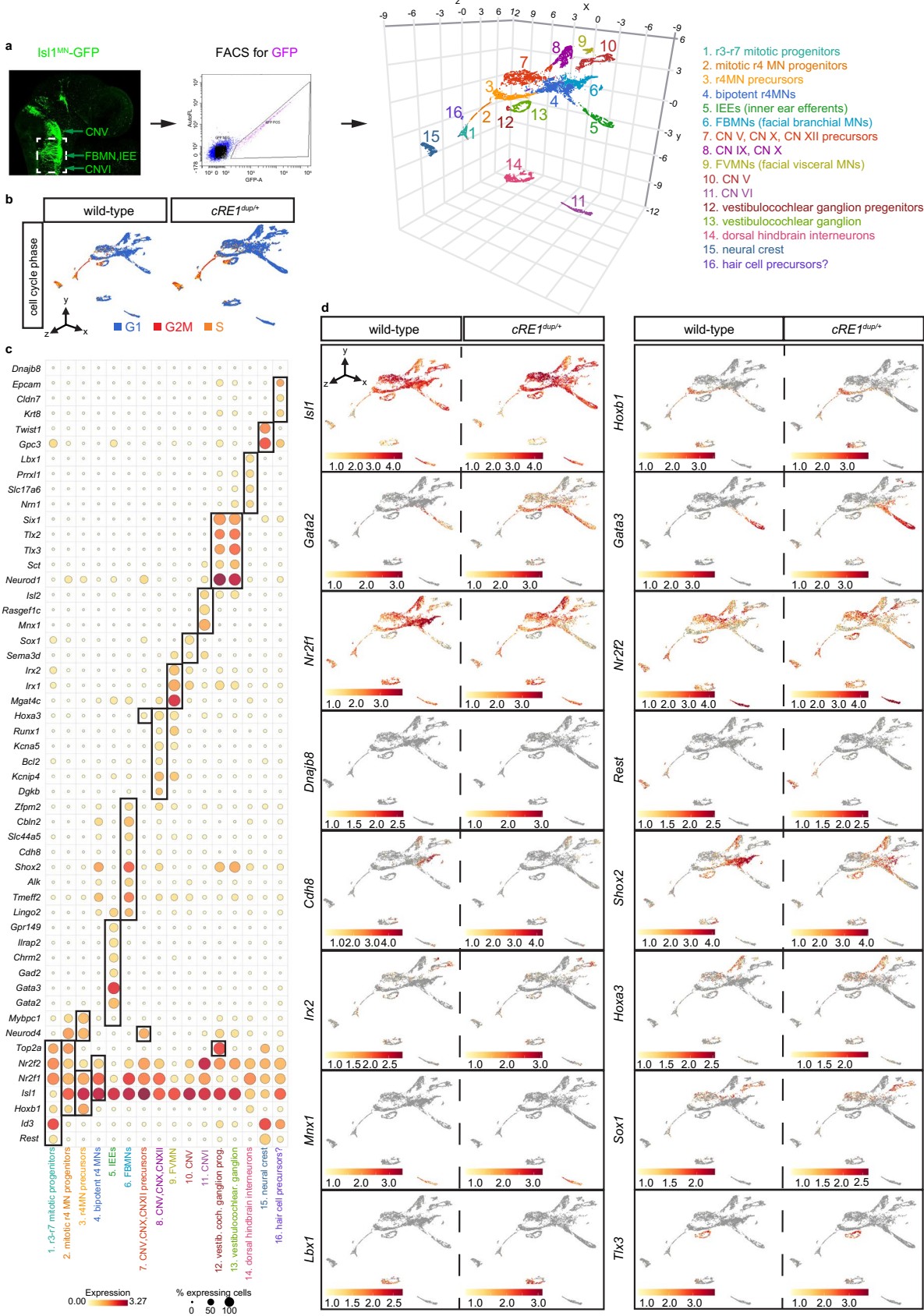

**Extended Data Fig. 6 | See next page for caption.**

**Extended Data Fig. 6 | *cRE1^dup*-mediated transcriptomic changes in the context of the developing hindbrain. (a)** scRNAseq workflow. r3-r7 GFP-positive and surrounding GFP-negative tissues were microdissected from E9.5-12.5 Isl1^MN-GFP control and *cRE1^dup/+*;Isl1^MN-GFP hindbrains, dissociated, pooled by age and genotype, and purified using FACS. In the FACS example shown, E11.5 WT Isl1^MN GFP+ r4MNs comprised 2.0% of total cellular input. Non-linear dimension reduction (clustering) was performed on a composite WT and *cRE1^dup/+* scRNAseq dataset for timespoints E9.5-E12.5. Plotted expression data was limited to *HoxB1*+ and/or *Isl1*+ cells to capture r4 ventricular zone progenitors and MNs. Proposed cluster identities are listed on the right. FACS sequential gating/sorting strategies as per Extended Data Fig. 7. **(b)** Cell cycle phase UMAP plot of E9.5-E12.5 clusters showing WT cells (left) and *cRE1^dup/+* cells (right). Cluster 1 was entirely mitotic and the likely source of r4MN progenitors. **(c)** Dot plots for marker gene expression (Y axis) in the 16 Seurat clusters (X axis). **(d)** Feature plots for select markers of r4MNs and other clusters identified in the WT (left column) and *cRE1^dup/+* (right column) *Hoxb1*+ and/or *Isl1*+ scRNAseq object. A shared FVMN (facial visceral motor neuron), CN IX, CN X trajectory is defined in part by *Hoxa3* expression (Clusters 7,8,9; see also Supplementary Table 3) and a motor CN V trajectory is marked by the expression of the previously unreported marker *Sox1* (Clusters 7,10).

**E11.5 WT limb bud dissection (GFP negative control)**

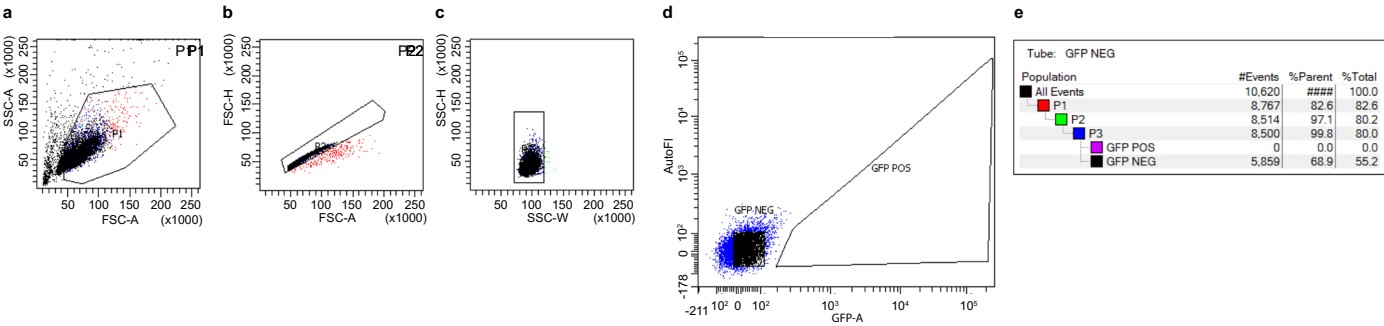

**E11.5 WT r3-r7 hindbrain dissection**

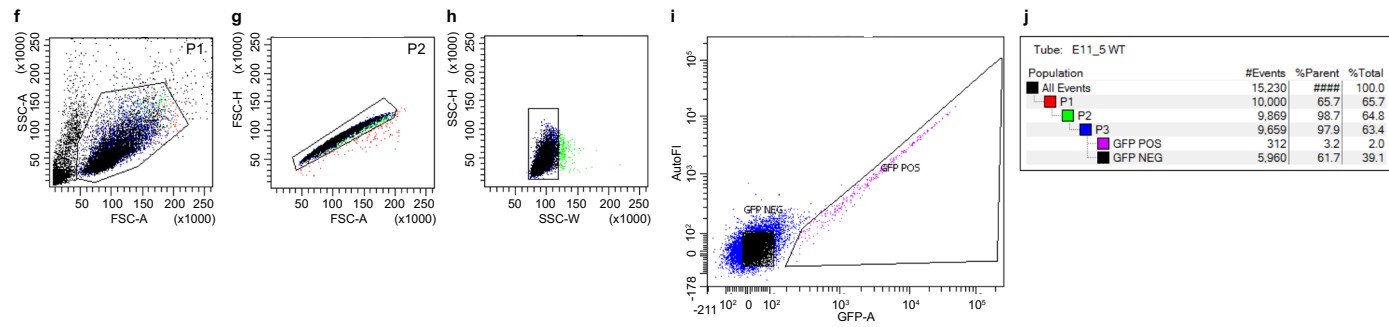

**E11.5 *cRE1^dup/+* r3-r7 hindbrain dissection**

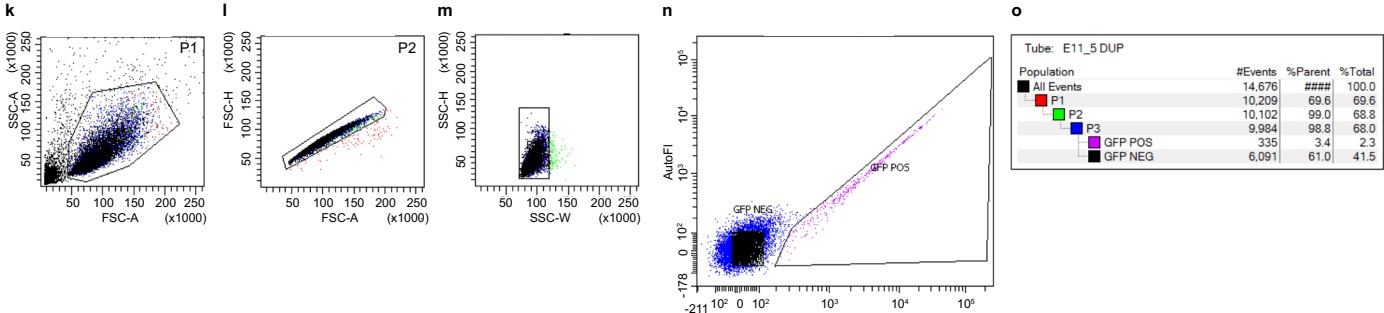

**Extended Data Fig. 7 | Representative fluorescence-activated cell sorting (FACS) gating strategy for E11.5 WT and *cRE1^dup/+* Isl1^MN^GFP⁺ r3-r7 cranial motor neurons.** Gating strategy for dissociated GFP-free limb buds collected from E11.5 WT;Isl1^MN^-GFP embryos (a-e) and r3-r7 hindbrains collected from E11.5 WT (f-j) and *cRE1^dup/+* (k-o) embryos. (**a,f,k**) P1 was drawn to include all cells and exclude debris and dead cells based on SSC-A (side scatter area) VS FSC-A (forward scatter area). (**b,g,l**) P2 was drawn for primary doublet removal using the ratio of FSC-H (forward scatter height) vs FSC-A to exclude doublets entering the point of interrogation vertically. (**c,h,m**) P3 was drawn as a secondary exclusion for horizontal doublets using the side scatter parameter of SSC-H (side scatter height) vs SSC-W (side scatter width). (**d,i,n**) GFP positive gate was drawn to include true GFP positive cells and exclude any possible autofluorescent signals from live or dead cells. GFP signal was plotted against autofluorescence (autoFl) detected as a second channel from the GFP laser and an emission filter of 575/40. (**e,j,o**) Gating summary, GFP⁺ cells comprised 0% of WT limb bud input, 2.0% of WT input hindbrain cells, and 2.3% of *cRE1^dup/+* input hindbrain cells.

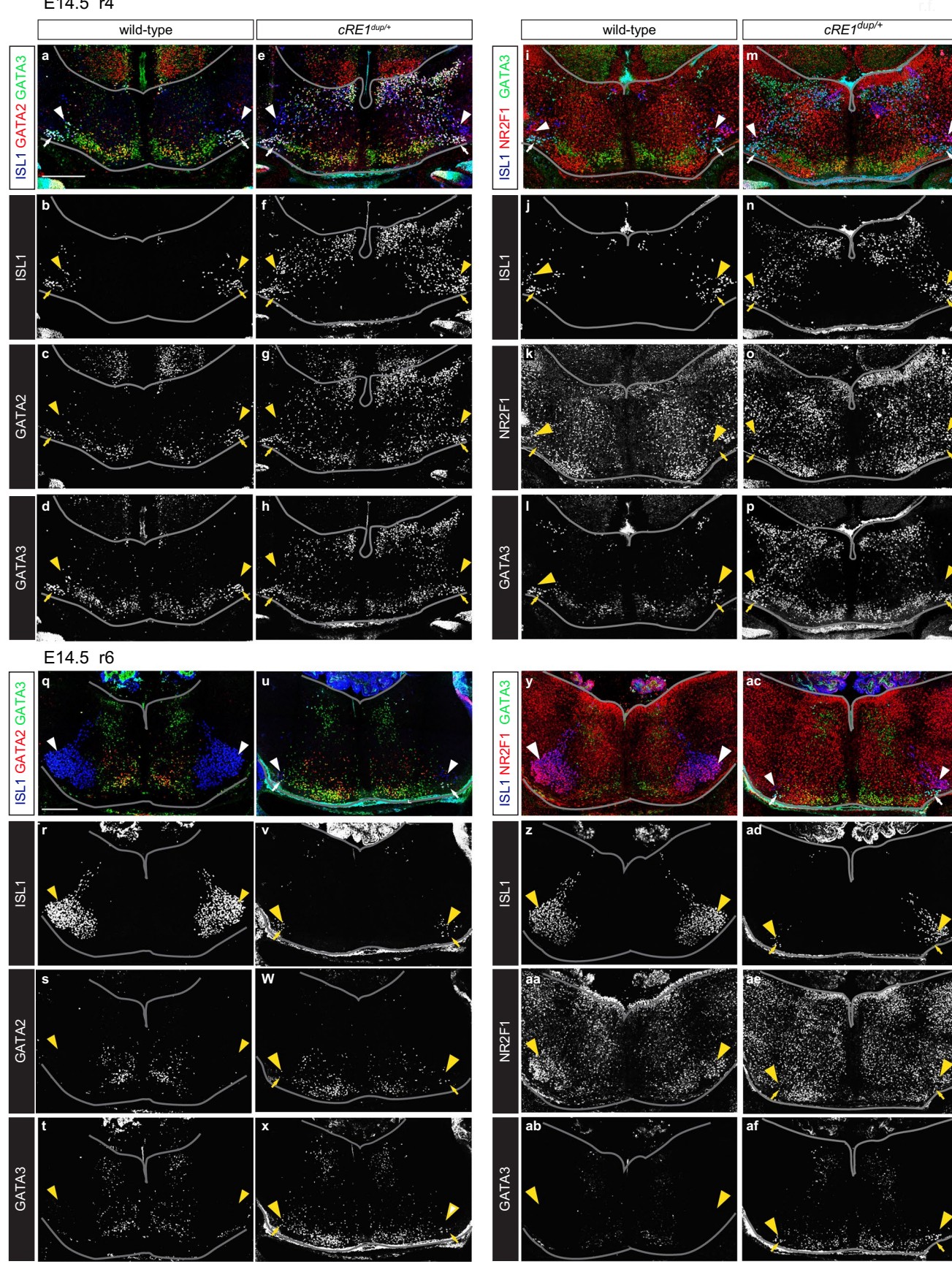

**Extended Data Fig. 8 | See next page for caption.**

**Extended Data Fig. 8 | Single channel images of Fig. 7 immunohistochemistry.** E14.5 r4 (**a-p**) and E14.5 r6 (**q-af**) Immunostaining from Fig. 7a–h presented as composite images (a,e,i,m,q,u,y,ac) with the corresponding single-channel images for each of the single antibodies shown below each composite image. Solid lines mark the approximate anatomic borders of the hindbrain, arrows indicate OCNs, and arrowheads mark FBMNs. 200μm scale bars in (a) and (q) apply to (a-p) and (q-af), respectively.

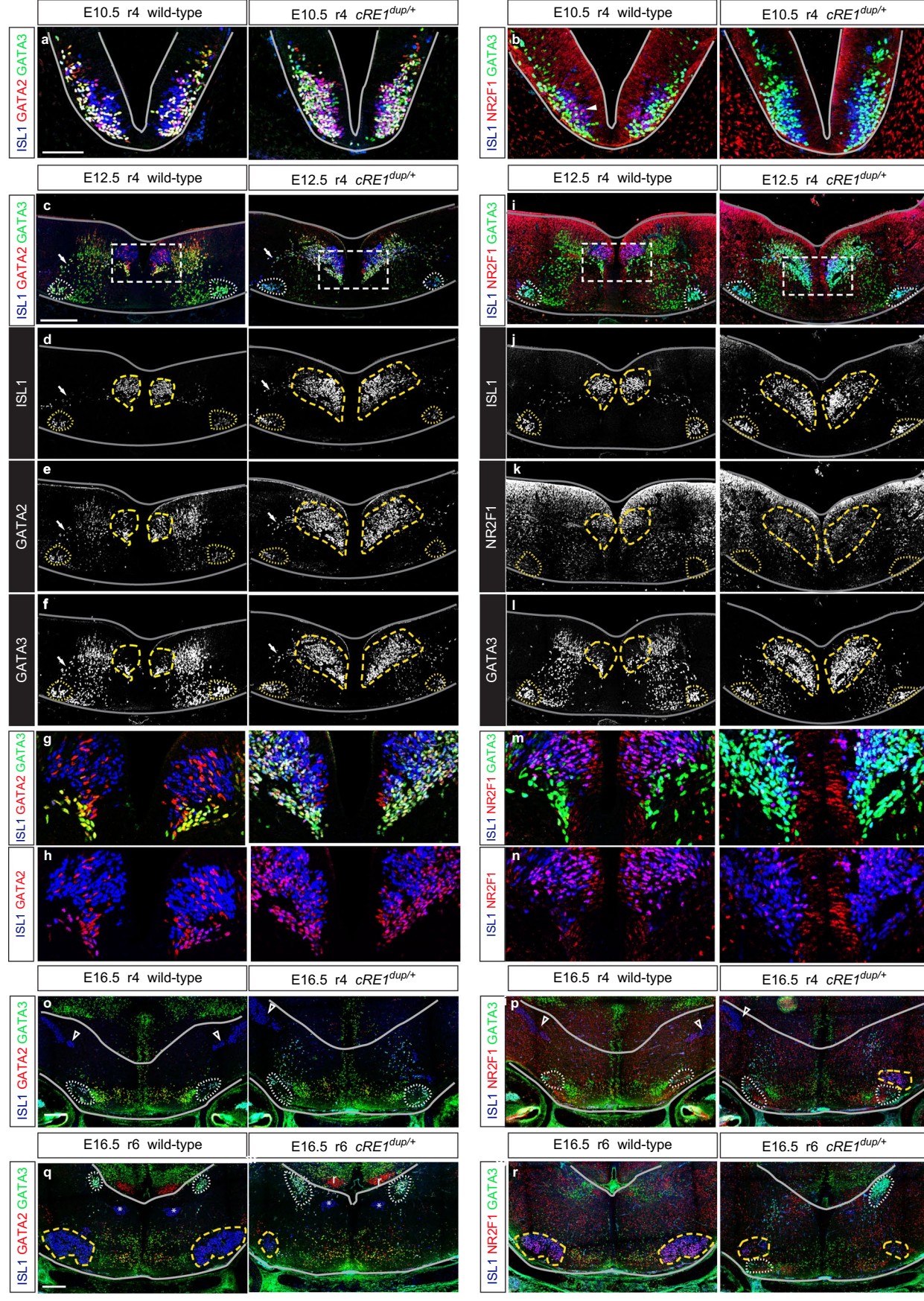

**Extended Data Fig. 9 | See next page for caption.**

**Extended Data Fig. 9 | Transcriptional and positional r4MN identity is disrupted in the c*RE1*<sup>*dup*/+</sup> embryonic hindbrain. (a,b)** Immunofluorescence of WT (left) and *cRE1*<sup>*dup*/+</sup> (right) E10.5 r4 axial hindbrain cryosections stained in (a) for ISL1 (blue), GATA2 (red), GATA3 (green), and in (b) for IS1 (blue), NR2F1 (red), GATA3 (green). In WT, a medial population of r4MNs excludes GATA2 and GATA3 expression (a, left, blue cells); in *cRE1*<sup>*dup*/+</sup>, GATA2 and GATA3 expression overlaps extensively with ISL1 (a, right). A subset of medial r4MNs express NR2F1 in WT (b, left, arrowhead) but not in *cRE1*<sup>*dup*/+</sup> hindbrains (b, right). **(c-n)** E12.5 r4 axial hindbrain cryosections stained in (c-h) as in (a), and in (i-n) as in (b) on WT (left) and *cRE1*<sup>*dup*/+</sup> (right) sections. Compound (c,i) and single (d-f,j-l) channels. Arrows mark migrating IEEs (c-f). ISL1, GATA2, GATA3 midline r4MNs are expanded in number in *cRE1*<sup>*dup*/+</sup> hindbrains. ISL1<sup>OFF</sup>;GATA2<sup>ON</sup>;GATA3<sup>ON</sup> interneurons form diffuse, bilateral columns overlapping r4MNs in WT and *cRE1*<sup>*dup*/+</sup> hindbrains. Midline r4MN clusters in (c) are enlarged in (g,h), and (i) in (m,n). Ventral IEEs are delineated from dorsal FBMNs by GATA2 and GATA3 expression in WT (g,h,

left), but not in *cRE1*<sup>*dup*/+</sup> hindbrains (g,h right). GATA2<sup>ON</sup>; ISL1<sup>OFF</sup> cells in the FBMN compartment are likely interneurons (h, left, dorsal red cells). NR2F1 expression detected in WT FBMNs (m,n, left, purple cells) is decreased in *cRE1*<sup>*dup*/+</sup> midline r4 MNs (m,n, right). **(o-r)** E16.5 r4 (o,p) and r6 (q,r) axial hindbrain cryosections with staining as per (a,b) on WT (o-r, left) and *cRE1*<sup>*dup*/+</sup> (o-r, right) sections. GATA2 is downregulated in OCNs at this stage in both genotypes (o,q), and NR2F1 is detected in FBMNs but not IEEs (p,r). *cRE1*<sup>*dup*/+</sup> but not WT embryos have scattered ectopic ISL1<sup>ON</sup>;GATA3<sup>ON</sup> r4 neurons (o,p right vs. left). The r6 VEN population is enlarged and FBMN nuclei smaller in *cRE1*<sup>*dup*/+</sup> compared to WT (q, r right vs. left). Solid lines = anatomic borders of the hindbrain, dotted ovals encircle IEEs, dashed ovals encircle FBMNs. Open arrowheads = trigeminal motor neurons, asterisks in (q) = abducens nucleus. Scale bars = 200μm. Scale bar in (a) applies to (a,b); (c) applies to (c-f, i-l); (o) and (q) apply to (o,p) and (q,r), respectively. n = 3 (a-n) and 2 (o-r) embryos.

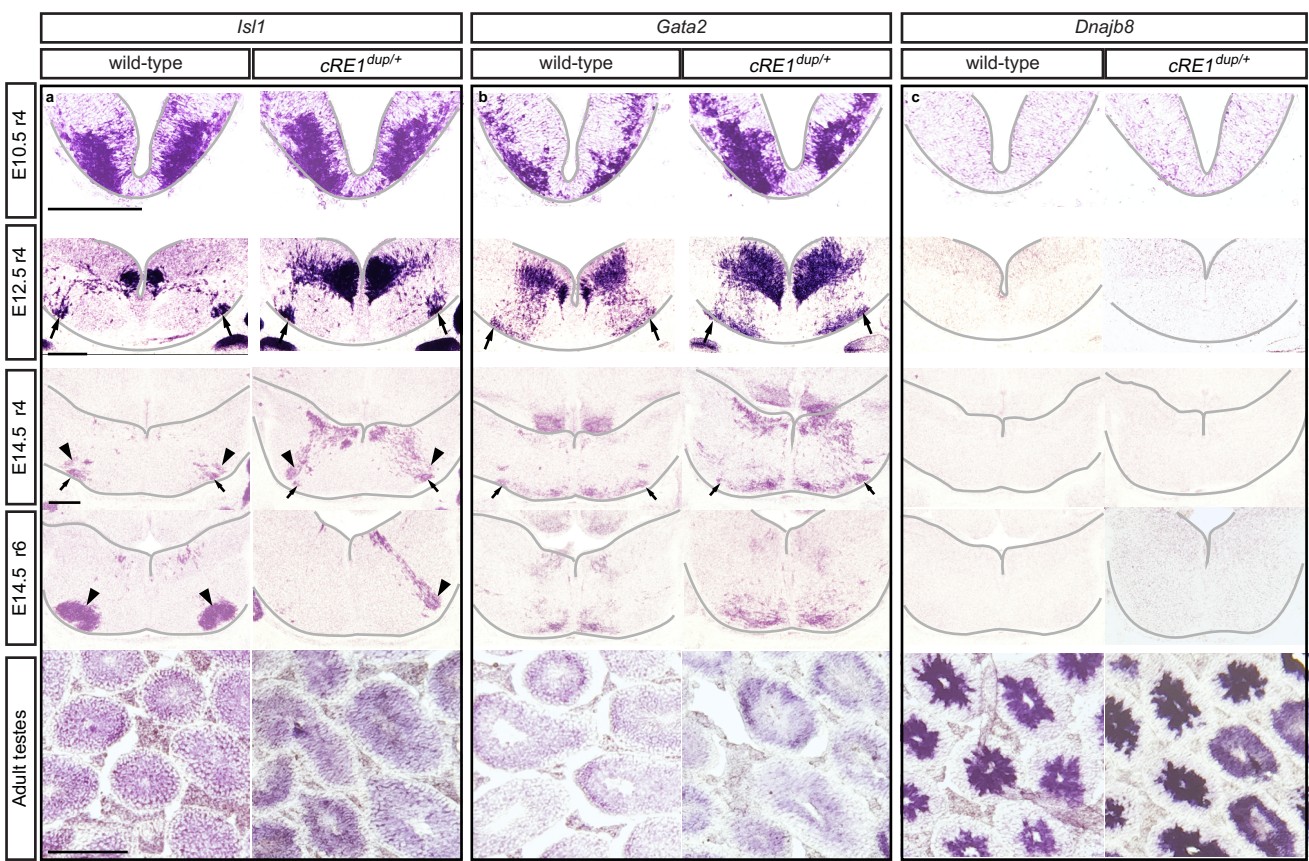

**Extended Data Fig. 10 | *Isl1, Gata2,* and *Dnajb8 in situ* hybridization. ((a-c)** *In situ* hybridization staining for *Isl1* (a), *Gata2* (b), and *Dnajb8* (c) expression in wild type (left column) and *cRE1^(dup/+)* (right column) embryos at the indicated ages and axial levels. In the *cRE1^(dup/+)* embryos, *Isl1* and *Gata2* expression was expanded in r4, and fewer *Isl1⁺* FBMNs were detected in r6 at E14.5. *Dnajb8* expression was not detected in the hindbrain region of developing r4, but the same probe did detect robust and specific expression in developing spermatids in adult WT and *cRE1^(dup/+)* testis[88]. n = 2 (E10.5), 4 (E12.5), 2 (E14.5), 3 (adult testes) samples each for WT and *cRE1^(dup/+)*. Arrows mark OCNs, arrowheads mark FBMNs. All scale bars = 200um and apply to all panels with the corresponding developmental age.

# Reporting Summary

Nature Research wishes to improve the reproducibility of the work that we publish. This form provides structure for consistency and transparency in reporting. For further information on Nature Research policies, see our Editorial Policies and the Editorial Policy Checklist.

## Statistics

For all statistical analyses, confirm that the following items are present in the figure legend, table legend, main text, or Methods section.

| n/a | Confirmed | |
|---|---|---|
| ☐ | ☒ | The exact sample size (*n*) for each experimental group/condition, given as a discrete number and unit of measurement |
| ☐ | ☒ | A statement on whether measurements were taken from distinct samples or whether the same sample was measured repeatedly |
| ☐ | ☒ | The statistical test(s) used AND whether they are one- or two-sided<br>*Only common tests should be described solely by name; describe more complex techniques in the Methods section.* |
| ☒ | ☐ | A description of all covariates tested |
| ☒ | ☐ | A description of any assumptions or corrections, such as tests of normality and adjustment for multiple comparisons |
| ☐ | ☒ | A full description of the statistical parameters including central tendency (e.g. means) or other basic estimates (e.g. regression coefficient) AND variation (e.g. standard deviation) or associated estimates of uncertainty (e.g. confidence intervals) |
| ☐ | ☒ | For null hypothesis testing, the test statistic (e.g. $F$, $t$, $r$) with confidence intervals, effect sizes, degrees of freedom and $P$ value noted<br>*Give P values as exact values whenever suitable.* |
| ☒ | ☐ | For Bayesian analysis, information on the choice of priors and Markov chain Monte Carlo settings |
| ☒ | ☐ | For hierarchical and complex designs, identification of the appropriate level for tests and full reporting of outcomes |
| ☒ | ☐ | Estimates of effect sizes (e.g. Cohen's *d*, Pearson's *r*), indicating how they were calculated |

*Our web collection on statistics for biologists contains articles on many of the points above.*

## Software and code

Policy information about availability of computer code

| Data collection | Odyssey imaging system (LI-COR Biosciences),<br>Zen Imaging Software (2012), |
|---|---|
| Data analysis | LINKDATAGEN (2016 release),<br>MERLIN (version 0.5.4),<br>PennCNV (version 1.05, 2011Jun16),<br>QuantiSNP (version 2.3),<br>LUMPY (version 0.2.13),<br>Manta (version 1.1.0),<br>CADD score (version 1.6)<br>Rstudio b554<br>Clustal Omega (omega is version)<br>Genome Analysis Toolkit (GATK 4.0 HaplotypeCaller),<br>GATK's Variant Quality Score Recalibrator (v3.5-0-g36282e4)<br>Integrative Genomics Viewer (IGV 2.8.0),<br>seqr (version 1.0),<br>gnomAD (version 2.1 and 2.1.1)<br>Picard toolkit (version 1.141)<br>1000 genomes (version 8)<br>SpliceAI (https://spliceailookup.broadinstitute.org/)<br>BWA (version 0.78),<br>Qualimap (version 2.2.1),<br>xAtlas  (version 0.1), |

VEP (version 105),
Exomizer (version 13.0.0),
ReMM (version 0.3.1)
Control-Freec (version 11.6),
ANNOVAR (version 2022Aug02),
Gencode (V.34lift37)
Canvas Copy Number Variant Caller (version 1.40.0),
Genome Studio (version 2.0)
UCSC Genome Browser (hg19, mm10),
Geneious Prime (version 2021.1.1),
ImageStudio Software (version 5.2).
Cell Ranger (version 7.1) analysis toolkit (including -ATAC v2.0.0, cell ranger mkfstq, cell ranger count from10x Genomics),
R (version 4.2.1, v4.2.11.2,)
Seurat (version 4.2.0) including  sctransform (version 0.2.0, Laboratory of Rajul Satijia, New York Genome Center),
Signac (version 1.5.0),
BBrowser Single Cell Browser (version 3.5.26, BioTuring),
BioVinci data visualization package (version 3.0.0, BioTuring),
ImageJ (NIH, version 1.53d),
arivis Vision4D x64,
tidyverse (version 1.3.1),
ggpubr (0.4.0 package),
rVista (version 2.0),
BD FACSDiva 8.0.2,
Adobe Photoshop 6 (version 13.0 x64)
Apple QuickTime Player (version 10.5)
Apple iMovie 10.3.5,
Microsoft 360 PowerPoint

For manuscripts utilizing custom algorithms or software that are central to the research but not yet described in published literature, software must be made available to editors and reviewers. We strongly encourage code deposition in a community repository (e.g. GitHub). See the Nature Research guidelines for submitting code & software for further information.

# Data

Policy information about availability of data

All manuscripts must include a data availability statement. This statement should provide the following information, where applicable:
- Accession codes, unique identifiers, or web links for publicly available datasets
- A list of figures that have associated raw data
- A description of any restrictions on data availability

Publically available ChipSEQ datasets used in this study: GSM1817193 and GSM714811 for NR2F1; GSM714812 for NR2F2; GSM935589 for GATA2; GSM1010738 and GSM1602667 for GATA3. Conserved TFs binding sites were obtained using rVista 2.0 (https://rvista.dcode.org/). Additional epigenetic data were explored using the ENCODE database (https://www.encodeproject.org/). GRCh37/hg19 human reference genome under accession number SRA PRJNA31257 and GRCm38/mm10 mouse reference genome under accession number SRA PRJNA20689 were used for the alignment of human and mouse sequencing data, respectively. Variant frequencies were extracted from gnomAD (https://gnomad.broadinstitute.org/) and 1000G (https://www.internationalgenome.org). Common structural variants data were obtained by DGV (http://dgv.tcag.ca/dgv/app/home) and from the GoNL SV database (https://www.nlgenome.nl/login). KEGG PATHWAY Database was also used. OMIM was queried for disease codes (omim.org). Exome sequence and SNP data from a subset of participants are available through dbGaP Phs001383.v1.p1. WGS data from a subset of participants are available through dbGaP Phs001247.v1.p1. Single-cell RNA and CUT&Tag sequencing data are available through NCBI GEO SuperSeries record GSE223274. LacZ images are uploaded to the Vista enhancer browser (https://enhancer.lbl.gov/) and can be retrieved by their human coordinates as follows: hs2664 (cRE1) chr3:128,175,331-128,177,163; hs2665 (cRE2) chr3:128,177,164-128,179,169; hs2666 (cRE3)chr3:128,186,421-128,188,215; hs2667 (cRE1+cRE2) chr3:128,175,331-128,179,169; hs2668 (cRE2+cRE3) chr3:128,177,164-128,188,215. Mice are available on request.

# Field-specific reporting

Please select the one below that is the best fit for your research. If you are not sure, read the appropriate sections before making your selection.

☒ Life sciences   ☐ Behavioural & social sciences   ☐ Ecological, evolutionary & environmental sciences

For a reference copy of the document with all sections, see nature.com/documents/nr-reporting-summary-flat.pdf

# Life sciences study design

All studies must disclose on these points even when the disclosure is negative.

| | |
|---|---|
| Sample size | Statistical methods were not used to determine sample size.  For human genetic study, sample size was limited by number of participants with congenital facial weakness available to us and in whom we identified rare variants. For rodent cell counts and birthdating experiments, sufficient replicates were used to show reproducible EdU bioavailability over multiple independent injections and demonstrate age-dependent trends in IEE and FBMN identities and birthdates. For rodent immunohistochemistry, sufficient replicates were performed to show the reproducibility of transcription factor expression dynamically marking developing r4 MN subpopulations over developmental time. For in vitro EMSA experiments, enough replicates were performed with the WT and mutant probes in different conditions for reproducibility. |
| Data exclusions | Data were excluded from the study only if rendered uninterpretable for technical reasons:  Damaged cryosections that precluded quantitation |

| Data exclusions | were excluded and a replicate sample was processed and included in the study; one E9.5 scRNAseq dataset was excluded due to high free RNA content and the experiment was repeated to generate a usable dataset; EMSA experiments were excluded if there were loading or gel-running technical problems. |
|---|---|
| Replication | All attempts at replication were successful. Rodent histology experiments were performed with a minimum of 3 biological replicates with the exception of immunostaining on E16.5 hindbrain cross sections which were performed twice. ample sizes are provided in the paper. For scRNAseq and scCUT&Tag, we conducted replicate experiments at each developmental time point from multiple pooled embryos. For EMSA, experiments were repeated multiple times in multiple days for reproducibility. |
| Randomization | The experiments were not randomized. Mice were allocated by genotype to wild-type or mutant categories. |
| Blinding | Investigators were blinded to genotypes of humans for video-based clinical interpretation of facial weakness and to mice for video-based interpretation of whisking and for cell counts and EdU quantification. Blinding was not necessary for qualitative analysis of histology. |

# Reporting for specific materials, systems and methods

We require information from authors about some types of materials, experimental systems and methods used in many studies. Here, indicate whether each material, system or method listed is relevant to your study. If you are not sure if a list item applies to your research, read the appropriate section before selecting a response.

| Materials & experimental systems | | Methods | |
|---|---|---|---|
| n/a | Involved in the study | n/a | Involved in the study |
| ☐ | ☒ Antibodies | ☒ | ☐ ChIP-seq |
| ☐ | ☒ Eukaryotic cell lines | ☐ | ☒ Flow cytometry |
| ☒ | ☐ Palaeontology and archaeology | ☐ | ☒ MRI-based neuroimaging |
| ☐ | ☒ Animals and other organisms | | |
| ☐ | ☒ Human research participants | | |
| ☐ | ☒ Clinical data | | |
| ☒ | ☐ Dual use research of concern | | |

## Antibodies

| Antibodies used | Primary antibodies:<br>guinea pig anti-ISL1/2 (Project ALS, RRID AB_2631974, Lot 1277),<br>rabbit anti-GATA2 (Abcam cat#A0677, Lot #0045400202),<br>rat anti-GATA3 (eBioscience cat#14-9966-80, Lot #2202643),<br>rabbit anti-NR2F1 (Millipore Sigma cat#ABE1425, Lot #3083591),<br>rabbit anti-NR2F1 (Cell Signaling Technologies D4H2 cat#6364, Lot #1)<br>mouse anti-NR2F1 (Perseus Proteomics #PP-H8124-00, lot number unavailable)<br>rabbit anti-WNT3A (Cell Signaling Technologies #2721, Lot #1)<br>mouse monoclonal anti-HA, IgG2a (Thermo Fisher Scientific #5B1D10)<br><br>Secondary antibodies (all from ThermoFisher Scientific):<br>AlexaFluor 488 anti-guinea pig (cat#A-11073, Lot#2160428),<br>AlexaFluor 568 anti-rabbit (cat#A-11011, Lot#2013083),<br>AlexaFluor 647 anti-rat (cat#A-21247, Lot#2156534),<br>Alexafluor 647 anti-rabbit (cat#A-37733, Lot#WL333239) |
|---|---|
| Validation | Commercially available antibodies have been validated with manufacturer-generated data and/or supporting publications found on manufacturer websites.<br><br>The specificity of the guinea pig anti-ISL1 antibody (Laboratory of Thomas Jessell and Project ALS, RRID RRID AB_2631974) was confirmed by the similarity of the sensory and motor nuclei marked by anti-ISL1 immunostaining (Figure 3,7,8; Extended Data Figures 8,9) and those marked by an Isl1 riboprobe in in situ hybridization on age-matched cryosections (Extended Data Figure 10a). This antibody was widely used by the laboratory of origin and others to mark developing motor neurons (Jung et. al. Cell 2018, PMID 29425489; Tanabe et. al Cell. 1998, PMID 9778248).<br><br>The specificity of the rabbit anti-GATA2 antibody (Abclonal A0677) was confirmed by the similarity of the IEE and hindbrain interneuron immunofluorescent staining (Figures 3,7,8; Extended Data Figure 8) and that of a Gata2 riboprobe for in situ hybridization ( Extended Data Figure 10b). Specificity of the antibody was also indicated by the loss of GATA2 immunostaining from the region normally occupied by IEEs in the Gata2ko/flox; Phox2bCre+ conditional knockouts and sustained expression in the hindbrain interneurons that do not express Phox2bCre (Figure 3d). Vendor-supplied images of antibody staining on paraffin-embedded brain cross-sections confirmed specificity in epitope detection.<br><br>The specificity of the rat anti-GATA3 antibody (eBioscience 14-9966-80) was confirmed by the loss of GATA3 immunostaining from the region normally occupied by IEEs in the Gata3tlz/flox; Phox2bCre+ conditional knockouts (Figure 3e). Our GATA3 immunostaining matched published Gata3 in situ hybridization on E10.5 hindbrain cryosections (Pata et. al. Development 1999, PMID 10556076) and chromogenic detection of the β-galactosidase Gata3tlz/+ knockin reporter predicted to accurately recapitulate native GATA3 expression (Karis et. al. J. Comp. Neurol. 2001, PMID 11135239). The antibody has been shown to mark tissues known to be enriched |

in GATA3, including the developing nephric duct (Sanchez-Ferras et. al., Nature Communications 2021, PMID 33976190).

The specificity of the rabbit anti-NR2F1 antibody (Millipore-Sigma ABE1425) was confirmed by published studies showing neuronal subtype-specific loss of immunofluorescent staining in NR2F1 conditional knockouts (Bovetti et. al. Development 2013, PMID 24227652; Alfano et. al. Development 2011, PMID 21965613).

For each antibody, staining of the primary and corresponding secondary antibodies was compared to that seen with secondary antibodies alone, and no confounding background staining from secondary antibodies was detected.

## Eukaryotic cell lines

Policy information about cell lines

| | |
|---|---|
| Cell line source(s) | HEK293T cells (ATCC CRL-3216 ). Note that HeLa cell nuclear extracts were purchased from a commercial source and were not cultured in the lab. |
| Authentication | HEK293T cell line was purchased from a commercial source and was not further authenticated. |
| Mycoplasma contamination | Cell line was tested repeatedly and tested negative for mycoplasma contamination. |
| Commonly misidentified lines (See ICLAC register) | None used in this study. |

## Animals and other organisms

Policy information about studies involving animals; ARRIVE guidelines recommended for reporting animal research

| | |
|---|---|
| Laboratory animals | Mice were maintained in pathogen-free environments and fed ad libitum with sterile standard diet and water in a temperature, humidity, and light-controlled rooms (Boston Children's Hospital: 22°C set-point +/- 1.3°C, RH35-70% +/- 5%, 12/12 light/dark cycle, 10-15 air changes per hour; Lawrence Berkeley National Laboratory: 19-23°C, RH30-70%, 12/12 light/dark cycle, 15-20 air changes per hour; Icahn School of Medicine at Mount Sinai: 20-23°C, RH 30-70%, 12/12 light/dark, and 10-15 air changes per hour). Animals were not involved in any previous experiments. Both male and female mice aged 1-6 months were bred to generate experimental embryonic litters. Both male and female mice aged 5 weeks-5 months were assessed in the whisking assay. Age did not correlate with genotype. Fam5snv/snv mice were maintained on a 129S1/C57BL/6J mixed background. Experimental Fam5snv/snv mice were generated by intercrossing Fam5snv/snv breeders. The Gata2KO allele was generated by intercrossing Gata2flox/flox with Ella-Cre (Jackson Labs 003724) and the resulting Gata2flox/+, Ella-Cre+ mice were back-crossed to Ella-Cre two times to ensure germline deletion. Gata2KO/+ mice were crossed to Phox2bCre+ (Jackson Labs 016223) mice to generate Gata2KO/+; Phox2bCre+ breeders that, along with Gata2flox/flox mice, were maintained on a C57/Bl6 background (Jackson Labs 000664). Gata2KO/flox;Phox2bCre+ cKO embryos and control littermates were generated by crossing Gata2KO/+;Phox2bCre+ breeders to Gata2flox/flox breeders. Gata3tlz/flox and Gata3tlz/+;Phox2bCre+ mice were maintained on their 129/C57BL6/CD1 mixed background. Gata3TLZ/flox;Phox2bCre+ cKOs and control littermates were generated by crossing Gata3tlz/+;Phox2bCre+ breeders to Gata3flox/flox breeders. cRE1dup/+ mice were maintained on their C57/Bl6/BalbC mixed background and were intercrossed with Isl1MN-GFP reporter line (Jackson Labs 017952) to generate timed litters for scRNAseq and scCUT&Tag. cRE1dup/+;Gata3tlz/flox;Phox2bCre+ rescue mice and littermate controls were generated by crossing cRE1dup/+;Gata3tlz/+;Phox2bCre+ breeders to Gata3flox/flox breeders. |
| Wild animals | The study did not involve wild animals. |
| Field-collected samples | The study did not involve field-collected samples. |
| Ethics oversight | Institutional Animal Care and Use Committees of Boston Children's Hospital (Protocol number 00001852), the Icahn School of Medicine at Mount Sinai (Protocol number 2015-0052), and the Lawrence Berkeley National Laboratory (Protocol numbers 290003 and 290008). |

Note that full information on the approval of the study protocol must also be provided in the manuscript.

## Human research participants

Policy information about studies involving human research participants

| | |
|---|---|
| Population characteristics | Male and female research participants of any age (range 6 weeks-87 years) diagnosed with isolated congenital facial weakness (some misdiagnosed with Moebius syndrome) and available family members were enrolled in the study. This was a cross-sectional observational study of subject/families enrolled over many decades through dedicated research protocols, as outlined in the Methods section. No population statistics are presented in the paper. Age, gender, genotype and phenotype information for each participant are provided in the Supplemental Clinical Data and Tables 1 and 2. |
| Recruitment | Subjects were referred to the research protocols in the different Institutions through their physicians, the family support group Moebius Syndrome Foundation (MSF), or self-referral. They carried a diagnosis of congenital facial weakness or Moebius syndrome. There was likely self-selection bias for subjects who have access to health care and/or interest to participate in research but this is not likely to impact results.  Representation of subjects/families in research was improved through the MSF patient advocacy group. No compensation was provided to participants, but all expenses for transportation, lodging, and evaluation at the NIH Clinical Center were covered, facilitating participation of families without insurance / access to medical care. |

| Ethics oversight | National Institutes of Health IRB (FWA00005897) has approved the NIH Clinical Center study. Enrollment occurred under protocols approved by the Institutional Review Boards of Boston Children's Hospital, Boston, MA; CMO Radboudumc and METC East Nijmegen, Netherlands; Icahn School of Medicine at Mount Sinai, New York, NY; National Human Genome Research Institute, NIH, Bethesda, MD; American University of Beirut Medical Center, Beirut, Lebanon; Royal Victorian Eye and Ear Hospital, VIC, Australia. |

Note that full information on the approval of the study protocol must also be provided in the manuscript.

# Clinical data

Policy information about clinical studies

All manuscripts should comply with the ICMJE guidelines for publication of clinical research and a completed CONSORT checklist must be included with all submissions.

| Clinical trial registration | NCT02055248 and NCT03059420 |
| Study protocol | See Clinicaltrials.gov: https://www.clinicaltrials.gov/ct2/show/NCT02055248 and https://clinicaltrials.gov/ct2/show/NCT03059420 |
| Data collection | Participants were enrolled between ~1990-present. Subjects of the NIH clinical protocol were recruited between 2014 and 2019 and data were collected at the NIH Clinical Center, Bethesda, MD. All other participants were enrolled between ~1990-present and data were collected in both clinical and research settings in Boston MA, New York City NY, Los Angeles CA, Nijmegen Netherlands, Harrow UK, Madrid Spain, Rennes France, Parana Brazil, Beirut Lebanon, and Victoria (Melbourne and Parkville), Australia. |
| Outcomes | Primary and secondary outcomes are provided at https://www.clinicaltrials.gov/ct2/show/NCT02055248. |

# Flow Cytometry

## Plots

Confirm that:

☒ The axis labels state the marker and fluorochrome used (e.g. CD4-FITC).

☒ The axis scales are clearly visible. Include numbers along axes only for bottom left plot of group (a 'group' is an analysis of identical markers).

☒ All plots are contour plots with outliers or pseudocolor plots.

☒ A numerical value for number of cells or percentage (with statistics) is provided.

## Methodology

| Sample preparation | ISL1MN-GFP-positive and surrounding GFP negative tissues were microdissected from the hindbrain region spanning rhombomere 3 to rhombomere 7 of E9.5, E10.5, E11.5, and E12.5 mouse hindbrains. GFP-free limb buds were collected as a negative control to set GFP gating. Single cell suspensions of hindbrain tissues were generated with enzymatic digestion and trituration (Papain Dissociation System, Worthington) and passed through a 35um mesh size cell strainer by gravity to remove cell aggregates. |
| Instrument | BD FACSAria2 |
| Software | BD FACSDiva 8.0.2 |
| Cell population abundance | Isl-GFP cells comprise 2-6% of total cells sorted. |
| Gating strategy | GFP-positive r4-r6 neurons were purified using FACS gated to forward versus side scatter (FSC vs SSC) to exclude debris, and forward scatter width versus orward scatter (FSC-W vs FSC) followed by side scatter width versus side scatter (SSC-W vs SSC) to exclude doublets and cell aggregates. Isl1MN-GFP r4-r6 MNS were then selected based on reporter expression. Immediately prior to completion of Isl1GFP cell sorting, GFP gates were lifted to sample a representative spike of GFP-negative cells from the surrounding tissues and to reach an optimal number of total cells for the 10x Chromium scRNAseq protocol. |

☒ Tick this box to confirm that a figure exemplifying the gating strategy is provided in the Supplementary Information.

# Magnetic resonance imaging

## Experimental design

| Design type | A structural brain MRI including the internal auditory canal (IAC) and posterior fossa was offered to all participants who could cooperate with the scanning procedure without sedation. |
| Design specifications | N/A |
| Behavioral performance measures | N/A |

## Acquisition

| | |
|---|---|
| Imaging type(s) | Structural |
| Field strength | 3.0T Philips MRI scanner |
| Sequence & imaging parameters | Philips Achieva MRI sequences include the following:  1) 1 mm sagittal 3D T1 turbo field echo (TFE) sensitivity encoding (SENSE); 2) 3 mm axial fluid attenuated inversion recovery (FLAIR); 3) 3 mm axial T2 constant level appearance (CLEAR), for brain and face/muscles of mastication; 4) 3 mm coronal short tau inversion recovery (STIR) olfactory bulbs, modified to extend through the pituitary; 5) 2 mm coronal T1 and T2 orbits, modified to extend from the mid-globe through the back of the sella to allow better imaging of the extraocular muscles and optic nerves; 6) 3D gradient echo balanced fast field echo (BFFE) of the brainstem for imaging of most of the cranial nerves; 7) 1mm 3D volume isotropic turbo spin echo acquisition (VISTA) for imaging of the facial nerve cisternal segment and within the internal auditory canal. |
| Area of acquisition | Whole brain and internal auditory canal (IAC) |

Diffusion MRI   ☐ Used   ☒ Not used

## Preprocessing

| | |
|---|---|
| Preprocessing software | N/A |
| Normalization | N/A |
| Normalization template | N/A |
| Noise and artifact removal | N/A |
| Volume censoring | N/A |

## Statistical modeling & inference

| | |
|---|---|
| Model type and settings | N/A |
| Effect(s) tested | N/A |

Specify type of analysis:   ☐ Whole brain   ☐ ROI-based   ☐ Both

| | |
|---|---|
| Statistic type for inference (See Eklund et al. 2016) | N/A |
| Correction | N/A |

## Models & analysis

| n/a | Involved in the study |
|---|---|
| ☒ | ☐ Functional and/or effective connectivity |
| ☒ | ☐ Graph analysis |
| ☒ | ☐ Multivariate modeling or predictive analysis |

