## [Peer Review File · Nature Genetics]

Peer Review Information

Manuscript Title: Non-coding variants alter GATA2 expression in rhombomere 4 motor neurons and cause dominant hereditary congenital facial paresis

Corresponding author name(s): Professor Elizabeth Engle

Reviewer Comments & Decisions:

Decision Letter, initial version:
--

19th Apr 2021

Dear Elizabeth,

Your Article, "Non-coding variants altering a conserved cis-regulatory element cause dominant hereditary congenital facial palsy" has now been seen by 4 referees. You will see from their comments copied below that while they find your work of considerable potential interest, they have raised quite substantial concerns that must be thoroughly addressed. In light of these comments, we cannot accept the manuscript for publication, but would be interested in considering a revised version that addresses these serious concerns.

Reviewer #1 supports eventual publication of this work but thinks that you need to provide more experimental evidence showing that the identified regulatory element (cRE2) is indeed acting as a silencer.

Reviewer #2 is quite positive. Their comments mainly warrant textual changes.

Reviewer #3 also thinks that the characterization of a silencer in the context of a Mendelian disorder is interesting but that you should add further data to fully support the conclusions drawn. They have several constructive suggestions for additional experiments/analyses.

Reviewer #4 also feels that this merits consideration at Nature Genetics but that you need to conduct more functional work, namely in vivo, to corroborate your claims.

In sum, the reviewers' feedback is positive overall but warrants a major revision.

We hope you will find the referees' comments useful as you decide how to proceed. If you wish to submit a substantially revised manuscript, please bear in mind that we will be reluctant to approach the referees again in the absence of major revisions.

If you choose to revise your manuscript taking into account all reviewer comments, please highlight all changes in the manuscript text file. At this stage we will need you to upload a copy of the manuscript in MS Word .docx or similar editable format.

We are committed to providing a fair and constructive peer-review process. Do not hesitate to contact me if there are specific requests from the reviewers that you believe are technically impossible or unlikely to yield a meaningful outcome.

*2) If you have not done so already please begin to revise your manuscript so that it conforms to our Article format instructions, available [here](http://www.nature.com/ng/authors/article_types/index.html). Refer also to any guidelines provided in this letter.

[redacted]

If you wish to submit a suitably revised manuscript we would hope to receive it within 6 months. If you cannot send it within this time, please let us know. We will be happy to consider your revision so long as nothing similar has been accepted for publication at Nature Genetics or published elsewhere. Should your manuscript be substantially delayed without notifying us in advance and your article is eventually published, the received date would be that of the revised, not the original, version.

Nature Genetics is committed to improving transparency in authorship. As part of our efforts in this direction, we are now requesting that all authors identified as 'corresponding author' on published papers create and link their Open Researcher and Contributor Identifier (ORCID) with their account on the Manuscript Tracking System (MTS), prior to acceptance. ORCID helps the scientific community achieve unambiguous attribution of all scholarly contributions. You can create and link your ORCID from the home page of the MTS by clicking on 'Modify my Springer Nature account'. For more information please visit please visit

[href="http://www.springernature.com/orcid">www.springernature.com/orcid.](http://www.springernature.com/orcid)

Thank you for the opportunity to review your work.

Sincerely,

Tiago

Tiago Faial, PhD
Senior Editor
Nature Genetics
<https://orcid.org/0000-0003-0864-1200>

Reviewers' Comments:

Reviewer #1:
Remarks to the Author:

Di Gioia & Tenney et al. report several families affected by hereditary congenital facial palsy (HCFP). Combining classical linkage analysis with WGS, the authors identify 9 unrelated families carrying noncoding SNVs and duplications near GATA2 that affect three potential candidate regulatory elements (cRE1-3). While the HCFP-associated duplications involve all three cREs, the mutations cluster in two regions of cRE2 approximately 20 kb downstream of the target gene. The authors use mobility shift assays (EMSA) to show that the mutated regions specifically bind the transcription factor NR2F1 and that SNVs reduce NR2F1 binding. Next, they use immunostaining and single-cell RNA sequencing to show the expression and some co-expression of Gata2 and Nr2f1 during mouse embryonic facial branch motor neuron (FBMN) development. Finally, the authors perform a series of in vivo LacZ reporter assays to examine the regulatory activity of cRE1-3 and the mutant version of cRE2 in combination with cRE1. They conclude that the regulatory element cRE2 acts as a silencer and cooperates with nearby enhancers to regulate cell type-specific expression of GATA2 during FBMN development.

The manuscript is extremely timely and goes far beyond a classic gene discovery story by addressing the major challenge of noncoding mutations as the cause of human Mendelian disorders. This makes the manuscript very relevant to the entire rare disease community, but also to the common disease field, which also struggles with the functional analysis and interpretation of non-coding variants. The manuscript is well written, the human genetic data are absolutely compelling, the authors combine established and novel experimental approaches, and the data essentially support the conclusions. Therefore, I support the publication of the manuscript, but still have some concerns with the data as currently presented and with the main conclusion about the silencer element:

Major:

I have only one major concern about the current manuscript, and that is about the authors' final conclusion that cRE2 is a silencer element. Why do the duplications and the point mutations result in the same phenotype?

Most duplications of noncoding elements have been shown to lead to increased expression or misexpression of their target genes. Examples include the ZRS at the *Shh* locus (Lettice et al HMG 2003), the *Ihh* locus (Will et al Nature Genetics 2017), and duplications at the *BMP2* locus (Dathe et al AJHG 2008). If cRE2 acts as a silencer element, it is extremely surprising that the duplications and the point mutations have the same effect. Is it possible that the mutations of cRE2 increase *Gata2* expression? The authors need to perform a LacZ reporter assay for the cRE2 with the mutations alone. So far, they have only performed a LacZ reporter assay of the mutant cRE2 together with cRE1. And why did the authors not test all three cREs in one construct? It could be possible that the mutant cRE2 is responsible for the phenotype by producing a gain of function that leads to misexpression of *Gata2*. A similar mutational mechanism has been described for mutations of the limb enhancer ZRS (Lettice et al HMG 2003, Kvon et al Cell 2016). The duplication allele would then also lead to a gain of function of all three cREs, resulting in similar misexpression of *Gata2*.

While the authors show compelling data suggesting loss of NR2F1 binding due to the mutations, no direct effect on *Gata2* expression is presented in the manuscript. The *in vivo* LacZ reporter mouse experiments are very convincing, and the hypothesis is intriguing. However, LacZ reporter constructs are extremely artificial, and the question remains whether cRE2 can actually act as a silencer on *Gata2* expression *in vivo*? Not many silencer elements (if any) have been studied in their normal genomic context. Why did the authors not generate transgenic mice harboring the noncoding mutations or the duplications and perform an *in situ* of *Gata2* expression or verified *Gata2* expression by sc-seq? This should be discussed in the manuscript. The current Figure 6 lacks an *in situ* of *Gata2*. This would make it much clearer.

Minor:

1) Figure 2 is mentioned only once in the manuscript and seems more appropriate for a more clinical journal. It is probably sufficient to show only one representative image of the phenotype, and it seems very difficult for a non-HCFP experienced reader to fully access the MR images of the facial nerves. I would suggest moving the figure to the Supplement and adding one or two representative images of the phenotype to Figure 1.

2) Figure 4 is a very nice figure, but it is not entirely clear what all the information really adds to the main story of the manuscript?

The critical information that of the potential cRE2 target genes (*Gata2* and *Dnajb8*), only *Gata2* is expressed in the hindbrain or head during r4 MN development is buried in the Appendix. This important information should be moved to the main Figure 4.

The entire section is very difficult to read. Why is Figure 4b-r so important here? The fact that the IEE birth precedes the FBMN birth is exciting, but perhaps this could be placed in the supplement. Rather, the figure should highlight the co-localization of NR2F1 and GATA3 / GATA2. The current section and figure should be restructured.

3) Figure 5

Figure 5 is simply stunning. A very nice analysis.

4) In family 6, the phenotype is not fully penetrant. Was there anything different about the phenotype of this family?

5) The Chr3:128178297 A>G variant found in both Fam7 and Fam8 is present in the heterozygous state in six individuals in gnomADv3.1. How is this possible? This is a severe phenotype. Any explanations?

6) Page 7: "Among the 34 HCFP1 locus mutation-positive participants identified in this study, one was clinically unaffected and three with mild weakness had considered themselves unaffected, while others had weakness, while others had more debilitating weakness. Eighty percent had bilateral facial weakness, which was typical"

How does this compare to the HCFP literature? Is this to be expected?

7) Page 8-9

Figure 3c is mentioned in the text before Figure 3b.

8) Page 10

Figure 4d is mentioned in the text before Figure 4b and c.

Malte Spielmann

Reviewer #2:

Remarks to the Author:

This manuscript titled "Non-coding variants altering a conserved cis-regulatory element cause dominant hereditary congenital facial palsy" (MS# NG-A57143; Engle and colleagues) describes the genetics, neurodevelopmental biology, and neuroscience of this specific and interesting dominant trait of hereditary congenital facial palsy (HCFP). Elizabeth Engle and colleagues report six non-coding single-nucleotide variants (SNV) in a conserved cis-regulatory element in seven families and two overlapping tandem duplication CNV (a 31 Kb and 20 Kb tandem duplications) in two families with hereditary facial palsy (HCFP). These variants fall within the chromosome 3 HCFP1 locus; the cis-regulatory elements defined in this disease locus seems to act as a silencer that cooperates with nearby enhancers to regulate cell-type specific expression of GATA2. The mutated cis-regulatory element no longer binds NR2F1 (COUP-TF1) or functions as a silencer.

I found their data and this paper on non-coding variation (CNV & SNV) particularly of interest, intriguing - beautifully executed experiments and science tying the developmental neurobiology of the disease to the genetics and molecular biology. Indeed, their results, and the complex regulation of facial neuron development, demonstrate how variants altering silencer elements can cause a human Mendelizing disease trait. Moreover, I would suggest these investigators also provide insights into penetrance, or lack thereof, and the molecular underpinnings of these genetic terms such as 'penetrance' a phenomenology that has existed for decades in the literature. I also feel they provide further supportive evidence for the gene dosage and expression model of birth defects [Wu et al. (2015) TBX6 Null Variants and a Common Hypomorphic Allele in Congenital Scoliosis. New England

Journal of Medicine 372:341-350; Yang et al. (2019) TBX6 compound inheritance leads to congenital vertebral malformations in humans and mice. Human Molecular Genetics 28:539-547]. This genetic model may be particularly relevant to developmental pathways and those birth defects that affect "organs/systems" that develop on mirror image plains of the body axis and can have asymmetric findings with respect to the birth defects manifestations.

This manuscript is likely to be of great interest to the broad readership of Nature Genetics. There are however, a few comments that the co-authors might want to consider:

1. Introduction; I am not sure I am ready to submit to the statement that non-coding variation has not been more clearly documented with Mendelizing disease traits and the 'implicit' common disease common variant non-coding association. In fact, I would argue that Stuart Orkin in his laboratory has not only understood and demonstrated the role of 'non-coding variation' in Mendelian disease, but they have developed and published the an effective molecular therapy for hemoglobinopathies based on their understanding of non-coding variation and manifestations of disease.
2. Further evidence that such an introductory statement is perhaps misleading, and clearly demonstrating the role of non-coding variation in birth defects, is the beautiful work described in TBX6 associated congenital scoliosis (TACS), congenital anomalies of the kidney and urinary tract (CAKUT), and the lethal lung developmental disease [Yang et al. (2020). Human and mouse studies establish TBX6 in Mendelian CAKUT and as a potential driver of kidney defects associated with the 16p11.2 microdeletion syndrome. Kidney International 98:1020-1030; Karolak et al. (2019). Complex compound inheritance of lethal lung development disorders due to disruption of the TBX-FGF pathway. American Journal of Human Genetics 104:213-228] and interruption of the development of the alveolar capillary system of the lung.
3. Minor, but on page 5, the last paragraph I would suggest changing "copy number variation (CNV)" to "copy number variant (CNV)".
4. I like the whole genome sequencing (WGS) to find the non-coding duplications, AND the ddPCR confirmation, but would like to see a little bit more on "mechanism". Is there microhomology / microhomeology at the breakpoint junction? AND does one observe a de novo junction by PCR in the affected Mom in Fam 2 and the grandparents for control? I could NOT see the breakpoint junction features in figure?
5. Could the authors comment on the thoughts regarding COUP-II (NR2F1) and other cranial nerve developmental abnormalities? Such as optic nerve abnormalities? I do note that they have at least cited this paper as reference 64, but seems to me that there may be more information available about this locus and gene discovery that was published seven years ago perhaps more clinical observations/details regarding associated cranial nerve findings?

Reviewer #3:

Remarks to the Author:

The manuscript by Di Gioia et al. identifies mutations in a putative cis regulatory element (cRE2) as the underlying cause of hereditary congenital facial palsy type 1 (HCFP1). Genetic and genomic data from multiple affected individuals in seven families uncovered two clusters (A and B) of single

nucleotide variants (SNVs) in cRE2 that segregated with the phenotype. Two additional families carried duplications of genomic regions encompassing cRE2 as well as two previously characterized enhancers (cRE1 and cRE3) that regulate GATA2 expression in the embryonic hindbrain. The authors demonstrate that Cluster A mutations interfere with the binding of NR2F1, which they predict is required to repress GATA2 expression in facial branchial motor neurons in order to promote their specification. This model is supported by correlative gene expression data, and an elegant *in vivo* reporter assay in mouse embryos that convincingly demonstrate the role of cRE2 as a repressor of cRE1/3 enhancer activity. The consequence of SNVs on the function of the cRE2 repressor element is also shown, albeit overinterpreted (see below). The study of repressor elements has received far less attention in the literature compared to enhancers, especially in association with human disease, so this study should be of potential interest to the readership of Nature Genetics. Nevertheless, several of the authors' principle conclusions and models are not supported by data and as such, this paper comes across as premature for publication. Listed below are major and minor queries that require attention.

Major concerns:

1. No experiments were performed in this study to address the effects of cRE2 mutations on the cell type specific expression of GATA2. The following statement in the abstract, and others like it in the body of the manuscript are misleading and should be removed.

"The cis-regulatory element acts as a silencer and cooperates with nearby enhancers to regulate celltype specific expression of GATA2."

For the aforementioned statement to be valid, the authors need to demonstrate that mice with Cluster A or B mutations in the endogenous cRE2 repressor, and/or its complete deletion, result in the disinhibition of GATA2 expression in migratory FBMNs. It may seem like a lot to ask, but without this experiment, the results are highly speculative and alternative hypotheses cannot be ruled out. For instance, cRE2 may repress other neighboring genes or lncRNAs.

2. Definitive evidence that SNVs identified in individuals with HCFP fail to bind NR4a2, resulting in GATA2 upregulation is lacking. The generation of mice with mutations in cRE2 would also allow the authors to formally test their claim that mutations in Cluster A (and possibly Cluster B) block NR2F1 recruitment *in vivo*, resulting in GATA2 upregulation in migrating PBMNs.

3. The authors indicate that cRE2 is occupied by NR2F1 in NCC derived iPSCs and shows an ATAC-seq signal in a neuroblastoma cell line. These cell types do not appear to be relevant for investigating cRE2 occupancy, especially since they express GATA2. The authors should evaluate other publicly available genomic datasets (ENCODE) that are more biologically relevant to the cell types of interest (embryonic hindbrain), for chromatin marks, ATAC-seq peaks, etc., associated with gene regulatory activity on cRE2, or generate these data themselves.

4. It is difficult to compare binding affinities with the different SNVs in Figure 3. Dose-response curves should be performed by varying the amount of radiolabeled probe containing each SNV exposed to a constant amount of NR2F1 protein and quantifying band intensity as a measure of NR2F1/cRE2 complex formation.

5. Candidate TFs that bind cluster B were not identified through bioinformatic searches. The authors

should consider mining their scRNA-seq data for TFs and test their ability to bind cluster B in the presence and absence of SNVs. Alternatively, they could perform EMSAs with nuclear extracts from mouse embryos and identify the DNA binding protein by mass spectrometry.

6. In Fig. 6i, Cluster B mutations do not appear to influence cRE2 dependent silencing activity in migrating FBMNs. The minimal X-gal staining outside of r4 in the one embryo (out of 4) is not any different from that displayed by embryos with wild type cRE2 sequence (compare embryo 2 in Fig. S5e with embryo 3 in Fig. S5g). The authors should be more transparent in describing this data and refrain from overinterpreting these results.

7. Of the three classes of variants associated with HCFP, only mutations in Cluster A have a clearly proposed mechanism of action. The model depicted in Fig. S4 is confusing and not representative of the data as it does not show the contribution of cRE2 to target gene repression in the wild type state. Moreover, the consequence of the duplication on repressor activity is not well described. The authors should consider the possibility that a rate limiting trans factor required for cRE2 mediated silencing is diluted below a critical threshold in the presence of the duplication.

8. What is the proposed mechanism of cRE2 mediated silencing of GATA2? Does it block cRE1 and cRE3 from contacting/activating the GATA2 promoter?

Minor concerns:

1. Why do some HCFP1 patients show abnormal ABR and hypoplastic cochlear vestibular ganglia (VIIIth cranial nerve)?

2. The authors refer to cRE1, 2 and 3 as a stretch enhancer, however these enhancers do not appear to fit the original definition, which stipulates that stretch enhancers are clusters of enhancers greater than or equal to 3kb with no gaps in signal (Parker et al., 2013).

3. Fig. S3. Arrows should be added to indicate the supershifted complex. In the last lane of Fig. S3A (both left and right panels) a '+' should be added to the pWT - IRDye 700 row.

4. The following statement suggests that lacZ expression driven from cRE1 is similar to Gata2.

"We found that reporter expression driven by cRE1 alone marked the region of r4 MN precursors and migrating FBMNs, as well as regions of the midbrain and spinal cord (Figure 6b,d, Supplementary Figure 5a), similar to published data and Gata2 expression pattern³⁰."

However, since Gata2 is downregulated in migrating FBMNs, which express lacZ, this statement is not entirely accurate.

5. Methods describing the generation of lacZ reporter knock-ins should be included in the manuscript.

6. According to the results shown in Fig. S5, a single (S) or tandem (T) copy of the transgene was introduced into a site in the mouse genome. This information should also be included in Fig. 6.

7. Fig. 6i. According to the text and figure legend, embryos expressing this reporter construct contain Cluster B mutations, not Cluster A mutations, as labeled in the figure.

Reviewer #4:

Remarks to the Author:

The manuscript entitled "Non-coding variants altering a conserved cis-regulatory element cause dominant hereditary congenital facial palsy" by Di Gioia et al., describes the characterization of new autosomal dominant genetic alterations in 9 families manifesting hereditary congenital facial palsy (HCFP). Interestingly, these alterations affect cis-regulatory elements of the GATA2 gene, encoding a transcription factor playing an important role during early stages of haematopoiesis, and also in several neuronal specification pathways, in particular neuron development in rhombomere 4. These neurons constitute the cranial nerves nVII (Facial branchial motoneurons, FBMNs) and nVIII (vestibulo-acoustic or Inner Ear Efferent, IEEs). In seven families, six single nucleotide variants are found in a conserved element, referred to as cRE2. In addition, a tandem repeat of a genomic region including cRE2, and two further GATA2 cis-regulatory elements, cRE1 and cRE3, were identified in two different families. Three of the 6 SNV overlap a binding site for the NR2F1/COUP-TF1 transcription factor. On these bases, the authors have built complementary experimental strategies to provide evidence that 1) NR2F1 plays an important role in the dichotomy of r4 neuronal development into IEEs and FBMNs, favoring the arising of FBMNs by repressing IEEs; 2) IEEs develop first and FBMNs development is linked to IEEs repression, and 3) repression of IEEs relies on the silencing activity of cRE2, itself depending on NR2F1 binding. The overall data interpretation by the authors is that NR2F1-mediated repression plays an important role in the sequential development of IEEs and FBMNs, allowing the emergence of FBMNs by repressing IEEs.

Due to the wide use of exonic sequencing, the identification of hereditary genetic anomalies on cis-regulatory elements responsible for disease transmission is rare, but has proved to be important to refine the understanding of cellular specific gene regulation and function. In this regard, the heavy genetic studies provided in this manuscript are original and of high interest. They are worth consideration for publication in Nature Genetics, provided the authors revise their functional studies and the misleading interpretation of some of them.

The edition and presentation of the manuscript are generally very satisfactory. My major comments are mainly related to experimental approaches, inappropriate or incomplete interpretation of data, and important information missing, in particular on some experimental procedures.

1. To attest NR2F1 binding to cRE2, the authors rely on different types of data:

- Available data sets generated with neuroblastoma cells or neural crest cells produced from human iPSCs in vitro, which indicate that NR2F1 is able to bind to cRE2
- Electrophoretic Mobility Shift Assays (EMSA), used by the authors to assess the impact of the studied SNVs on NR2F1 binding to the cRE2 cluster A. These experiments were performed with nuclear extracts of Hela and HEK293T cells overexpressing NR2F1.

Given that the binding of a transcription factor to its DNA specific target site largely depends on the cellular context (interaction with cell-type specific partners, proper expression level, post-translational modifications), the presented data are not sufficient to infer a binding of NR2F1 on cRE2 in rhombomere r4 neuroprecursors. To strengthen this idea and eventually provide further information helping better understanding of the mechanisms underlying the cRE2 repressive function, I strongly recommend the authors to repeat EMSA with nuclear protein extracts prepared from hindbrains collected at E10-E10.5, when NR2F1 activity has become critical. Given the low protein amount

required for each test and the easy mechanical dissociation of the neural tube cells at this early developmental stage, these experiments are quite feasible. Furthermore, they will be more appropriate to determine whether further factors bind to the probe, in particular to the cluster B. Ideally, performing methylation protection assays with hindbrain extracts and wild type/cluster A or B mutant probes would be very informative. With this regard, the core of a putative homeobox binding site (TAAT) is to be noted in the close vicinity of the NR2F1 site. In any case, the possible interference of this site occupancy with NR2F1 binding should be discussed, according to the available literature on the Hox genes expressed in r4 and their respective functions.

2. GATA2/GATA3/NR2F1 expression studies

In Figure 4, the data show some GATA2 positive/GATA3 negative (panel s, v, respectively E9.5 and E10.5) and GATA2 positive neurons in FBMN (panel ab, E12.5). What is the nature of these cells? Further, I am very surprised by the ventral axon fascicle emerging from FBMNs and indicated by open arrowheads on panel ab, ac and ad. This is very strange, since, as IEEs axons, FBMNs axons exit in r4, not ventrally in r6, where only the FBMNs cell bodies settle. Hence, the FBMN axon fascicle should be directed dorsally, as it is shown at E10.5 in panels v-x. Is it possible that the picture is taken more posteriorly than at the r6 level? If this is the case, pictures at the r6 level should be provided. Is there any another explanation?

3. r4 neuron birthdating

The authors tried to determine whether IEEs and FBMNs develop in a specific order by performing birthdating experiments, consisting in analyzing the fate of neuronal precursors having incorporated EdU at sequential development stages. EdU was injected at E8.5, E9.0, E10.0 and E10.5, and the neuronal fates were analyzed at E14.5.

- Rather than the total number of EdU positive nuclei, it would be more informative to provide the percentage of EdU positive cells relative to the number of DAPI stained nuclei in each neuron population. This would better reflect the dynamic of each population development.

- EdU exposure from E8.5 comes up with no EdU positive neurons, and this is consistent with high cell proliferation leading to high dilution of EdU within the derived cell lineage and with the fact that the author took into account nuclei exhibiting EdU staining greater than 50%. Fate analysis at E14.5 is far too late, and these data are meaningless. I propose them to be removed.

- Regarding EdU exposure from later stages, I consider that the data have been misinterpreted.

Indeed, the author show clearly in Fig. 4c that IEEs exit cell cycle earlier (E9-E9.5), which is consistent with the small size of this neuron population, and lack of EdU detection after exposure from E10.5.

About FBMNs, EdU detection after EdU exposure from E10 and E10.5 and not from E9 cannot be explained by a switch in the fate, but rather, again, by EdU dilution resulting from high proliferation between E9 and E10. As indicated by the bending curve (figure 4c), cell proliferation decreases from E10, which allows EdU detection in some FBMNs at E14.5.

For these reasons, the provided data do not support the hypothesis of a sequential development of IEEs and FBMNs neurons, but simply that IEEs production is completed earlier, which is more consistent with the literature. Indeed, none of the published data on this topic exclude any function of GATA2/GATA3 in the development of FBMNs, and related citations in the manuscript should be revised accordingly.

To properly assess this issue, the authors have to perform EdU injections between E9 and 10,5 and analyze neuronal fates no later than 24 hours after each injection. For better interpretation, I recommend to determine the percentage of EdU positive neurons in each neuron population at each time point.

Importantly, since NR2F1 has been largely involved in functions linked to neuronal migration, the

authors should take into account the hypothesis according to which impaired NR2F1 binding to cRE2 could result in blocking FBMN in r4, and in the death of part of them. Counting IEEs and FBMNs neurons, and checking survival and migration of FBMNs in NR2F1 mutant embryos, or in transgenic embryos carrying cRE2 mutated in cluster A, would be also important to gain insight into the function of NR2F1 and the impact of its binding to cRE2 during neurogenesis r4 .

4. Comments on scRNAseq analyses

- Indicating the number of each cell cluster on the trajectory plot in figure 5a would make the data easier to read. As well, it is important to indicate at least one or two master genes for each one of the 13 cell clusters.
- The authors mention that clustering from scRNAseq analysis included only Hoxb1 positive cells, Isl1 positive or not. Hoxb1 is indeed widely expressed in r4 progenitors as early as E8.5 and is considered as being the master gene of neuronal fate determination in r4. However, in Figure 5b, the distribution of Hoxb1 positive cells appears to be low in the progenitor cluster, even lower than that of cells expressing NR2F1.
- GATA3 and GATA2 are clearly expressed in ventral interneurons (ref 26 and 44), and expression should have been detected in at least one of the three interneuron clusters, but it is not the case. Can the authors explain these two results and eventually adjust their clustering strategy? In any case, these points have to be discussed in more details.

5. The authors have assessed the *in vivo* functionality of GATA2 cis-regulatory elements, wild type and mutated, by generating spare transgenic embryos. They give no information on the procedure used to generate mutated versions of the elements. In the Methods section (p45), the authors mention the injection of sgRNA, cas9 and donor plasmid into pronuclei, with no further information. This procedure, including sgRNAs specificities and how mutations were checked, has to be described in details. The authors provide the number of transgenic embryos expressing the reporter gene LacZ obtained with each construct. In order to provide a better idea of the relative functionality of the cis-regulatory elements, in particular of the impact of the association of cRE2, wild type or mutated, with cRE1 or cRE3 versus cRE1 and cRE3 alone, on the ratio of the embryos showing lacZ expression to the total number of transgenic embryos would be worth mentioning.

- As now, summary and conclusions are not appropriate and will have to be revised according to the above comments and the new information included in the revised manuscript.

Minor points

1. In the second line of the last paragraph page 9, in addition to ref 26, the fundamental work of Studer et al. (PMID: 8967950) should be also cited
2. Only truncated pictures of EMSA are provided in figure 3 and supplementary figure 3s, showing separately the gel area containing the band shift due to NR2F1 binding and the one with the free probe only. Since binding of further factors cannot be excluded, it would be more correct to show pictures of the entire gel.
2. Fig 6i, cluster A is indicated instead of cluster B
3. Fig 6k, if the authors confirm their hypothesis, the schema 3 should display a lower number of FBMNs in the HCFP1 context.
4. The authors suggest only Id2 as a possible NR2F1 partner responsible for repression mediation. Another NR2F1 partner that could be taken into consideration is FOG2. Did the author get any information on FOG2 expression in their scRNAseqs?

Author Rebuttal to Initial comments**Response to Referees**

Dear Referees,

Thank you for your thorough and thoughtful review of our manuscript, NG-A57143. We have made major revisions to the initial submission to address your valuable comments, concerns, and suggestions. This includes the addition of mouse models and additional experiments, and hence has taken an extended period of time. Major additions to the paper include (but are not limited to) the following:

1. We studied conditional *Gata2* (unpublished, developed by Dr. Stu Orkin) and *Gata3* (published) knock-out mice in which the genes are deleted from r4 motor neurons and the mice survive to adulthood. Embryos have loss of IEEs but preservation of the facial nucleus. We developed a whisking assay to assess facial nerve function in adult mice and found whisking to be normal in both conditional knockouts. This establishes that *Gata2* and *Gata3* are essential for IEE but not FBMN development.
2. We conducted single cell CUT&Tag with an NR2F1 antibody on FACS-sorted embryonic r4 motor neurons. We show that NR2F1 binds to cRE1, cRE2, and minimally to cRE3 in developing wildtype r4 motor neurons. We then developed a cRE2 Cluster A SNV mouse model and show that binding of NR2F1 to cRE2 is reduced specifically in developing r4 motor neurons in this HCFP1 model. Notably, this mouse does not have an anatomic or whisking phenotype; we are continuing to investigate this, but hypothesize that it is because of the NR2F1 binding site present in mouse cRE1 (based on both TF binding site and CUT&Tag data. This binding site is not present in human cRE1 (based on lack of conservation of the TF binding site sequence), and hence the cREs may be being used slightly differently in mice compared to humans.
3. We made an HCFP1 duplication mouse that has CFW. To circumvent the NR2F1 binding to cRE1, we inserted two copies of human cRE1 adjacent to the endogenous mouse copy. *cRE1^{dup/+}* embryos have a small to absent facial nucleus, significant loss of ISL1^{ON};GATA2^{OFF} facial motor neurons, and a significant increase in ISL1^{ON};GATA2^{ON} neurons that are most consistent with an IEE cell identity. Profiling of r4MN in these mice revealed ectopic GATA2 expression in FMN precursors, supporting the regulatory role cRE1. We show these findings by immunohistochemistry, semi-automated cell counts, and scRNAseq. Moreover, birthdating of r4MN in WT versus the cRE1 duplication embryos reveals a temporal extension of ISL1^{ON};GATA2^{ON} IEE cell birth in the mutant compared to WT. Finally, *cRE1^{dup/+}* adult mice have absent whisking.

4. We partially rescue the $cRE1^{dup/+}$ embryonic facial nucleus in r6 and the ectopic $ISL1^{ON};GATA2^{ON}$ and $ISL1^{ON};GATA2^{OFF}$ neurons in r4 by crossing the $cRE1^{dup/+}$ mice to conditional *Gata3* knockout mice. Adult mice have variable rescue of whisking.
5. While not requested, we have also further strengthened the genetics of HCFP1 by establishing collaboration with the Dutch geneticists who previously defined the HCFP1 locus in two pedigrees (Kremer et al, 1996; Michielse et al, 2006). We now include both these original two pedigrees, one that harbors an overlapping duplication (Fam10) and one that harbors an SNV (Fam13), as well as two additional duplications (Fam11, Fam12) and an additional SNV (Fam14) found in their WGS analysis of isolated CFW probands. Thus, we now report 14 pedigrees, of whom 5 harbor overlapping duplications of *cRE1*, *cRE2*, and *cRE3*, and 9 harbor SNVs in *cRE2*.

In sum, we feel we demonstrate more convincingly that *Gata2* is indeed the target of these noncoding variants, and that NR2F1 binds to r4MN *in vivo* and this binding is reduced in the SNV mutant. Thus, we submit for re-review a substantially revised and far more comprehensive manuscript.

Below we detail, point-by-point, how we have addressed each of your comments.

Reviewer #1:

Major:

I have only one major concern about the current manuscript, and that is about the authors' final conclusion that cRE2 is a silencer element.

*Most duplications of noncoding elements have been shown to lead to increased expression or misexpression of their target genes. Examples include the ZRS at the *Shh* locus (Lettice et al HMG 2003), the *Ihh* locus (Will et al Nature Genetics 2017), and duplications at the *BMP2* locus (Dathe et al AJHG 2008). If *cRE2* acts as a silencer element, it is extremely surprising that the duplications and the point mutations have the same effect. Is it possible that the mutations of *cRE2* increase *Gata2* expression? The authors need to perform a LacZ reporter assay for the *cRE2* with the mutations alone. So far, they have only performed a LacZ reporter assay of the mutant *cRE2* together with *cRE1*. And why did the authors not test all three *cREs* in one construct? It could be possible that the mutant *cRE2* is responsible for the phenotype by producing a gain of function that leads to misexpression of *Gata2*. A similar mutational mechanism has been described for mutations of the limb enhancer ZRS (Lettice et al HMG 2003, Kvon et al Cell 2016). The duplication allele would then also lead to a gain of function of all three *cREs*, resulting in similar misexpression of *Gata2*.*

We propose that the SNVs and duplications both indeed result in gain of *Gata2* expression (and we now show this is the case in the duplication mouse model). The SNVs would do so through loss-of-function alleles that eliminate/attenuate cRE2 function (SNVs in the silencer would result in its inability to silence, resulting in increased *Gata2* expression).

Because we have identified 7 different SNVs altering 6 nucleotides among 9 pedigrees in two distinct clusters in cRE2, they were more likely to abolish/attenuate cRE2 function (acting as loss-of-function alleles) rather than each result in similar gain of function activity. If cRE2 has the ability to silence cRE1 and cRE3 activity in migrating facial motor neurons (as supported by the *in vivo* LacZ experiments), the SNVs would abolish this silencing and result in ectopic *Gata2* expression. Similarly, duplication of the two enhancers and one silencer would result in a net increase in overall enhancer activity, also increasing *Gata2* expression. In this way, duplications and point mutations would result in the same phenotype.

That said, it is possible that mutant cRE2 alone does gain enhancer function - in this case we would expect that if this gain-of-function results in the CFW phenotype, then the same gain of function would be seen for each of the SNVs, given they result in the same phenotype. To address this, as suggested by the reviewer, we performed a LacZ reporter assay for the cRE2 cluster A mutants (containing 3 of the SNVs) and for the cluster B mutants (also containing 3 of the SNVs) alone and present these data in Extended Data Figure 3h,i. Among the 6 embryos expressing the cRE2 Cluster A SNVs alone, two single insertions and one tandem insertion are white (no expression, similar to WT cRE2), while three additional tandems do gain expression in a FMN/IEE pattern. Among the 8 embryos expressing the cRE2 Cluster B SNVs alone, all five singles are white, while two tandems have very faint IEE expression and one tandem has more widespread expression. Therefore, it is possible that abolishing the binding of NR2F1 to cRE2 might reveal residual enhancer activity. We have now rewritten the text to include this possibility as follows:

Result, page 15: Interestingly, expression of cRE2*A or cRE2*B alone showed some neuronal signal in tandem but not single transgenic embryos (Extended Data Fig. 3h,i).

Discussion, page 26: Finally, introduction of the cRE2 SNVs in our lacZ assay unveils enhancer activity, likely through the opportunistic binding of other TFs which could also vary between mice and humans⁷⁹. For example, in humans but not in mice, there is a potential homeobox consensus sequence, TAAT, 20 bp upstream of Cluster A that could

become more accessible by loss of the NR2F1 binding site. We do not yet know if introducing a single cRE2 Cluster A SNV in the lacZ assay would enhance β -galactosidase expression, or if this in vivo enhancer activity contributes to the HCFP1-SNV phenotype.

We did not test all three cREs in one construct given that the new mouse model with human cRE1 duplication ($cRE1^{dup/+}$) demonstrates Gata2 misexpression and congenital facial paralysis.

While the authors show compelling data suggesting loss of NR2F1 binding due to the mutations, no direct effect on Gata2 expression is presented in the manuscript. The in vivo LacZ reporter mouse experiments are very convincing, and the hypothesis is intriguing. However, LacZ reporter constructs are extremely artificial, and the question remains whether cRE2 can actually act as a silencer on Gata2 expression in vivo? Not many silencer elements (if any) have been studied in their normal genomic context. Why did the authors not generate transgenic mice harboring the noncoding mutations or the duplications and perform an in situ of Gata2 expression or verified Gata2 expression by sc-seq? This should be discussed in the manuscript. The current Figure 6 lacks an in situ of Gata2. This would make it much clearer.

We now present data of 2 HCFP1 mouse models ($Fam5^{snv/snv}$ and $cRE1^{dup/+}$). Our *in vivo* scCUT&Tag on WT r4MNs shows that NR2F1 does indeed bind to cRE2 and also reveals strong binding to cRE1 and minimal binding to cRE3. The $Fam5^{snv/snv}$ mouse line (which harbors a single cRE2 cluster A SNV) does not have a phenotype (and hence does not recapitulate human HCFP1), but NR2F1 binding to cRE2 is reduced in $Fam5^{snv/snv}$ r4MNs *in vivo* by scCUT&Tag (Fig. 4l-n). We hypothesize that the absence of an anatomic or behavioral phenotype in these mice may be due to NR2F1 binding to a site in cRE1 in mice (as shown by scCUT&Tag), which is not predicted in humans based on differing sequences between the two species. Thus, we generated a second mouse model that introduced two copies of the human cRE1 in *cis*. This mouse ($cRE1^{dup/+}$) has congenital facial weakness, and the embryo has a reduction in the size of the facial nucleus and the number of ISL1^{ON};GATA2^{OFF};GATA3^{OFF} facial motor neurons (Fig. 6a-k), and an increase in the number of ISL1^{ON};GATA2^{ON};GATA3^{ON} neurons, primarily located in rhombomere 4 (IEEs - both OCNs, VENs as well as ectopic neurons) (Fig. 6i-o, Extended Data Fig. 7,8). These shifts in cell identity and increase in *Gata2*-expressing neurons are also seen by single-cell RNAseq at multiple ages (Fig. 5a-c,f, Extended Data Fig. 6c). The conditional deletion of the *Gata2* target gene, *Gata3*, in $cRE1^{dup/+}$ embryos rescues the facial nucleus and in adults rescues whisking partially to fully. We believe the congruence between the gene regulatory etiology we propose for HCFP1 and the molecular,

histological and behavioral phenotypes in the $cRE1^{dup/+}$ HCFP1 model mice support our assertions that HCFP1 SNVs and duplications both lead to expanded *Gata2* expression and the loss of FBMNs.

We have added images of *Gata2* and *Isl1* whole mount *in situ* hybridization on E11.5 embryos (Fig. 3b) oriented to serve as anatomic references for the LacZ transgenic embryos (Fig. 4a-i and Extended Data Fig. 3).

Minor:

1) Figure 2 is mentioned only once in the manuscript and seems more appropriate for a more clinical journal. It is probably sufficient to show only one representative image of the phenotype, and it seems very difficult for a non- HCFP experienced reader to fully access the MR images of the facial nerves. I would suggest moving the figure to the Supplement and adding one or two representative images of the phenotype to Figure 1.

We appreciate the reviewer's comments. Unfortunately, there is not adequate space in Fig. 1 to add these photos. We do, however, have space for this figure in the main text and we would like to present the clinical phenotype and MR results together in one figure (Fig. 2). We have edited the text to emphasize the importance of determining that the cRE2 CNVs and cRE1-3 duplications cause indistinguishable phenotypes, as this supports a common neurodevelopmental mechanism.

*2) Figure 4 is a very nice figure, but it is not entirely clear what all the information really adds to the main story of the manuscript? The critical information that of the potential cRE2 target genes (*Gata2* and *Dnajb8*), only *Gata2* is expressed in the hindbrain or head during r4 MN development is buried in the Appendix. This important information should be moved to the main Figure 4. The entire section is very difficult to read. Why is Figure 4b-r so important here? The fact that the IEE birth precedes the FBMN birth is exciting, but perhaps this could be placed in the supplement. Rather, the figure should highlight the co-localization of NR2F1 and GATA3 / GATA2. The current section and figure should be restructured.*

We have reorganized and expanded the histological data to incorporate analyses of our new *Gata2* and *Gata3* conditional loss-of-function, HCFP1 mouse models, and the rescue mouse model, highlighting expressions of ISL1, GATA2, GATA3, and NR2F1. This histology supports our overarching model of an IEE to FBMN cell fate switch that, when impaired, results in facial paralysis. We now provide additional data that *Dnajb8* expression is absent from the hindbrain of WT and *cRE1^{dup/+}* mice as assayed across a developmental time course using single cell transcriptomics (Extended Data Fig. 6c,d) and histology (Extended Data Fig. 9), making it an unlikely mediator of the HCFP1 phenotype. We have also updated the previous presentation of the sequential IEE and FBMN birth order with new cell count and birthdating data showing that the switch from IEE to FBMN generation is disrupted in the *cRE1^{dup/+}* HCFP1 model mice, likely causing a reduction in FBMN number and the observed HCFP1 phenotypes (Fig. 6l-p).

3) *Figure 5 is simply stunning. A very nice analysis.*

Thank you. For the revision, we redid the scRNAseq analysis of developing r4MNs, this time dissecting and dissociating r3-r7 from E9.5-E12.5 *Isl1^{MN}-GFP⁺* WT and *cRE1^{dup/+}* littermates (Fig. 5, Extended Data Fig. 6). Cranial motor neurons were purified with FACS, and we “spiked in” a smaller number of cells from the surrounding GFP⁻ tissue to capture the early steps of r4N development that precede expression of the reporter. The resulting WT and *cRE1^{dup/+}* scRNAseq data integrated well when merged into a single Seurat object and appeared free of batch effects. Using both previously validated and newly discovered markers, we identified the r4MN developmental trajectory and found that, while r4MN cell number did not change between the genotypes, the *cRE1* duplication caused expansive, ectopic *Gata2* expression at earlier stages in this lineage compared to WT. IEE cluster size increased, and FBMN cluster size correspondingly shrank in the *cRE1^{dup/+}* embryos compared to WT, presaging the significant changes in IEE and FBMN number we subsequently quantified histologically (Fig. 6l-o). We also show that the *cRE1^{dup/+}* r4MN trajectory clusters have an increase in cells expressing IEE markers (*Gata2*, *Gata3*) with a reduction in cells expressing FBMN markers (*Syt4*, *Nr2f1*) (Fig. 5e).

Our updated scRNAseq analysis also shows how the complementarity of *Gata2* and *Nr2f1* expression seen in WT r4MNs at E10.5 is erased in the *cRE1^{dup/+}* embryos by the retrograde expansion of *Gata2* into Cluster 2 mitotic r4 MN progenitors (Fig. 5f, Extended Data Fig. 6d). This extension of *Gata2* and *Nr2f1* coexpression could disrupt the generation of *Gata2⁺;Nr2f1⁺* FBMNs (Fig. 5g).

4) In family 6, the phenotype is not fully penetrant. Was there anything different about the phenotype of this family?

Notably, there are 4 individuals (one in Fam3, two in Fam8, one in Fam14) who thought they were unaffected but in whom mild facial weakness was diagnosed by author TAH (an expert in facial nerve function) by analysis of video and photos while blinded to genotype. By contrast, 2 individuals (one in Fam7 and one in Fam14) were similarly reviewed and had no evidence of weakness, and hence are scored as nonpenetrant variant carriers. Unfortunately, we do not have access to photos or videos from the mother and grandmother in Fam6, and so we do not know if they have mild facial weakness or are truly unaffected; for this reason they are labeled with a '?' in Fig. 1a and noted as such in Table 1.

We are able to make suggestive phenotype-genotype correlations as follows:

Results, page 11: Among the 37 variant-positive participants with phenotypic documentation through a National Institute of Health (NIH) Clinical Research Center evaluation, videos, or informative photos, 2 were clinically unaffected and 4 with mild weakness had considered themselves unaffected (Fig. 1, Table 1). These 6 individuals all harbor SNVs and 3 harbor rs987263273, suggesting that these SNVs potentially cause a milder CFP phenotype than the duplications.

and

Thus, HCFP1 is neurogenic¹⁵, and both SNVs and duplications cause nonsyndromic, mild- to moderate-severity CFP phenotypes that were not distinguishable by our clinical evaluation, supporting a shared neurodevelopmental mechanism.

5) The Chr3:128178297 A>G variant found in both Fam7 and Fam8 is present in the heterozygous state in six individuals in gnomADv3.1. How is this possible? This is a severe phenotype. Any explanations?

CFW that results from these noncoding variants at the HCFP1 locus is not necessarily a severe phenotype, and varies from facial weakness treated with surgery to non-penetrant carriers.

Notably, Chr3:128178297 A>G that is in gnomAD also appears to segregate with the most mild phenotype. Of the five individuals among 2 families who harbor this variant, only two (one in each family) had been diagnosed with facial weakness. Enrolling the entire nuclear family, however, and blindly assessing videos of their facial movements without knowledge of genotype, we determined that two individuals who self-reported as unaffected actually have very mild facial weakness, while one had no detectable weakness and is indicated as a carrier. Thus, it is not surprising that this particular SNV is in gnomAD at very low frequency. From a molecular standpoint, this change localizes on what we define as Cluster B, for which we haven't identified a conserved TF binding site that might be affected by the variants. Therefore, we hypothesize that this mutation simply attenuates rather than abolishes the binding of a specific TF, causing the observed reduced penetrance.

Please also refer to the response to question 4.

6) Page 7: "Among the 34 HCFP1 locus mutation-positive participants identified in this study, one was clinically unaffected and three with mild weakness had considered themselves unaffected, while others had weakness, while others had more debilitating weakness. Eighty percent had bilateral facial weakness, which was typical' How does this compare to the HCFP literature? Is this to be expected?"

While we have been unable to find studies of the sidedness of hereditary congenital facial palsy, it is reassuring that the addition of five pedigrees in revision did not change the bilateral statistic significantly (from 80% to 83%). This statement has been revised to include data from these new pedigrees. In addition, we do have data from a subset of the simplex (singleton) variant-negative research participants showing that most have unilateral facial weakness, and this is included in the Supplementary Clinical note.

Results, page 11: Among the 35 participants with visible facial weakness, 83% (29/35) had bilateral weakness which was typically asymmetrical both with regard to sidedness and upper versus lower face (Fig. 2)

Please also refer to the response to question 4 above.

Supplementary Clinical Note: Of the 3 simplex cases who were mutation-positive, one harbored a *de novo* variant (Fam5:II-1), the second was the offspring of the examined and clinically unaffected mutation-positive father (Fam7:I-1), and the third was the offspring of the unexamined but reportedly unaffected mutation-positive parent and grandparent (Fam6:III-1). All three of these individuals had bilateral facial weakness. By contrast, among the 32 mutation-negative simplex cases we screened, only 4 (12.5%) were reported to have bilateral facial weakness.

We feel these findings are consistent with the genetic forms of CCDDs, and many disorders, having a higher rate of bilateral affection compared to simplex cases that are less likely to result from germline variants.

7) Page 8-9 Figure 3c is mentioned in the text before Figure 3b.

8) Page 10 Figure 4d is mentioned in the text before Figure 4b and c.

Thank you, we have fixed these and checked that all are mentioned in order. The only exception is Fig. 6, which we refer to as a whole first, but then could not find a way to have the panels be referred to in alphabetical order without changing the format and likely confusing the reader. So they are mentioned a bit out of order.

Malte Spielmann

Reviewer #2:

This manuscript titled “Non-coding variants altering a conserved cis-regulatory element cause dominant hereditary congenital facial palsy” (MS# NG-A57143; Engle and colleagues) describes the genetics, neurodevelopmental biology, and neuroscience of this specific and interesting dominant trait of hereditary congenital facial palsy (HCFP). Elizabeth Engle and colleagues report six non-coding single-nucleotide variants (SNV) in a conserved cis-regulatory element in

seven families and two overlapping tandem duplication CNV (a 31 Kb and 20 Kb tandem duplications) in two families with hereditary facial palsy (HCFP). These variants fall within the chromosome 3 HCFP1 locus; the cis-regulatory elements defined in this disease locus seems to act as a silencer that cooperates with nearby enhancers to regulate cell-type specific expression of GATA2. The mutated cis-regulatory element no longer binds NR2F1 (COUP-TF1) or functions as a silencer.

I found their data and this paper on non-coding variation (CNV & SNV) particularly of interest, intriguing – beautifully executed experiments and science tying the developmental neurobiology of the disease to the genetics and molecular biology. Indeed, their results, and the complex regulation of facial neuron development, demonstrate how variants altering silencer elements can cause a human Mendelizing disease trait. Moreover, I would suggest these investigators also provide insights into penetrance, or lack thereof, and the molecular underpinnings of these genetic terms such as ‘penetrance’ a phenomenology that has existed for decades in the literature. I also feel they provide further supportive evidence for the gene dosage and expression model of birth defects [Wu et al. (2015) TBX6 Null Variants and a Common Hypomorphic Allele in Congenital Scoliosis. New England Journal of Medicine 372:341-350; Yang et al. (2019) TBX6 compound inheritance leads to congenital vertebral malformations in humans and mice. Human Molecular Genetics 28:539-547]. This genetic model may be particularly relevant to developmental pathways and those birth defects that affect “organs/systems” that develop on mirror image plains of the body axis and can have asymmetric findings with respect to the birth defects manifestations.

This manuscript is likely to be of great interest to the broad readership of Nature Genetics. There are however, a few comments that the co-authors might want to consider:

1. Introduction; I am not sure I am ready to submit to the statement that non-coding variation has not been more clearly documented with Mendelizing disease traits and the ‘implicit’ common disease common variant non-coding association. In fact, I would argue that Stuart Orkin in his laboratory has not only understood and demonstrated the role of ‘non-coding variation’ in Mendelian disease, but they have developed and published the an effective molecular therapy for hemoglobinopathies based on their understanding of non-coding variation and manifestations of disease.

2. Further evidence that such an introductory statement is perhaps misleading, and clearly demonstrating the role of non-coding variation in birth defects, is the beautiful work described in TBX6 associated congenital scoliosis (TACS), congenital anomalies of the kidney and urinary

tract (CAKUT), and the lethal lung developmental disease [Yang et al. (2020). Human and mouse studies establish TBX6 in Mendelian CAKUT and as a potential driver of kidney defects associated with the 16p11.2 microdeletion syndrome. Kidney International 98:1020-1030; Karolak et al. (2019). Complex compound inheritance of lethal lung development disorders due to disruption of the TBX-FGF pathway. American Journal of Human Genetics 104:213-228] and interruption of the development of the alveolar capillary system of the lung.

Thank you for these comments, and we agree. We have shortened (necessary due to the length of the revised manuscript) and refocused our statements, and added additional non-coding references.

Introduction, page 7:The non-coding human genome contains *cis*-regulatory elements (cREs) that can be bound by transcription factors (TFs) and act as cell-type specific enhancers or silencers to define complex, combinatorial gene regulatory programs¹⁻³. Recent genomic advances have revealed important contributions of cRE variants to rare Mendelian disorders⁴⁻⁶. However, determining the mechanistic effects of pathogenic cRE variants can be difficult due to the need to study them in their relevant cellular and temporal context⁸⁻¹⁴. Such studies can be even more challenging for developmental disorders where the fate of a small number of progenitor cells are defined by tightly regulated gene expression networks.

With regard to penetrance in humans, we do discuss the lack of penetrance of one individual with the Cluster B SNV that is present in gnomAD, the trend toward a less severe CFW with SNVs compared to duplications, and the high frequency of bilateral CFW as well as variable severity between the left and right side of the face. Interestingly, the *cRE1^{dup/+}* mice have complete penetrance of absent whisking bilaterally.

3. *Minor, but on page 5, the last paragraph I would suggest changing “copy number variation (CNV)” to “copy number variant (CNV)”.*

We appreciate the suggestion and have edited the text accordingly.

4. I like the whole genome sequencing (WGS) to find the non-coding duplications, AND the ddPCR confirmation, but would like to see a little bit more on “mechanism”. Is there microhomology / microhomeology at the breakpoint junction? AND does one observe a *de novo* junction by PCR in the affected Mom in Fam 2 and the grandparents for control? I could NOT see the breakpoint junction features in figure?

Thank you for this suggestion. We have now provided the breakpoint PCR data and evaluated the pedigrees with duplications for microhomology at the breakpoint junctions with Clustal Omega (<http://www.ebi.ac.uk/Tools/msa/clustalo/>; European Bioinformatics Institute). We have added the following text, as well as indicated the microhomology in Extended Data Fig. 1e.

Results, page 10: Fam2 and Fam10-12 each had microhomology identified at the breakpoint junctions, suggesting these duplications originated by replication-based microhomology-mediated repair mechanisms^{26,27}. Fam1 did not display microhomology, and instead had a three nucleotide base pair insertion (GAA) at the breakpoint of the tandem duplication (Extended Data Fig. 1e)

Extended Data Fig 1 legend, page 39: Sanger sequencing traces that define the tandem duplication breakpoints (vertical black line) for each pedigree. Arrows preceding and following the vertical line indicate the most distal and proximal nucleotide in the duplication, respectively. Fam1 has an insertion of nucleotides GAA at the breakpoint (underlined). Fam2, Fam10, Fam11, and Fam12 have microhomology identified at the breakpoint (red boxed nucleotides).

For Fam2, as shown in the haplotypes in Extended Data Fig. 1b, the maternal duplication arose *de novo* on the maternal grandfather’s allele. The maternal grandfather was deceased at the start of the study and thus we do not have his DNA, and his haplotype was assumed based on the SNP data of the mother, maternal grandmother and two maternal uncles. Thus, while we are able to show (1) the haplotype to establish the *de novo* status, (2) the breakpoint PCR in the mother (Extended Data Fig. 1e) and (3) the segregation of the duplication by ddCPR data for this family, we do not have the grandfather’s DNA to sequence in order to show the *de novo* breakpoint directly.

5. Could the authors comment on the thoughts regarding COUP-II (NR2F1) and other cranial nerve developmental abnormalities? Such as optic nerve abnormalities? I do note that they

have at least cited this paper as reference 64, but seems to me that there may be more information available about this locus and gene discovery that was published seven years ago perhaps more clinical observations/details regarding associated cranial nerve findings?

We did not observe other cranial nerve phenotypes in the research participants (refer to Table 1), and the participants seen at NIH underwent detailed exams by ophthalmologists. Notably, however, one might not expect optic nerve or other non-facial/IEE abnormalities given that we feel that these cRE motifs are cell-type specific in their actions, and cRE2 and cRE3 may only be active in a small number of cells in r4. Thus, the variants would only be expected to perturb NR2F1 function in these specific cell types. Of note, we did find reports of several patients harboring *NR2F1* variants who have thin facial nerves or mild facial weakness as part of their more extensive phenotype. We mention this in the discussion with references:

Discussion page 25: Fourth, while human haploinsufficiency of *NR2F1* (MIM #615722) causes a variable phenotype characterized primarily by intellectual disability and optic nerve degeneration⁷³, several individuals have been reported to also have a thin facial nerve or mild facial weakness^{74,75}.

Reviewer #3:

The manuscript by Di Gioia et al. identifies mutations in a putative cis regulatory element (cRE2) as the underlying cause of hereditary congenital facial palsy type 1 (HCFP1). Genetic and genomic data from multiple affected individuals in seven families uncovered two clusters (A and B) of single nucleotide variants (SNVs) in cRE2 that segregated with the phenotype. Two additional families carried duplications of genomic regions encompassing cRE2 as well as two previously characterized enhancers (cRE1 and cRE3) that regulate GATA2 expression in the embryonic hindbrain. The authors demonstrate that Cluster A mutations interfere with the binding of NR2F1, which they predict is required to repress GATA2 expression in facial branchial motor neurons in order to promote their specification. This model is supported by correlative gene expression data, and an elegant in vivo reporter assay in mouse embryos that convincingly demonstrate the role of cRE2 as a repressor of cRE1/3 enhancer activity. The consequence of SNVs on the function of the cRE2 repressor element is also shown, albeit overinterpreted (see below). The study of repressor elements has received far less attention in

the literature compared to enhancers, especially in association with human disease, so this study should be of potential interest to the readership of Nature Genetics. Nevertheless, several of the authors' principle conclusions and models are not supported by data and as such, this paper comes across as premature for publication. Listed below are major and minor queries that require attention.

Major concerns:

1. No experiments were performed in this study to address the effects of cRE2 mutations on the cell type specific expression of GATA2. The following statement in the abstract, and others like it in the body of the manuscript are misleading and should be removed. "The cis-regulatory element acts as a silencer and cooperates with nearby enhancers to regulate celltype specific expression of GATA2." For the aforementioned statement to be valid, the authors need to demonstrate that mice with Cluster A or B mutations in the endogenous cRE2 repressor, and/or its complete deletion, result in the disinhibition of GATA2 expression in migratory FBMNs. It may seem like a lot to ask, but without this experiment, the results are highly speculative and alternative hypotheses cannot be ruled out. For instance, cRE2 may repress other neighboring genes or lncRNAs.

As stated above in response to Reviewer 1: We now present data of 2 HCFP1 mouse models (*Fam5^{snv/snv}* and *cRE1^{dup/+}*). Our *in vivo* scCUT&Tag on WT r4MNs shows that NR2F1 does indeed bind to cRE2 and also reveals strong binding to cRE1 and minimal binding to cRE3. The *Fam5^{snv/snv}* mouse line (which harbors a single cRE2 cluster A SNV) does not have a phenotype (and hence does not recapitulate human HCFP1), but NR2F1 binding to cRE2 is reduced in *Fam5^{snv/snv}* r4MNs *in vivo* by scCUT&Tag (Fig. 4I-n). We hypothesize that the absence of an anatomic or behavioral phenotype in these mice may be due to NR2F1 binding to a site in cRE1 in mice (as shown by scCUT&Tag), which is not predicted in humans based on differing sequences between the two species. Thus, we generated a second mouse model that introduced two copies of the human cRE1 in *cis*. This mouse (*cRE1^{dup/+}*) has congenital facial weakness, and the embryo has a reduction in the size of the facial nucleus and the number of ISL1^{ON};GATA2^{OFF};GATA3^{OFF} facial motor neurons, and an increase in the number of ISL1^{ON};GATA2^{ON};GATA3^{ON} neurons, primarily located in rhombomere 4 (IEEs - both OCNs, VENs as well as ectopic neurons) (Fig. 6; Extended Data Fig. 7,8). These shifts in cell identity and increase in *Gata2*-expressing neurons are also seen by single-cell RNAseq at multiple ages (Fig. 5b,c,f; Extended Data Fig. 6c). The conditional deletion of *Gata3* in *cRE1^{dup/+}* embryos rescues the facial nucleus and in adults rescues whisking partially to fully. We believe the congruence between the gene regulatory etiology we propose for HCFP1 and the molecular, histological and behavioral phenotypes in the *cRE1^{dup/+}* HCFP1 model mice support our

assertions that HCFP1 SNVs and duplications both lead to expanded *GATA2* expression and the loss of FBMNs. These data unequivocally point to *GATA2* and its downstream effector *GATA3* as major regulators of FBM/IEE switch and showed how altering this pathway can lead to facial weakness in humans and mice.

The locus for the molecular chaperone *Dnajb8* is within this regulatory region and could be a target of cRE2-mediated regulation, but we found it is not expressed in the developing hindbrains of either WT or *cRE1^{dup/+}* embryos (Extended Data Fig. 9). Similarly, *DNAJB8-AS1* expression in human appears to be limited to testis (<https://www.gtexportal.org/home/gene/DNAJB8-AS1>), it is not conserved in mouse, and two of the duplications added in revision duplicate only a portion of the lncRNA. That said, we cannot formally exclude the possibility that the variants regulate *Gata2* indirectly through a lncRNA.

2. Definitive evidence that SNVs identified in individuals with HCFP fail to bind NR4a2, resulting in GATA2 upregulation is lacking. The generation of mice with mutations in cRE2 would also allow the authors to formally test their claim that mutations in Cluster A (and possibly Cluster B) block NR2F1 recruitment in vivo, resulting in GATA2 upregulation in migrating PBMNs.

As mentioned in the preceding response, we have generated a mouse model of a Cluster A HCFP1 variant (*Fam5^{snv/snv}*) (Extended Data Fig. 5a-e). Through the optimization of a recently published protocol for single cell CUT&Tag, we now show formally that NR2F1 binds to the predicted consensus site in cRE2 in WT r4MNs at E10.5 *in vivo*, and this binding is dramatically reduced in r4MNs derived from *Fam5^{snv/snv}* embryos. Facial motor whisking behavior, FBMN development, and *Gata2* expression were, however, unperturbed in the *Fam5^{snv/snv}* mice (Fig. 3j,k; Extended Data Fig. 5a-e). While we do not yet know why the *Fam5^{snv/snv}* mice did not display changes in r4MN development, we note that NR2F1 also binds to cRE1 in mice and is not predicted to do so in humans, and this may alter its use between the two species. This would explain why the humanized cRE1 duplication has a phenotype while the SNV does not. This is not so unexpected despite sequence conservation, because TF occupancy, especially for distal elements, is not always conserved between human and mouse (Cheng et al., Nature 2014 PMID: 25409826)

3. The authors indicate that cRE2 is occupied by NR2F1 in NCC derived iPSCs and shows an ATAC-seq signal in a neuroblastoma cell line. These cell types do not appear to be relevant for investigating cRE2 occupancy, especially since they express GATA2. The authors should evaluate other publicly available genomic datasets (ENCODE) that are more biologically relevant to the cell types of interest (embryonic hindbrain), for chromatin marks, ATAC-seq peaks, etc., associated with gene regulatory activity on cRE2, or generate these data themselves.

We have now generated single cell CUT&Tag data specifically for embryonic r4 *Isl1^{MN}-GFP⁺* motor neurons from WT and *cRE1^{dup/+}* mice. These data establish that NR2F1 binds cRE2 and that this binding is attenuated in the cRE2 SNV mouse model (Revision Fig. 4I-m).

4. It is difficult to compare binding affinities with the different SNVs in Figure 3. Dose-response curves should be performed by varying the amount of radiolabeled probe containing each SNV exposed to a constant amount of NR2F1 protein and quantifying band intensity as a measure of NR2F1/cRE2 complex formation.

While we respect this request, the EMSA was not intended to be quantitative or to compare SNVs to one another, but simply to determine if NR2F1 binds to cRE2. Given the addition of the CUT&Tag data in the correct cell type at the correct embryonic age (Revision Fig. 4I-m), we now provide *in vivo* evidence that a Cluster A SNV reduces the binding of NR2F1 to cRE2.

5. Candidate TFs that bind cluster B were not identified through bioinformatic searches. The authors should consider mining their scRNA-seq data for TFs and test their ability to bind cluster B in the presence and absence of SNVs. Alternatively, they could perform EMSAs with nuclear extracts from mouse embryos and identify the DNA binding protein by mass spectrometry.

We examined the mouse cRE2 sequence using RVista to identify consensus transcription factor binding sites. While the region of Cluster A contained predicted consensus sites shared by NR2F/COUP-TF, PPARA, PPARG, HNF4G and, to a lesser extent, THRA nuclear receptor transcription factors, only expression of *Nr2f1* and *Nr2f2* was detected in the r4MN scRNAseq data.

Notably, we did not identify any transcription factor binding sites that were disrupted by the Cluster B variants. We did find a TCF4 consensus site immediately adjacent to Cluster B, and confirmed *Tcf4* expression early in the r4MN developmental trajectory (see Response Fig. 1 below; not included in manuscript). We also screened the wild type scRNAseq data broadly for transcription factors enriched in FBMNs and found that *Shox2* (Extended Data Fig. 6b,c), as well as *Zfhx3*, *Zfhx4*, *Tox*, and *Pou3f1* are enriched in this population and could participate in regulating *Gata2* expression. Since only 2-5% of the cells in our embryonic hindbrain dissociations are r4MNs, we are not confident that EMSA would have the sensitivity to detect TF binding using nuclear extracts from these preparations. Therefore, we feel that testing these candidates for Cluster B binding is beyond the scope of this study. We do, however, continue to consider approaches to understanding how Cluster B could function in *Gata2* gene regulation.

Response Figure 1: Candidate cRE1 Cluster B interacting TFs. **a.** Alignment of mouse and human Cluster A,B region. Predicted binding sites for TCF4 and NR2F1 are indicated by orange bars. **b.** Feature plots for *Tcf4*, *Nr2f1*, and *Gata2* in E10.5 r4MN trajectory clusters from WT (upper row) and *cRE1^{dup/+}* (lower row) embryos. *Nr2f1* and *Gata2* features plots are included in the manuscript.

6. In Fig. 6i, Cluster B mutations do not appear to influence cRE2 dependent silencing activity in migrating FBMNs. The minimal X-gal staining outside of r4 in the one embryo (out of 4) is not any different from that displayed by embryos with wild type cRE2 sequence (compare embryo 2 in Fig. S5e with embryo 3 in Fig. S5g). The authors should be more transparent in describing this data and refrain from overinterpreting these results.

We thank the reviewer for highlighting this. We have added additional transgenics, tested the effect of cRE2*A and cRE2*B alone (Extended Data Fig. 3h,i), and included a more accurate description of the effect of cRE1+cRE2*A (Fig. 4h, Extended Data Fig. 3f). Indeed, among the eight cRE1+cRE2*B embryos, only one of the tandem embryos shows a pattern consistent with partial loss of silencing, whereas the others are white (3 of the singles) or are similar to cRE1+cRE2 WT. We have now modified the results section and expanded the discussion as follows:

Results, page 15: The effect of CRE1+CRE2*B was less clear, as it no longer attenuated the signal in only 1 of 8 embryos tested, which was an embryo with a tandem insertion (Fig. 4i, Extended Data Fig. 3g). Interestingly, expression of cRE2*A or cRE2*B alone showed some neuronal signal in tandem but not single transgenic embryos (Extended Data Fig. 3h,i). Similarly, cRE1+cRE2*A transgenic embryos showed an overall stronger and more intricate lacZ pattern compared to cRE1+cRE2 (Extended Data Fig. 3d,f).

Discussion, pages 26: Finally, introduction of the cRE2 SNVs in our lacZ assay unveils enhancer activity, likely through the opportunistic binding of other TFs which could also vary between mice and humans⁷⁹. For example, in humans but not in mice, there is a potential homeobox consensus sequence, TAAT, 20 bp upstream of Cluster A that could become more accessible by loss of the NR2F1 binding site. We do not yet know if introducing a single cRE2 Cluster A SNV in the lacZ assay would enhance β -galactosidase expression, or if this in vivo enhancer activity contributes to the HCFP1-SNV phenotype

and

We have not yet determined the mechanism of the Cluster B SNVs. *In silico* analysis predicted few if any TF consensus sequences in Cluster B WT sequence. By EMSA, Cluster B SNVs did not alter NR2F1 binding and, by lacZ, they had less effect on β -galactosidase expression, consistent with reduced penetrance of Cluster B variants.

7. Of the three classes of variants associated with HCFP, only mutations in Cluster A have a clearly proposed mechanism of action. The model depicted in Fig. S4 is confusing and not representative of the data as it does not show the contribution of cRE2 to target gene repression in the wild type state. Moreover, the consequence of the duplication on repressor activity is not well described. The authors should consider the possibility that a rate limiting trans factor required for cRE2 mediated silencing is diluted below a critical threshold in the presence of the duplication.

For the revision, we have outlined possible gene regulatory mechanisms underlying the etiology of HCFP1. In the case of human Cluster A SNVs, we believe alterations of the cRE2 NR2F1 binding site in humans reduces cRE2 silencing activity, and this would cause (in humans) an expansion of *Gata2* expression and increased IEE development at the expense of FBMNs (Fig. 4h,n, Fig. 7k). We have not found any conserved TF binding sites similarly disrupted by the Cluster B SNVs, but cryptic TF binding sites could still overlap with these variants. Using a range of approaches, we now have evidence that the HCFP1 duplications amplify two enhancers and one silencer that collaborate to control *Gata2* expression, and the overall numerical amplification of the enhancer activities in the duplications likewise causes a net increase in *Gata2* expression and IEE production at the expense of FBMNs (Fig. 5-7, Extended Data Fig. 5-9). Additionally, by generating a *cRE1^{dup}* mouse, we have shown that altering this balance can lead to facial weakness phenotype in the mouse as well, supporting the proposed model.

We agree with the reviewer that the schematic in Figure S4 of the original submission was not clear and we have removed it from the revised manuscript. In addition, we have edited and developed new schemas for the *Gata2* and *Gata3* conditional knock-outs (Fig 3i) and the *cRE1^{dup+/-}* mice (Fig 6i-k). We have then summarized our HCFP1 findings in schematic form in Fig. 7k. This summary includes a depiction of the noncoding changes in the *Gata2* regulatory regions: in stage 2 WT (left side), there are green and blue arrows from cRE1 and cRE3, respectively, to the *Gata2* promoter indicating transcriptional activation, and red blunted lines from cRE2 to the *Gata2* promoter indicating transcriptional repression. In Stage 2 in the mutant (right side) the graphics highlight how loss of cRE2 or duplication of cRE1 and cRE3 (2

enhancers) and cRE2 (1 silencer) could have the net effect of increasing *Gata2* expression. This schematic then also provides an anatomic depiction of IEE and FBMN development in WT (left) and mutant (right) embryos based on Fig. 6 histology and cell counts. We hope this is no longer confusing.

Although an interesting hypothesis, our new data do not support a model in which a limiting silencing factor binds to cRE because duplication in mouse of human *cre1* alone leads to the HCFP phenotype. Instead, these data support the model above in which duplication generates an enhancer-silencer imbalance.

8. What is the proposed mechanism of cRE2 mediated silencing of GATA2? Does it block cRE1 and cRE3 from contacting/activating the GATA2 promoter?

We do not yet know the mechanism of cRE2 silencing of *Gata2*, but our data suggests it is mediated by binding of NR2F1. One possible mechanism is as the reviewer described; NR2F1 or another transcription factor(s) that binds to cRE2 and mediates its silencing activity could, through DNA looping, have antagonistic protein-protein contact with factors bound to cRE1 and cRE3 that mediate *Gata2* enhancer activity. We do not feel that single cell HiC technology could resolve this question due to the relatively short distances between these regulatory elements. We have included the following text describing this model:

Discussion, page 25: While the silencing mechanism remains unknown, cRE1-3 and *Gata2* are within the same regulatory region and they might compete for binding to the *Gata2* promoter.

Minor concerns:

1. Why do some HCFP1 patients show abnormal ABR and hypoplastic cochlear vestibular ganglia (VIIIth cranial nerve)?

We are not sure if there is a misprint that we have missed, but while acoustic stapedial reflexes (ASRs) are abnormal (the stapedius muscle is innervated by the facial nerve), auditory

brainstem responses (ABRs) are normal in everyone tested (as per Table 1). Similarly, while VII is hypoplastic, VIII (and other cranial nerves) is normal by MR imaging (as per Fig 2, Table 1, Supplementary Clinical Note).

2. The authors refer to cRE1, 2 and 3 as a stretch enhancer, however these enhancers do not appear to fit the original definition, which stipulates that stretch enhancers are clusters of enhancers greater than or equal to 3kb with no gaps in signal (Parker et al., 2013).

The stretch enhancer definition from Parker*, Stitzel*, et al., 2013 is based on the chromatin state segmentation output from ChromHMM and indeed stipulates uninterrupted enhancer chromatin states ≥ 3 kb. In Extended Data Figure 2b (top track), we show that in SK-N-SH neuroblastoma cells, there are uninterrupted enhancer regions on Chr 3 of 14.8kb in length encompassing cRE3 and another region 3.4kb in length encompassing cRE1 and cRE2. So, all of these cRE's are stretch enhancers. Around these SK-N-SH stretch enhancer chromatin states are active promoter and active transcription chromatin states, which indicate the overall high regulatory activity of this region in SK-N-SH cells. The largely repressive chromatin state of the corresponding region in a wide range of other tissues and cells (remaining tracks) highlights how cell-type specific epigenomic states could potentially influence cRE activity and *GATA2* expression.

We have added this more detailed explanation, and cited Parker, Stitzel et al, in the Extended Data Fig. 2b legend.

3. Fig. S3. Arrows should be added to indicate the supershiped complex. In the last lane of Fig. S3A (both left and right panels) a '+' should be added to the pWT – IRDye 700 row.

Thank you. We have added arrows to what is now Extended Data Fig. 4, and have corrected the error. The figure and legend have been edited.

4. *The following statement suggests that lacZ expression driven from cRE1 is similar to Gata2. “We found that reporter expression driven by cRE1 alone marked the region of r4 MN precursors and migrating FBMNs, as well as regions of the midbrain and spinal cord (Figure 6b,d, Supplementary Figure 5a), similar to published data and Gata2 expression pacern30.” However, since Gata2 is downregulated in migrating FBMNs, which express lacZ, this statement is not entirely accurate.*

We thank the reviewer for highlighting this inconsistency in the text. We have edited the Revision text to read:

Results, page 14: We found that β -galactosidase expression driven by cRE1 alone marked the region of r4MN precursors and migrating FBMNs, as well as regions of the midbrain and spinal cord (Fig. 4b,c, Extended Data Fig. 3a), similar to published data³².

5. *Methods describing the generation of lacZ reporter knock-ins should be included in the manuscript.*

The generation of WT and mutant lacZ reporter constructs and their injection into oocytes is now outlined in the Methods section in greater detail, including a description of, and citation for, the enSERT method of reproducible and efficient CRISPR/Cas9-mediated transgene targeting used in this study.

6. *According to the results shown in Fig. S5, a single (S) or tandem (T) copy of the transgene was introduced into a site in the mouse genome. This information should also be included in Fig. 6.*

We have now indicated in Revision Fig. 4a-i and Revision Extended Data Fig. 3 if an embryo is Single, White Single, Tandem, or White Tandem.

7. Fig. 6i. According to the text and figure legend, embryos expressing this reporter construct contain Cluster B mutations, not Cluster A mutations, as labeled in the figure.

We are grateful to the Reviewer for finding the error in labeling Original Submission Figure 6i. The corresponding data is now correctly labeled in Revision Fig. 4i.

Reviewer #4:

The manuscript entitled “Non-coding variants altering a conserved cis-regulatory element cause dominant hereditary congenital facial palsy” by Di Gioia et al., describes the characterization of new autosomal dominant genetic alterations in 9 families manifesting hereditary congenital facial palsy (HCFP). Interestingly, these alterations affect cis-regulatory elements of the GATA2 gene, encoding a transcription factor playing an important role during early stages of haematopoiesis, and also in several neuronal specification pathways, in particular neuron development in rhombomere 4. These neurons constitute the cranial nerves nVII (Facial branchial motoneurons, FBMNs) and nVIII (vestibulo-acoustic or Inner Ear Efferent, IEEs). In seven families, six single nucleotide variants are found in a conserved element, referred to as cRE2. In addition, a tandem repeat of a genomic region including cRE2, and two further GATA2 cis-regulatory elements, cRE1 and cRE3, were identified in two different families. Three of the 6 SNV overlap a binding site for the NR2F1/COUP-TF1 transcription factor. On these bases, the authors have built complementary experimental strategies to provide evidence that 1) NR2F1 plays an important role in the dichotomy of r4 neuronal development into IEEs and FBMNs, favoring the arising of FBMNs by repressing IEEs; 2) IEEs develop first and FBMNs development is linked to IEEs repression, and 3) repression of IEEs relies on the silencing activity of cRE2, itself depending on NR2F1 binding. The overall data interpretation by the authors is that NR2F1-mediated repression plays an important role in the sequential development of IEEs and FBMNs, allowing the emergence of FBMNs by repressing IEEs.

Due to the wide use of exonic sequencing, the identification of hereditary genetic anomalies on cis-regulatory elements responsible for disease transmission is rare, but has proved to be important to refine the understanding of cellular specific gene regulation and function. In this regard, the heavy genetic studies provided in this manuscript are original and of high interest. They are worth consideration for publication in Nature Genetics, provided the authors revise

their functional studies and the misleading interpretation of some of them. The edition and presentation of the manuscript are generally very satisfactory. My major comments are mainly related to experimental approaches, inappropriate or incomplete interpretation of data, and important information missing, in particular on some experimental procedures.

1. To access NR2F1 binding to cRE2, the authors rely on different types of data:

- Available datasets generated with neuroblastoma cells or neural crest cells produced from human iPSCs *in vitro*, which indicate that NR2F1 is able to bind to cRE2
- Electrophoretic Mobility Shift Assays (EMSA), used by the authors to assess the impact of the studied SNVs on NR2F1 binding to the cRE2 cluster A. These experiments were performed with nuclear extracts of HeLa and HEK293T cells overexpressing NR2F1.

Given that the binding of a transcription factor to its DNA specific target site largely depends on the cellular context (interaction with cell-type specific partners, proper expression level, post-translational modifications), the presented data are not sufficient to infer a binding of NR2F1 on cRE2 in rhombomere r4 neuroprecursors. To strengthen this idea and eventually provide further information helping better understanding of the mechanisms underlying the cRE2 repressive function, I strongly recommend the authors to repeat EMSA with nuclear protein extracts prepared from hindbrains collected at E10-E10.5, when NR2F1 activity has become critical. Given the low protein amount required for each test and the easy mechanical dissociation of the neural tube cells at this early developmental stage, these experiments are quite feasible. Furthermore, they will be more appropriate to determine whether further factors bind to the probe, in particular to the cluster B. Ideally, performing methylation protection assays with hindbrain extracts and wild type/cluster A or B mutant probes would be very informative. With this regard, the core of a putative homeobox binding site (TAAT) is to be noted in the close vicinity of the NR2F1 site. In any case, the possible interference of this site occupancy with NR2F1 binding should be discussed, according to the available literature on the Hox genes expressed in r4 and their respective functions.

In place of additional EMSA with r4MN nuclear protein extracts, we have used scCUT&Tag to confirm that NR2F1 binds to cRE2 in E10.5 specifically in r4MNs *in vivo* and that the Fam5 SNV disrupts this interaction (Fig. 4I-n) (please see response to related Reviewer 1 comment above). We are grateful for the observation of the TAAT box close to the NR2F1 consensus binding site, and have incorporated this information in the Discussion of how loss of NR2F1 binding could allow opportunistic binding of other transcription factors that could disrupt cRE2-mediated *Gata2* gene regulation.

Discussion, page 26: Finally, introduction of the cRE2 SNVs in our lacZ assay unveils enhancer activity, likely through the opportunistic binding of other TFs which could also vary between mice and humans⁷⁹. For example, in humans but not in mice, there is a potential homeobox consensus sequence, TAAT, 20 bp upstream of Cluster A that could become more accessible by loss of the NR2F1 binding site. We do not yet know if introducing a single cRE2 Cluster A SNV in the lacZ assay would enhance β -galactosidase expression, or if this in vivo enhancer activity contributes to the HCFP1-SNV phenotype.

2. GATA2/GATA3/NR2F1 expression studies:

In Figure 4, the data show some GATA2 positive/GATA3 negative (panel s, v, respectively E9.5 and E10.5) and GATA2 positive neurons in FBMN (panel ab, E12.5). What is the nature of these cells? Further, I am very surprised by the ventral axon fascicle emerging from FBMNs and indicated by open arrowheads on panel ab, ac and ad. This is very strange, since, as IEEs axons, FBMNs axons exit in r4, not ventrally in r6, where only the FBMNs cell bodies secrete. Hence, the FBMN axon fascicle should be directed dorsally, as it is shown at E10.5 in panels v-x. Is it possible that the picture is taken more posteriorly than at the r6 level? If this is the case, pictures at the r6 level should be provided. Is there any another explanation?

Thank you for highlighting aspects of the original expression studies that require clarification. In our revised manuscript, we identify a population of GATA2^{ON} cells that are interspersed with the ISL1^{ON};GATA2^{OFF} FBMNs in the WT at E12.5 (Extended Data Fig. 7c-h). We show these GATA2^{ON} cells do not coexpress ISL1 and are therefore not of a motor neuron subtype (Extended Data Fig. 7g-h). We provisionally identify these cells as GATA2^{ON} interneurons.

We also appreciate the opportunity to clarify the E11.5-E12.5 histology data from the first submission and highlight our efforts to simplify and focus the presentation in the Revision. In Figure 4y-ad of the original submission, we showed the lateral growth of combined IEE CNVIII and FBMN CNVII axons within the hindbrain, and indicated with open arrowheads the point where the nerve splits into dorsal IEE and ventral FBMN branches (Karis et al J Comp Neurol 2001, Figure 3d-f). The data in these panels were at the level of the r4 progenitor zone, a point we did not make clear in the original text and legend. That said, in order to include the essential histology of the *Gata2* cKO, *Gata3* cKO, *cRE1*^{dup/+}, and *Fam5*^{SNV} mice, we have removed the *Isl-GFP* intraparenchymal axon staining from the figures as they did not add critical data.

3. r4 neuron birthdating:

The authors tried to determine whether IEEs and FBMNs develop in a specific order by performing birthdating experiments, consisting in analyzing the fate of neuronal precursors having incorporated EdU at sequential development stages. EdU was injected at E8.5, E9.0, E10.0 and E10.5, and the neuronal fates were analyzed at E14.5.

- Rather than the total number of EdU positive nuclei, it would be more informative to provide the percentage of EdU positive cells relative to the number of DAPI stained nuclei in each neuron population. This would better reflect the dynamic of each population development.

- EdU exposure from E8.5 comes up with no EdU positive neurons, and this is consistent with high cell proliferation leading to high dilution of EdU within the derived cell lineage and with the fact that the author took into account nuclei exhibiting EdU staining greater than 50%. Fate analysis at E14.5 is far too late, and these data are meaningless. I propose them to be removed.

- Regarding EdU exposure from later stages, I consider that the data have been misinterpreted. Indeed, the author show clearly in Fig. 4c that IEEs exit cell cycle earlier (E9-E9.5), which is consistent with the small size of this neuron population, and lack of EdU detection after exposure from E10.5. About FBMNs, EdU detection after EdU exposure from E10 and E10.5 and not from E9 cannot be explained by a switch in the fate, but rather, again, by EdU dilution resulting from high proliferation between E9 and E10. As indicated by the bending curve (figure 4c), cell proliferation decreases from E10, which allows EdU detection in some FBMNs at E14.5.

For these reasons, the provided data do not support the hypothesis of a sequential development of IEEs and FBMNs neurons, but simply that IEEs production is completed earlier, which is more consistent with the literature. Indeed, none of the published data on this topic exclude any function of GATA2/GATA3 in the development of FBMNs, and related citations in the manuscript should be revised accordingly.

To properly assess this issue, the authors have to perform EdU injections between E9 and 10,5 and analyze neuronal fates no later than 24 hours after each injection. For better interpretation, I recommend to determine the percentage of EdU positive neurons in each neuron population at each time point.

We appreciate the reviewer's analysis of the conclusions we drew from our birthdating experiments, but respectfully disagree with aspects of it. Our goal of EdU labeling is to identify

the cells that underwent their **terminal cell division** during the EdU pulse. These cells are indelibly marked with EdU; tissue can be harvested days, weeks, or months later and cells that were born (i.e. underwent their terminal cell division) during the pulse can be identified by their intense EdU staining. This is based on the following:

- EdU labels mitotically active cells and remains bioavailable in the CNS for approximately 2 hours following injection (so all cells that divide within that 2 hour window are labeled).
- Cells that are labeled during that window and then continue to divide will dilute the label, decreasing the EdU intensity by ~50% with each subsequent division. Thus, these offspring will not have an intense EdU label and will not be considered born during the pulse (because they are born later).
- Cells that are postmitotic will not incorporate EdU (because they were born prior to the pulse).
- Cells that undergo their terminal round of DNA synthesis preceding cell cycle exit during the time window of the EdU pulse will display the greatest incorporation of the label and are identified as those born during the window of the EdU pulse.
- EdU and BrDU incorporation remains indelible and invariant following terminal cell division and can be assayed weeks to months after injection (see, for example: Miller and Nowakowski, Brain Research 1988, PMID 3167568; Chehrehasa, J Neuroscience Methods 2009, PMID 18996411; Landy et al, Dev Biol 2021, PMID 34352273).

By providing a pulse of EdU at E9.25, E10.0, and E10.5 we most strongly labeled those cells that had their terminal cell division at each of these ages (and thus were “born” at that age). Because our goal requires us to know what the cells born at each of these ages become, we chose E14.5 as the age to analyze the cells born at these earlier timepoints. At E14.5, the IEEs and FBMNs have migrated to separate hindbrain compartments in WT embryos and GATA2 is still highly expressed in the IEEs (it subsequently is downregulated) (Fig. 6a,a”,b,b’). This permitted us to assign an IEE or FBMN identity - based on location and/or gene expression - to the cells born at each of these ages. Analyzing these data no more than 24 hours after each injection would not permit us to define the cell type as convincingly. We have added this explanation to the text as follows:

Results, page 21-22: To determine if IEEs were born prior to FBMN and if the 7.7-fold increase in the proportion of IEEs in the *cRE1^{dup/+}* vs. WT embryos was caused by an extension of the IEE birth epoch, we applied the birthdating label 5-ethynyl-2'-deoxyuridine (EdU) *in utero* to litters containing both WT and *cRE1^{dup/+}* embryos across an E9.25-E10.5 developmental time course. We then classified and counted E14.5 EdU-positive cells as IEEs (ISL1^{ON};GATA2^{ON}) or FBMN (ISL1^{ON};GATA2^{OFF}), regardless of position (Fig. 6p). EdU indelibly marks cells that undergo terminal cell division during the EdU pulse. At E14.5, IEEs and FBMNs have migrated to separate hindbrain

compartments in WT embryos and GATA2 is still highly expressed in the IEEs, permitting us to assign an IEE or FBMN identity to the cells born at each of these ages.

The reviewer suggested that we “provide the percentage of EdU positive cells relative to the number of DAPI stained nuclei in each neuron population”. While indeed this would “better reflect the dynamic of each population development”, this was not our goal. Instead, our goal was to determine whether an r4MN born on a given day became an IEE or FBMN, and so we calculated what % of total cells born on a given injection day became IEE and what % became FBMN (Figure 6p). This confirmed our previous finding of initial IEE birth followed by a switch to FBMN birth. We now also find that in *cRE1^{dup/+}* embryos, r4MNs fail to complete this transition, with IEEs still comprising a significant proportion of r4MNs at E10.5; this indeed supports the hypothesis that these cells share a progenitor pool and that there is a delay in the IEE to FBMN shift in HCFP1. Importantly, we multiplexed the birth dating with a semiautomated, rapid analysis pipeline that allowed us to score the molecular and positional identity of *ISL1^{ON};GATA2^{ON}* IEEs and *ISL1^{ON};GATA2^{OFF}* FBMNs in the region of ancestral r4-r6 at E14.5. These cell counts quantify the increase in *ISL1^{ON};GATA2^{ON}* IEEs and decrease in *ISL1^{ON};GATA2^{OFF}* FBMNs in the *cRE1^{dup/+}* embryos. They also show no statistically significant difference in the total number of r4MNs in *cRE1^{dup/+}* E14.5 hindbrains compared to WT (Fig. 6l), indicating that r4MN cell death did not increase in the mutants despite drastic changes in r4MN migration and identity.

We agree with the reviewer’s assessment of the developmental timeline indicated by our data, in that we show IEEs exit the cell cycle earlier (E9-E9.5) than FBMNs (E10.5), and that the exclusion of FBMN labeling at ~E9.25 indicates the FBMN progenitors continue to divide between E9 and E10. We submit, however, that these data are consistent with a model in which r4MNs initially generate postmitotic r4MNs that are dispatched down an IEE developmental path, but after subsequent cell divisions this r4MN progenitor pool executes a developmental switch and instead directs their progeny down a path to FBMN identity.

Finally, we agree with the reviewer that the existing literature assigns a role for *Gata2* in determining both IEE and FBMN identity. The early published studies that showed FBMN migration defects in constitutive *Gata2* and *Gata3* knockouts were potentially confounded by the overall deterioration of these embryos in advance of early embryonic lethality due to disrupted hematopoiesis. To circumvent this lethality and test our hypothesis that *Gata2* acts as an IEE but not a FBMN determinant, we have now generated viable *Gata2* and *Gata3* conditional cranial motor neuron knockouts and show that while both genes are required for IEE development, they are dispensable for FBMN formation (Fig. 3c-h’). We submit that this is consistent with a *Gata2*-mediated IEE to FBMN developmental switch.

Of note, in revision we omitted the E8.5 time point because no r4MNs were labeled by those treatments, and we shifted the E9.0 injection to E9.25 because labeling of r4MNs at E9.0 had become sparse and uninformative under the prevailing housing and breeding conditions. We also increased the stringency of our EdU labeling criteria, with r4MNs marked as EdU positive if approximately 90% of the ISL1 staining overlapped with that of EdU.

Importantly, since NR2F1 has been largely involved in functions linked to neuronal migration, the authors should take into account the hypothesis according to which impaired NR2F1 binding to cRE2 could result in blocking FBMN in r4, and in the death of part of them. Counting IEEs and FBMNs neurons, and checking survival and migration of FBMNs in NR2F1 mutant embryos, or in transgenic embryos carrying cRE2 mutated in cluster A, would be also important to gain insight into the function of NR2F1 and the impact of its binding to cRE2 during neurogenesis r4 .

We thank the reviewer for their careful consideration of our model and additional ways to test it. Please see our response immediately above and also our response to a related query from Reviewer 1. While we have not yet examined r4MN development in *Nr2f1* knockout embryos, we have generated and examined mouse models for an HCFP1-causing SNV (*Fam5^{snv/snv}*) and duplication (*cRE1^{dup/+}*).

Hindbrain neuroanatomy and r4MN development are unchanged in the *Fam5^{snv/snv}* embryos (Extended Data Fig. 5b-e) and they do not have facial weakness (Fig. 3j-k). However, we do show NR2F1 binding to cRE2 is disrupted by the SNV in r4MNs *in vivo* (Fig. 4l-n).

The *cRE1^{dup/+}* mice have facial weakness, and we quantified their embryonic changes in r4MN migration and molecular identity. As predicted by the reviewer, cell counts at E14.5 documented ectopic ISL1^{ON} motor neurons in r4 in *cRE1^{dup/+}* embryos (expressing a combination of GATA2 and GATA3, or neither) and hypoplastic facial nuclei, but no change in overall r4 MN number compared to WT littermates.

4. Comments on scRNAseq analyses

- Indicating the number of each cell cluster on the trajectory plot in figure 5a would make the data easier to read. As well, it is important to indicate at least one or two master genes for each one of the 13 cell clusters.

We now provide a numerical label to each cluster in our scRNAseq objects (Fig. 5a-c, Extended Data Fig. 6a). Marker genes for each cluster in the *Isl1^{ON}* and/or *Hoxb1^{ON}* scRNAseq object are provided in a dot plot, with a subset also presented as separate feature plots for WT and *cRE1^{dup/+}* data sets (Extended Data Fig. 6b,c).

- The authors mention that clustering from scRNAseq analysis included only *Hoxb1* positive cells, *Isl1* positive or not. *Hoxb1* is indeed widely expressed in r4 progenitors as early as E8.5 and is considered as being the master gene of neuronal fate determination in r4. However, in Figure 5b, the distribution of *Hoxb1* positive cells appears to be low in the progenitor cluster, even lower than that of cells expressing *NR2F1*.

The scRNAseq analysis included all *Hoxb1* and all *Isl1* positive cells (*Hoxb1⁺;Isl1⁺*, *Hoxb1⁺;Isl1⁻*, *Hoxb1⁻;Isl1⁺*). For the previous scRNAseq analysis of WT r4MNs, and our revised analysis of r4MN from both WT and *cRE1^{dup/+}* littermate embryos (Fig. 5, Extended Data Fig. 6), we used FACS to enrich for *Isl1^{MN}-GFP⁺* hindbrain motor neurons. Importantly, we also “spiked in” a far smaller number of *GFP⁻* cells to provide transcriptional context and capture cells early in the r4MN trajectory prior to *Isl1^{MN}-GFP* transgene expression. Filtering for *Isl1* and/or *Hoxb1* expression and reclustering allowed us to maximize the representation of the r4MN trajectory. Since the earliest *Hoxb1⁺* r4 progenitors likely do not express detectable levels of the *Isl1^{MN}-GFP* transgene (based on which the cells were sorted), the only path for their inclusion in our scRNAseq object was as a component of the *GFP⁻* “spike in” cells, of which they would be only a small fraction. Moreover, the *Hoxb1⁺;Isl1⁺* cells in Cluster 1 progenitors are likely midline progenitors from not only r4 but surrounding rhombomeres (r3-r7 mitotic progenitors).

Representation of *Nr2f1⁺* expressing cells exceeds that of *Hoxb1⁺* cells for a similar reason; a large portion of r4MN coexpress *Nr2f1* and *Isl1/Isl1^{MN}-GFP*, leading to a greater number of *Nr2f1* cells in the *Hoxb1⁺;Isl1⁺* object. *Nr2f1* is a vital identity determinant for multiple neuronal subtypes in the developing CNS, including corticospinal motor neurons (Bertacchi et al, PNAS 2010, PMID 20133588). *Nr2f1* has been shown to mark the developing hindbrain at E8.5 (Berenguer et al, PLoS Biology 2020, PMID 32421711) and cranial motor nuclei by E11.5 (Allen Developing Mouse Brain Atlas, <https://developingmouse.brain->

[map.org/experiment/show/100046929](https://www.nature.com/experiments/show/100046929)). These observations are consistent with the broad expression we see in the clusters comprising the r4MN/FBMN scRNAseq developmental trajectory (Fig. 5f).

- GATA3 and GATA2 are clearly expressed in ventral interneurons (ref 26 and 44), and expression should have been detected in at least one of the three interneuron clusters, but it is not the case. Can the authors explain these two results and eventually adjust their clustering strategy? In any case, these points have to be discussed in more details.

We thank the reviewer for highlighting the need to clarify how filtering for *Isl1* and/or *Hoxb1* affects the cell subpopulation representation in the scRNAseq object, including the previously reported ISL1⁻;GATA2⁺;GATA3⁺ interneurons we detected in our hindbrain histology (Fig. 6a-h'; Extended Data Fig. 7g-h). These cells would be excluded from our scRNAseq object because they do not express *Hoxb1* or *Isl1*. Indeed, when we do not limit the sort to *Hoxb1* and/or *Isl1* cells, we identify this *Gata2* / *Gata3* population. We added to the text:

Extended Data Fig 7 legend, page 43: Enlargements of the left side midline r4MN cluster in (c) are shown in (g,h) and in (i) are shown in (m,n). Ventral IEEs are delineated from dorsal FBMs by GATA2 and GATA3 expression in WT (g,h, left), a distinction that is eroded in the *cRE1^{dup/+}* hindbrains as more cells express GATA2 and GATA3 (g,h right). The GATA2^{ON} (red) cells in the FBMN compartment do not express ISL1 and are not FBMs but likely interneurons (h, left, dorsal red cells), and would be excluded from the scRNAseq object (Fig. 5) because they do not express *Hoxb1* or *Isl1*.

5. The authors have assessed the in vivo functionality of GATA2 cis-regulatory elements, wild type and mutated, by generating spare transgenic embryos. They give no information on the procedure used to generate mutated versions of the elements. In the Methods section (p45), the authors mention the injection of sgRNA, cas9 and donor plasmid into pronuclei, with no further information. This procedure, including sgRNAs specificities and how mutations were checked, has to be described in details. The authors provide the number of transgenic embryos expressing the reporter gene LacZ obtained with each construct. In order to provide a better idea of the relative functionality of the cis-regulatory elements, in particular of the impact of the

association of cRE2, wild type or mutated, with cRE1 or cRE3 versus cRE1 and cRE3 alone, on the ratio of the embryos showing lacZ expression to the total number of transgenic embryos would be worth mentioning.

Please see response to similar questions from Reviewer 3 above. Instead of a ratio, we present all embryos (including white embryos) in the Extended Data Fig. 3.

- As now, summary and conclusions are not appropriate and will have to be revised according to the above comments and the new information included in the revised manuscript.

We hope that you agree that, in the course of our revision, we have added single cell epigenomic, transcriptomic, histological, and behavioral analyses of WT and mutant mice that provide multimodal support for the model of HCFP1 arising from the disruption of a GATA2-dependent developmental switch from IEE to FBMN generation.

Minor points

1. In the second line of the last paragraph page 9, in addition to ref 26, the fundamental work of Studer et al. (PMID: 8967950) should be also cited

Thank you for catching this omission. We now include a reference to Studer et al, Nature 1996.

2. Only truncated pictures of EMSA are provided in figure 3 and supplementary figure 3s, showing separately the gel area containing the band ship due to NR2F1 binding and the one with the free probe only. Since binding of further factors cannot be excluded, it would be more correct to show pictures of the entire gel. 2. Fig 6i, cluster A is indicated instead of cluster B

We have now added full, uncropped scans of the gels in Revision Fig. 4k to Supplementary Fig. 1. Gels scans from Licor in the Extended Data Fig. 4 are already uncropped. We are grateful to

the Reviewer for finding the error in labeling Original Submission Fig. 6i. The corresponding data is now correctly labeled in Revision Fig. 4i.

3. Fig 6k, if the authors confirm their hypothesis, the schema 3 should display a lower number of FBMNs in the HCFP1 context.

Thank you for highlighting this opportunity to present a schema that reflects both the anatomy and the stoichiometry of the HCFP mouse models. We present schemas of the ventral hindbrain in WT (left side) and *cRE1^{dup/+}* (right side) at the outset (upper) and conclusion (lower) of the r4MN birth epoch. The relative number of IEEs (red cells) and FBMNs (gray cells) matches their proportions that we counted *in vivo* at E14.5 (Revision Fig. 7k).

4. The authors suggest only Id2 as a possible NR2F1 partner responsible for repression mediation. Another NR2F1 partner that could be taken into consideration is FOG2. Did the author get any information on FOG2 expression in their scRNAseqs?

We thank the reviewer for suggesting the examination of other known NR2F1 interacting factors in our scRNAseq dataset. We found *Zfp2* (FOG2), like *Nr2f1*, was enriched in FBMNs and CNVI/X/XII, and excluded from IEEs (Extended Data Fig. 6b). *Bcl11b/CTIP2* (Chan et al, Nucleic Acids Research 2013, PMID 23975195) and *Ncoa7/Src1* (Montemayor, PLoS 2010, PMID 20111703) have also been reported to bind to NR2F1 and modulate gene regulation, and both genes displayed some enrichment in FBMNs compared to IEEs in our scRNAseq dataset. The potential roles these factors may play in FBMN and IEE development are avenues of inquiry we hope to undertake in future studies.

Decision Letter, first revision:

13th Mar 2023

Dear Elizabeth,

Thank you for submitting your revised manuscript entitled "Non-coding variants alter Gata2 expression in rhombomere 4 motor neurons and cause dominant hereditary congenital facial paresis" (NG-A57143R). It has now been seen by the original referees and their comments are below. The reviewers find that the paper has improved in revision, and therefore we'll be happy in principle to publish it in Nature Genetics, pending minor revisions to satisfy the referees' final requests and to comply with our editorial and formatting guidelines.

Thank you again for your interest in Nature Genetics. Please do not hesitate to contact me if you have any questions.

Congratulations!

Sincerely,

Tiago

Tiago Faial, PhD
Chief Editor
Nature Genetics
<https://orcid.org/0000-0003-0864-1200>

Reviewer #1 (Remarks to the Author):

I want to congratulate the authors for a great revised manuscript that includes several new mouse models and makes a much stronger case for the pathogenicity of the non-coding mutations causing HCFP. The authors have gone far beyond what I asked for in the first round of revision. Overall, the functional data are extremely convincing and the cRE1dup/+ mouse models plus the rescue are simply beautiful and establish the final functional link.

I only have a minor suggestion:

In Figure 5 the authors provide single-cell RNAseq data at multiple stages of wt and cRE1dup/+.. The visualization is not optimal. What does the overlay in (5 a) colored by WT and mutant look like? How about changing the order to a) WT plus numbers b) mutant c) overlay in colored by WT and mutant.

MS

It's not quite clear in the text which cells are shown in 5f? all cells or only a subset?

Reviewer #2 (Remarks to the Author):

Beautiful story! Fantastic science!
I now find the complete story and conclusions compelling!!!
Congratulations to all involved.

Reviewer #3 (Remarks to the Author):

The authors have done a masterful job in revising the manuscript and have fully addressed all of my previous concerns. In particular, the development of new mouse models and inclusion of mechanistic studies (nicely summarized at the beginning of their rebuttal letter) convincingly demonstrate that disruption of cell type specific regulatory sequences controlling GATA2 expression is a major cause of hereditary congenital facial palsy type 1 (HCFP1). The work is novel and should be of tremendous interest to the readership of Nature Genetics.

Reviewer #4 (Remarks to the Author):

The authors have considerably implemented they manuscript with a substantial amount of significant work. By using new strategies such as single cell CUT&tag and further transgenic mouse models, some generated by the authors themselves, they have alleviated the principal weaknesses of the previous version. They took advantage of some of these mouse models to refine strategies for single cell sequencing and clustering, and also to comfort some of their conclusions.

I consider the authors have fulfilled the revisions that I recommended and properly replied to my questions. The significance of EdU tracing experiments has been strengthened by better clarification of the goal of their experimental strategy and new tracing experiments based on EdU injections at more adequate developmental stages. Nonetheless, to ascertain the timing of cell cycle exit of IEE progenitors and the birth dating of FBM neurons, it would have been more appropriate to follow a more common and reliable procedure, which would have consisted in one injection of EdU at E9.5, followed by one injection of BrdU at E10 or E10.5. Using specific antibodies that distinguish both molecules, cells which have exited the cell cycle shortly after the first injection will not incorporate BrdU and remain only EdU positive.

Minor points:

- The authors mention several times in the text the E9.25 developmental stage, whereas in all the figures, E9.5 is indicated instead of E9.25. Given the difficulty to exactly stage mouse embryos at 9.25, and for the sake of homogeneity, I recommend to replace E9.25 by E9.5 in the text.
- In Figure 4b, the indications for rhombomeres r5 and r6 are displaced and should be placed properly.

Author Rebuttal, first revision:

NG-A57143R1 (Tenney et al.) Reviewer #1 (Remarks to the Author):

I want to congratulate the authors for a great revised manuscript that includes several new mouse models and makes a much stronger case for the pathogenicity of the non-coding mutations causing HCFP. The authors have gone far beyond what I asked for in the first round of revision. Overall, the functional data are extremely convincing and the cRE1dup/+ mouse models plus the rescue are simply beautiful and establish the final functional link.

I only have a minor suggestion:

In Figure 5 the authors provide single-cell RNAseq data at multiple stages of wt and cRE1dup/+. The visualization is not optimal. What does the overlay in (5 a) colored by WT and mutant look like? How about changing the order to a) WT plus numbers b) mutant c) overlay in colored by WT and mutant.

We thank the reviewer for their careful consideration of the presentation of the scRNAseq data. Based on these helpful suggestions we have

- *reordered Figure 5a,b to now first present the scRNAseq Seurat object divided into its (a) WT component annotated with cluster identities, followed by (b) the corresponding cRE1^{dup/+} component.*
- *added a new UMAP plot (c) that displays the distribution of WT (blue) and cRE1^{dup/+} (red) cells within the shared Seurat object; opacity of the red cRE1^{dup/+} cell dots that are superimposed on the blue WT cell dots has been reduced by 40% to facilitate visualization of the underlying WT cells.*
- *We have also added axes, the additional FACs figure, etc as requested.*

MS

It's not quite clear in the text which cells are shown in 5f? all cells or only a subset?

We thank the reviewer for identifying this point of needed clarification. We have added labels "E9.5 clusters 1-6" and "E10.5 clusters 1-6" to the corresponding rows of feature plots in Fig. 5f.

Reviewer #2 (Remarks to the Author):

Beautiful story! Fantastic science!
I now find the complete story and conclusions compelling!!! Congratulations to all involved.

Thank you!

Reviewer #3 (Remarks to the Author):

The authors have done a masterful job in revising the manuscript and have fully addressed all of my previous concerns. In particular, the development of new mouse models and inclusion of mechanistic studies (nicely summarized at the beginning of their rebuttal letter) convincingly demonstrate that disruption of cell type specific regulatory sequences controlling GATA2 expression is a major cause of hereditary congenital facial palsy type 1 (HCFP1). The work is novel and should be of tremendous interest to the readership of Nature Genetics.

Thank you!

Reviewer #4 (Remarks to the Author):

The authors have considerably improved their manuscript with a substantial amount of significant work. By using new strategies such as single cell CUT&tag and further transgenic mouse models, some generated by the authors themselves, they have alleviated the principal weaknesses of the previous version. They took advantage of some of these mouse models to refine strategies for single cell sequencing and clustering, and also to confirm some of their conclusions.

I consider the authors have fulfilled the revisions that I recommended and properly replied to my questions. The significance of EdU tracing experiments has been strengthened by better clarification of the goal of their experimental strategy and new tracing experiments based on EdU injections at more adequate developmental stages. Nonetheless, to ascertain the timing of cell cycle exit of IEE progenitors and the birth dating of FBM neurons, it would have been more appropriate to follow a more common and reliable procedure, which would have consisted in one injection of EdU at E9.5, followed by one injection of BrdU at E10 or E10.5. Using specific antibodies that distinguish both molecules, cells which have exited the cell cycle shortly after the first injection will not incorporate BrdU and remain only EdU positive.

Minor points:

- The authors mention several times in the text the E9.25 developmental stage, whereas in all the figures, E9.5 is indicated instead of E9.25. Given the difficulty to exactly stage mouse embryos at 9.25, and for the sake of homogeneity, I recommend to replace E9.25 by E9.5 in the text.

We thank the reviewer for this suggestion for simplifying the text but feel that the precision of the E9.25 birth dating is accurate and would like to leave it as such. We found that EdU injections at E8.5 and E9.0 did not strongly label any r4MNs, while injections at E9.5 marked IEEs and a small number of FBMNs. Injections at E9.25 reliably and strongly labeled IEEs, but very few FBMNs, indicating a stereotyped and brief epoch around E9.25 during which only IEEs emerge from the cell cycle. We submit that the accurate description of our findings is aided by the admittedly nuanced distinction between the E9.25 and E9.5 birth dating time

point. Of note, the reviewer seems to have missed that the birth dating figure does read E9.25 rather than E9.5. The histology was performed at the more canonical E9.5 developmental time point and is labeled as such.

- In Figure 4b, the indications for rhombomeres r5 and r6 are displaced and should be placed properly.

We thank the reviewer for catching the error in labeling the anatomic schema in Figure 4b. The panel has been corrected.

Final Decision Letter:

10th May 2023

Dear Elizabeth,

I am delighted to say that your manuscript "Non-coding variants alter GATA2 expression in rhombomere 4 motor neurons and cause dominant hereditary congenital facial paresis" has been accepted for publication in an upcoming issue of Nature Genetics.

Your paper will be published online after we receive your corrections and will appear in print in the next available issue. You can find out your date of online publication by contacting the Nature Press Office (press@nature.com) after sending your e-proof corrections. Now is the time to inform your Public Relations or Press Office about your paper, as they might be interested in promoting its publication. This will allow them time to prepare an accurate and satisfactory press release. Include your manuscript tracking number (NG-A57143R1) and the name of the journal, which they will need when they contact our Press Office.

Before your paper is published online, we shall be distributing a press release to news organizations worldwide, which may very well include details of your work. We are happy for your institution or

funding agency to prepare its own press release, but it must mention the embargo date and Nature Genetics. Our Press Office may contact you closer to the time of publication, but if you or your Press Office have any enquiries in the meantime, please contact press@nature.com.

Please note that *Nature Genetics* is a Transformative Journal (TJ). Authors may publish their research with us through the traditional subscription access route or make their paper immediately open access through payment of an article-processing charge (APC). Authors will not be required to make a final decision about access to their article until it has been accepted. [Find out more about Transformative Journals](https://www.springernature.com/gp/open-research/transformative-journals)

Authors may need to take specific actions to achieve [compliance](https://www.springernature.com/gp/open-research/funding/policy-compliance-faqs) with funder and institutional open access mandates. If your research is supported by a funder that requires immediate open access (e.g. according to [Plan S principles](https://www.springernature.com/gp/open-research/plan-s-compliance)) then you should select the gold OA route, and we will direct you to the compliant route where possible. For authors selecting the subscription publication route, the journal's standard licensing terms will need to be accepted, including [self-archiving-and-license-to-publish](https://www.nature.com/nature-portfolio/editorial-policies/self-archiving-and-license-to-publish). Those licensing terms will supersede any other terms that the author or any third party may assert apply to any version of the manuscript.

Please note that Nature Portfolio offers an immediate open access option only for papers that were first submitted after 1 January, 2021.

Sincerely,

Tiago

Tiago Faial, PhD
Chief Editor
Nature Genetics
<https://orcid.org/0000-0003-0864-1200>